

# Quantum current and holographic categorical symmetry

**Tian Lan⋆ and Jing-Ren Zhou**

Department of Physics, The Chinese University of Hong Kong ,
Shatin, New Territories, Hong Kong, China

⋆ tlan@cuhk.edu.hk

## Abstract

We establish the formulation for quantum current. Given a symmetry group $G$, let $\mathcal{C} := \operatorname{Rep} G$ be its representation category. Physically, symmetry charges are objects of $\mathcal{C}$ and symmetric operators are morphisms in $\mathcal{C}$. The addition of charges is given by the tensor product of representations. For any symmetric operator $O$ crossing two subsystems, the exact symmetry charge transported by $O$ can be extracted. The quantum current is defined as symmetric operators that can transport symmetry charges over an arbitrary long distance. A quantum current exactly corresponds to an object in the Drinfeld center $Z_1(\mathcal{C})$. The condition for quantum currents to be superconducting is also specified, which corresponds to condensation of anyons in one higher dimension. To express the local conservation, the internal hom must be used to compute the charge difference, and the framework of enriched category is inevitable. To illustrate these ideas, we develop a rigorous scheme of renormalization in one-dimensional lattice systems and analyse the fixed-point models. It is proved that in the fixed-point models, superconducting quantum currents form a Lagrangian algebra in $Z_1(\mathcal{C})$ and the boundary-bulk correspondence is verified in the enriched setting. Overall, the quantum current provides a natural physical interpretation to the holographic categorical symmetry.

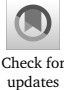

# 1  Introduction

The electric charge is a conserved quantity. Classically, we think that the electric charge is a continuous quantity and we talk about the charge density $\rho$. The global charge $Q = \int \rho$ is conserved. If we divide the whole system into two parts **A** and **B**, and denote $Q_{\mathbf{A}} = \int_{\mathbf{A}} \rho$ and $Q_{\mathbf{B}} = \int_{\mathbf{B}} \rho$, then

$$0 = \Delta Q = \Delta Q_{\mathbf{A}} + \Delta Q_{\mathbf{B}} \implies \Delta Q_{\mathbf{A}} = -\Delta Q_{\mathbf{B}}. \tag{1}$$

The change of charge in one subsystem must compensate that in the other. When the charge in **A** increases, there must be charge flowing from **B** to **A**, i.e., a current. Again one can consider the current density $\boldsymbol{j}$, and we have the famous differential equation for local conservation of charge

$$\frac{\partial \rho}{\partial t} + \nabla \cdot \boldsymbol{j} = 0. \tag{2}$$

However, we know in reality that electric charge is discrete instead of continuous. The above treatment, including the density of charge or current as well as the differential equation, is only an effective approximation at a macroscopic or statistical level.

The case becomes even worse when we consider other symmetries. The electric charge is just the conserved quantity of global $U(1)$ symmetry, taking value in integers (with appropriate unit). The addition of electric charges is the usual addition of integers. We may switch to, for example, angular momentum, which is the conserved quantity of the rotation symmetry, $SU(2)$. The quantum angular momentum takes value in, no longer ordinary numbers, but the representations of the symmetry group $SU(2)$. The addition of angular momentum is also more

complicated than the addition of numbers. For example, the total angular momentum of two spin 1/2's together is the "direct sum" of a spin 0 singlet and a spin 1 triplet: $1/2 \otimes 1/2 = 0 \oplus 1$.

In this paper we give a serious treatment to the local conservation of quantum symmetry charge, which seems long missing in the literature, although the mathematics involved is quite fundamental. The essential technical step is to realize that in a quantum system with symmetry, the symmetric operators are just the morphisms in the tensor (or fusion when the group is finite) category of symmetry charges, or equivalently, symmetric (or invariant) tensor networks. With this in mind, we show that for any symmetric operator crossing two subsystems, a symmetry charge can be extracted which we interpret as the symmetry charge transported by the operator. Moreover, we define the notion of quantum current, which is the collection of symmetric operators that can transport a symmetry charge over a long distance.

It can be seen directly from our definition that a quantum current is an object in the Drinfeld center of the fusion category of symmetry charges. Our work is a generalization of the *Noether theorem*: when the symmetry, which can be discrete, is given, the possible current is fully determined, by computing the Drinfeld center. The local conservation is now written as

$$\mathrm{Hom}_{\mathcal{C}}(Q \otimes X, Y) \cong \mathrm{Hom}_{Z_1(\mathcal{C})}((Q, \beta), [X, Y]_{Z_1(\mathcal{C})}), \tag{3}$$

where $\mathcal{C}$ is the category of symmetry charges and $Z_1(\mathcal{C})$ is the category of quantum currents. The functor on the left hand side $\mathrm{Hom}(-\otimes X, Y)$ compute the ways how $X$ can be converted to $Y$. The internal hom $[X, Y]_{Z_1(\mathcal{C})}$ is an object in $Z_1(\mathcal{C})$ that represents the functor $\mathrm{Hom}(-\otimes X, Y)$. Physically, $[X, Y]_{Z_1(\mathcal{C})}$ is the quantum current providing the universal answer for the ways how $X$ can be converted to $Y$. We have to use the internal hom to compute the quantum current, including the charge difference and how the charge flows, for discrete symmetry charges.

We like to comment on the difference between our formulation and the traditional notion of current in quantum mechanics or current operator in quantum field theory. The current carried by a charged quantum particle is traditionally defined as the charge times the probability current. Based on such notion, one can only conclude that the expectation value of the charge is locally conserved. Similarly, a current operator in quantum field theory is such an operator that its expectation value (correlation function) satisfies a local conservation condition. We consider these traditional notions of current as only "semi-classical", in that (1) the local conservation is only satisfied on average, at a macroscopic or statistical level; (2) they can not be used to deal with the discrete or quantized charge transport in a single quantum mechanical process; and (3) they usually require continuous space-time and continuous symmetry. There are recent works [1, 2] extending the "semi-classical" current to lattice systems. Our formulation, on the contrary, is truly quantum: it can apply to discrete space and discrete symmetry, and can be used to analyse quantized charge transport exactly instead of on average.

On the other hand, quantum currents can be identified with excitations in a topological order in one higher dimension. In fact, our work provides a concrete physical interpretation to the *categorical symmetry* [3–15].

The term categorical symmetry was first proposed in Ref. [3], which aims at providing an invariant shared by gapped phases before and after a phase transition. For example, the 1+1D transverse field Ising model has two gapped phases, $\mathbb{Z}_2$ symmetric and $\mathbb{Z}_2$ symmetry breaking: One exhibits the $\mathbb{Z}_2$ symmetry while the other exhibits the dual $\mathbb{Z}_2$ symmetry. Putting the two $\mathbb{Z}_2$ together, one says that there is a categorical symmetry of 2+1D $\mathbb{Z}_2$ topological order (or toric code model [16]), which has both $\mathbb{Z}_2$ charges and $\mathbb{Z}_2$ fluxes. From a purely mathematical point of view, this categorical symmetry is the Drinfeld center of the fusion category $\mathrm{Rep}\,\mathbb{Z}_2$ of symmetry charges. Because of the boundary-bulk correspondence of topological ordered phases [17–19], one can say, virtually, that categorical symmetry is the topological order in one dimension higher. Because of this relation, in this paper we call such invariant (topological order in one dimension higher) as the *holographic* categorical symmetry, in order to avoid pos-

sible confusion with e.g., fusion category symmetry [20–23] and other similar notions which may also be referred to as categorical symmetry.[1]

However, the physical meaning of holographic categorical symmetry remains mysterious. There are even other names for the same notion, such as symmetry TFT [24] and topological symmetry [25]. Ref. [6, 7] proposed to view holographic categorical symmetry as the transparent patch operators. Based on the idea of *topological Wick rotation* [13], Refs. [5, 15] proposed to view holographic categorical symmetry (or the background category of the enriched category description [8, 11] of gapped phase) as the sectors of non-local symmetric operators. Moreover, in the context of topological order, one can condense the excitations in the bulk topological order to obtain a boundary theory, or fuse defects in the bulk with the boundary theory. Thus, by topological Wick rotation, the algebras and defects in the holographic categorical symmetry (corresponding to the topological order in the bulk) should play an important role in classifying phases and phase transitions (corresponding to the boundary theory). Such point of view has been emphasized in Refs. [4, 7, 12].

Our formulation of quantum currents clarifies the confusion around various concepts regarding holographic categorical symmetry. We give a rigorous definition, Definition 4.13, for what operators qualify as quantum currents. We also give a definition, Definition 4.19, for what quantum currents are superconducting. These definitions are tested in concrete one dimensional fixed-point lattice models. We prove that the superconducting quantum currents in the fixed-point model indeed form a Lagrangian algebra in the Drinfeld center, corresponding to a maximal anyon condensation in one higher dimension, which also determines the fixed-point defects (or excitations) in the fixed-point model. Therefore, we established the correspondence between quantum currents and the holographic categorical symmetry; holographic categorical symmetry is about the transport property of the physical system. We also want to emphasize that the quantum currents can be measured, observed and understood, in the same dimension of the physical system; no fictional one dimensional higher bulk is required.

The paper is organized as follows: in Section 2 we give some necessary preliminary notions and fix our notations; in Section 3 we explain basic techniques on how to manipulate symmetric operators, viewing them as graphs or tensor networks; in Section 4 we motivate the definition for quantum current and introduce some related notions such as the charge transported by symmetric operators and the condensation of quantum currents; in Section 5 we give a rigorous treatment to the renormalization process in 1+1D lattice models; finally in Section 6 we show that in the fixed-point lattice models superconducting quantum currents form a Lagrangian algebra and give a series of examples. We summarize main contents in this paper in Table 1.

Table 1: A summary of contents.

| Physical quantity | Mathematical description | Graphical representation |
|---|---|---|
| Symmetry charge/ Hilbert space on local region $\mathbf{K}$ | $(\mathcal{H}_{\mathbf{K}}, \rho^{\mathbf{K}}) \in \mathcal{C} := \operatorname{Rep} G$ | $\mathcal{H}_{\mathbf{K}}$ |
| General symmetric operator/intertwiner | $\operatorname{Hom}(\mathcal{H}_{\mathbf{K}_1}, \mathcal{H}_{\mathbf{K}_2})$ consists of $f \in \operatorname{Hom}_{\mathbf{Vec}}(\mathcal{H}_{\mathbf{K}_1}, \mathcal{H}_{\mathbf{K}_2})$ such that $\forall g \in G$, $\rho_g^{\mathbf{K}_2} f (\rho_g^{\mathbf{K}_1})^{-1} = f$ | $\mathcal{H}_{\mathbf{K}_2}$ $\boxed{f}$ $\mathcal{H}_{\mathbf{K}_1}$ |

---

[1]In a more general sense, any generalized version of symmetry relating to category theory may be called categorical symmetry.

| | | |
|---|---|---|
| Symmetric operator on bipartite system **A** and **B** | $O \in \mathrm{Hom}(\mathcal{H}_\mathbf{A} \otimes \mathcal{H}_\mathbf{B}, \mathcal{H}_\mathbf{A} \otimes \mathcal{H}_\mathbf{B})$ | $\mathcal{H}_\mathbf{A}$ $\mathcal{H}_\mathbf{B}$ — $O$ — $\mathcal{H}_\mathbf{A}$ $\mathcal{H}_\mathbf{B}$ |
| Transformation on $O$ to extract transported symmetry charge | $O \xrightarrow{\epsilon} \bar{O} = \bar{r}\bar{l} \xrightarrow{\epsilon^{-1}} O = rl$ (Definition 4.2) | $\bar{O} =$ $\mathcal{H}_\mathbf{B}$ $\mathcal{H}_\mathbf{B}^*$ $\bar{r}$ $\mathrm{Im}\,\bar{O}$ $\bar{l}$ $\mathcal{H}_\mathbf{A}^*$ $\mathcal{H}_\mathbf{A}$ |
| Transported symmetry charge from region **A** to **B** | $O{\uparrow_\mathbf{A}^\mathbf{B}} = \mathrm{Im}\,\bar{O}$ | $O =$ $\mathcal{H}_\mathbf{A}$ $\mathcal{H}_\mathbf{B}$ $O{\uparrow_\mathbf{A}^\mathbf{B}}$ $r$ $l$ $\mathcal{H}_\mathbf{A}$ $\mathcal{H}_\mathbf{B}$ |
| Quantum current | $(Q, \beta_{Q,-}) \in Z_1(\mathcal{C})$ (Definition 4.13) Simple objects in $Z_1(\mathrm{Rep}\,G)$: $(C_x, \tau)$ (Appendix A) | $\mathcal{H}_s$ $\mathcal{H}_\mathbf{M}$ $\mathcal{H}_t$ $Q$ $r$ $\beta_{Q,\mathcal{H}_\mathbf{M}}$ $Q$ $l$ $\mathcal{H}_s$ $\mathcal{H}_\mathbf{M}$ $\mathcal{H}_t$ |
| Translation invariant commuting-projector fixed-point model $(\mathbb{Z}, A, -m^\dagger m)$ | *Hilbert space*: $\mathcal{H}_i = A, \forall i \in \mathbb{Z}$, where $(A, m, \eta)$ is a Frobenius algebra in $\mathcal{C}$ *Commuting-projector Hamiltonian*: $H = -\sum_i (m^\dagger m)_i$ *Ground state subspace*: $A$ | $(m^\dagger m)_i =$ $A$ $A$ $m^\dagger$ $A$ $m$ $A$ $A$ |

| | | |
|---|---|---|
| Half-infinite chain model with boundary condition $(\mathbb{N}, A, -m^\dagger m, M, -\rho^\dagger \rho)$ | *Boundary Hilbert space*: $\mathcal{H}_0 = M$, where $(M, \rho)$ is a right $A$-module in $\mathcal{C}$<br><br>*Hamiltonian*: $H = -(\rho^\dagger \rho)_0 - \sum_{i \geq 1} (m^\dagger m)_i$<br><br>*Ground state subspace*: $M$<br><br>*Boundary change*: $A$-module map<br><br>*All boundary conditions*: $\mathcal{C}_A$ | $(\rho^\dagger \rho)_0 =$ (diagram with $M$, $A$, $\rho^\dagger$, $\rho$, $M$, $A$) |
| Finite chain model with two-side boundary conditions $(\mathbf{L} = \{0, \ldots, J\}, A, -m^\dagger m,$ $M, -\rho^\dagger \rho, N, -\lambda^\dagger \lambda)$ | *Boundary Hilbert spaces*: $\mathcal{H}_0 = M$, $\mathcal{H}_J = N$, where $(M, \rho)$ is a right $A$-module and $(N, \lambda)$ is a left $A$-module in $\mathcal{C}$<br><br>*Hamiltonian*: $H = -(\rho^\dagger \rho)_0$ $-(\lambda^\dagger \lambda)_{J-1} - \sum_{1 \leq i \leq J-2} (m^\dagger m)_i$<br><br>*Ground state subspace*: $\operatorname{Im} P = M \underset{A}{\otimes} N$ | $P =$ (diagram with $M$, $N$, $\rho$, $A$, $\lambda^\dagger$, $M$, $N$) |
| Fixed-point defects/ Excitations | *Defect Hilbert space*: $A$-$A'$-bimodule $B$ in $\mathcal{C}$<br><br>*Defect change*: $A$-$A'$-bimodule map<br><br>*Ground state subspace of two defects $_{A''}B'_A$ and $_A B_{A'}$*: $\operatorname{Im} P' = B' \underset{A}{\otimes} B$<br><br>*All defects*: $_A\mathcal{C}_{A'} \cong \operatorname{Fun}_{\mathcal{C}}(\mathcal{C}_A, \mathcal{C}'_A)$ *Excitations*: $_A\mathcal{C}_A \cong$ $\operatorname{Fun}_{\mathcal{C}}(\mathcal{C}_A, \mathcal{C}_A)^{\operatorname{rev}} \cong Z_1(\mathcal{C})_{[A,A]}$ | $P' =$ (diagram with $B'$, $B$, $\rho'$, $A$, $\lambda^\dagger$, $B'$, $B$) |
| Superconducting quantum currents | Internal hom $[A, A] \in Z_1(\mathcal{C})$, which is also a Lagrangian algebra in $Z_1(\mathcal{C})$ | |
| Holographic categorical symmetry | $Z_0(^\mathcal{C}\mathcal{C}_A) \cong^{Z_1(\mathcal{C})} \operatorname{Fun}_{\mathcal{C}}(\mathcal{C}_A, \mathcal{C}_A)$ | |

| Universal model $(\mathbb{Z}, \mathrm{Fun}(G), -(\iota_A \otimes \iota_A)m^\dagger m(\iota_A^\dagger \otimes \iota_A^\dagger))$ for $\mathcal{C} = \mathrm{Rep}\,G$ with trivial $\omega_2$ | Frobenius algebras in $\mathrm{Rep}\,G$ are classified by $(H \subset G, \omega_2)$, where $\omega_2 \in H^2(H, U(1))$ <br><br> For trivial $\omega_2$, $A := \mathrm{Fun}(G/H)$, $\iota_A : A \to \mathrm{Fun}(G)$ (Subsection 6.2) <br><br> Realizing 1+1D spontaneous symmetry breaking phases with unbroken subgroup $H$ for different choices of $H$ | <br><br><br> $\mathrm{Fun}(G)\qquad\mathrm{Fun}(G)$ <br><br> $\iota_A \qquad\qquad \iota_A$ <br><br> $A\;m^\dagger\;A$ <br> $A$ <br> $A\;m\;A$ <br><br> $\iota_A^\dagger \qquad\qquad \iota_A^\dagger$ <br><br> $\mathrm{Fun}(G)\qquad\mathrm{Fun}(G)$ |
| Universal model for $\mathcal{C} = \mathrm{Rep}\,G$ with nontrivial $\omega_2$ | For nontrivial $\omega_2$, $A$ is given by Example F.10 <br><br> Realizing generic 1+1D $G$-symmetric phases labeled by $(H, \omega_2)$ for different choices of $H$ and $\omega_2$ ($H$ is the unbroken subgroup and $\omega_2$ labels symmetry protected topological order under $H$) | |

## 2 Preliminaries and notations

We first review the group representation category and fix our notation for graphical calculus.

**Definition 2.1.** Let $G$ be a compact group. The *group representation category* is the functor category $\mathrm{Fun}(BG, \mathbf{Vec}) =: \mathrm{Rep}\,G$, where $BG$ is the category with one object $\bullet$ and $BG(\bullet, \bullet) = G$, and $\mathbf{Vec}$ is the category of finite dimensional vector spaces over $\mathbb{C}$. Spelling it out,

- An object in $\mathrm{Rep}\,G$ is a pair $(V, \rho)$, called a group representation, where $V$ is a vector space and $\rho : G \to GL(V)$ is a group homomorphism, i.e. there are $g$-indexed invertible linear maps on $V$, denoted by $\rho_g$, such that

$$\rho_{gh} = \rho_g \rho_h. \tag{4}$$

$\rho_g$ is referred to as the group action or symmetry action. An object in $\mathrm{Rep}\,G$ is physically referred to as a $(G$-$)$symmetry charge.

We may use simply $V$ to denote a representation $(V, \rho)$, and write the group action $\rho_g(a)$ as $g \triangleright a$ or $ga$, when the explicit form of $\rho$ is not important for the discussion.

- A morphism between two representations $(V, \rho)$ and $(W, \tau)$ is a linear map $f : V \to W$ that commutes with group actions, $f \rho_g = \tau_g f$ for all $g \in G$. The space of morphisms from $(V, \rho)$ to $(W, \tau)$ is denoted by $\mathrm{Hom}((V, \rho), (W, \tau))$. In particular, the endomorphism space $\mathrm{Hom}((V, \rho), (V, \rho))$ is exactly the subspace of symmetric operators on $V$, which satisfy $\rho_g f \rho_{g^{-1}} = f$. The following terms

  – morphism in $\mathrm{Rep}\,G$,
  – intertwiner, intertwining operator

     – invariant tensor,

     – symmetric tensor,[2]

     – symmetric operator,

will be used interchangeably.

Graphically, an object is represented by a line and a morphism is represented by a node between two lines

$$f : (V, \rho) \to (W, \tau) = \quad \boxed{f} \quad . \tag{5}$$

Composition of morphisms is done from bottom to top

$$gf = \quad \boxed{g} \atop \boxed{f} \quad . \tag{6}$$

The representation category enjoys additional nice structures, one of which is the tensor product. Given two representations $(V, \rho)$ and $(W, \tau)$, their tensor product

$$(V, \rho) \otimes (W, \tau) := (V \otimes W, \rho \otimes \tau), \tag{7}$$

is again a representation with action given by

$$(\rho \otimes \tau)_g = \rho_g \otimes \tau_g. \tag{8}$$

Tensor product is graphically represented by juxtaposing lines

$$(V, \rho) \otimes (W, \tau) = \quad (V, \rho) \quad \Big| \quad \Big| \quad (W, \tau) \quad . \tag{9}$$

There is always the trivial representation $\mathbf{1} := (\mathbb{C}, \mathrm{id})$, $\mathrm{id}_g = \mathrm{id}_{\mathbb{C}}$ for all $g \in G$. $\mathbf{1}$ is the unit of tensor product. For any representation $(V, \rho)$, there is a dual representation $(V^*, \rho^\star)$[3] where the underlying vector space is the dual vector space $V^* := \mathrm{Hom}_{\mathbf{Vec}}(V, \mathbb{C})$,[4] and the group action is $\rho_g^\star := - \circ \rho_{g^{-1}} = (\rho_{g^{-1}})^*$. More concretely

$$\rho_g^\star : V^* \to V^*,$$
$$\varphi \mapsto \varphi \rho_{g^{-1}}, \tag{10}$$

---

[2]In this paper "symmetric" always mean invariant under group action, and never mean invariant under permutation of tensor indices.

[3]Dual group action and dual object, dual morphism are distinct. We denote dual group action of $\rho$ as $\rho^\star$, and dual object of $V$, dual morphism of $f$ as $V^*$, $f^*$ respectively.

[4]When confusion is possible, subscripts are added to clarify $\mathrm{Hom}(-, -)$ in different categories.

graphically represented as

$$
\begin{array}{|c|}\hline \rho_g^\star \\\hline\end{array}
\;=\;
\begin{array}{|c|}\hline (\rho_{g^{-1}})^* \\\hline\end{array}
\;=\;
\boxed{\rho_{g^{-1}}}
\quad . \tag{11}
$$

If one choose a basis of $V$ and the corresponding dual basis of $V^*$, then the matrix representation of $g$ on $V^*$ is the transpose of the matrix representation of $g^{-1}$ on $V$. We can check that the above defined $(V^*, \rho^\star)$ is indeed the dual object of $(V, \rho)$ in $\operatorname{Rep} G$: The pairing between $V^*$ and $V$

$$
\begin{aligned}
V^* \otimes V &\to \mathbb{C}, \\
\varphi \otimes v &\mapsto \varphi(v),
\end{aligned} \tag{12}
$$

is symmetric:

$$
\rho_g^\star \varphi(\rho_g(v)) = \varphi \circ \rho_{g^{-1}} \circ \rho_g(v) = \varphi(v). \tag{13}
$$

The copairing

$$
\begin{aligned}
\mathbb{C} &\to V \otimes V^*, \\
1 &\mapsto \sum_a a \otimes \delta_a,
\end{aligned} \tag{14}
$$

where $\{a\}$ is a basis of $V$ and $\{\delta_a\}$ is the corresponding dual basis $\delta_a(b) = \delta_{a,b}$, is also symmetric

$$
\sum_a \rho_g(a) \otimes \rho_g^\star(\delta_a) = \sum_a \rho_g(a) \otimes (\delta_a \circ \rho_{g^{-1}}) = \sum_a \rho_g(a) \otimes \delta_{\rho_g(a)} = \sum_a a \otimes \delta_a. \tag{15}
$$

Therefore, the pairing and copairing both remain as morphisms in $\operatorname{Rep} G$ and exhibit $(V^*, \rho^\star)$ as the dual object of $(V, \rho)$.

Another structure is the direct sum. Given two representations $(V, \rho)$ and $(W, \tau)$, their direct sum $(V, \rho) \oplus (W, \tau) := (V \oplus W, \rho \oplus \tau)$ is again a representation with action given by

$$
(\rho \oplus \tau)_g = \rho_g \oplus \tau_g \in \operatorname{End}_{\mathbf{Vec}}(V) \oplus \operatorname{End}_{\mathbf{Vec}}(W) \subset \operatorname{End}_{\mathbf{Vec}}(V \oplus W, V \oplus W). \tag{16}
$$

An isomorphism is a morphism invertible under composition. Two representations are isomorphic $V \cong W$ when there is an isomorphism between them; in other words, $V$ and $W$ differ by a basis change which commutes with group actions. A nonzero representation is irreducible (or simple) if it is not isomorphic to a direct sum of two nonzero representations. For a compact group $G$, all finite dimensional representations are completely reducible, i.e. isomorphic to a direct sum of irreducible representations. For reader's convenience, we review the categorical formulation of direct sum here

**Definition 2.2.** In a category whose hom-sets form abelian groups and composition is bilinear, the *direct sum* of two objects $A, B$, if exists, is an object $Y$ together with two pairs of morphisms $p_A : Y \to A$, $q_A : A \to Y$, $p_B : Y \to B$, $q_B : B \to Y$ satisfying

$$
\begin{aligned}
p_A q_A &= \operatorname{id}_A, & p_B q_B &= \operatorname{id}_B, \\
p_A q_B &= 0, & p_B q_A &= 0, \\
& q_A p_A + q_B p_B &&= \operatorname{id}_Y.
\end{aligned} \tag{17}
$$

We refer to $p_A, p_B$ as projections[5] and $q_A, q_B$ as embeddings.

**Remark 2.3.** Such an object $Y$ is simultaneously a product and coproduct of $A$ and $B$, and by the universal property of limit, is unique up to unique isomorphism. Thus it is fine to talk about *the* direct sum and denote it by $A \oplus B$.

Rep $G$ is also a unitary category with unitary structure given by the usual Hermitian conjugate.

**Remark 2.4.** In a unitary category such as Rep $G$, it is always possible to choose $q_i = p_i^\dagger$.

Now suppose that

$$V \otimes W \cong \oplus_i X_i \,, \tag{18}$$

where $X_i$ are irreps (irreducible representations). In more elementary words, the above means that after a change of basis of $V \otimes W$, the group actions all become block-diagonal. We depict the projection map $p_i$ from $V \otimes W$ to $X_i$ by

$$p_i : V \otimes W \to X_i = \qquad\qquad , \tag{19}$$

and the embedding map by

$$q_i = p_i^\dagger : X_i \to V \otimes W = \qquad\qquad . \tag{20}$$

The normalization is taken to be

$$p_j p_i^\dagger = \delta_{ij} \mathrm{id}_{X_i} \,, \quad \sum_i p_i^\dagger p_i = \mathrm{id}_{V \otimes W} \,, \tag{21}$$

so that $p_i, p_i^\dagger$ exhibit $V \otimes W$ as a direct sum of $X_i$'s.

Two fundamental constructions will be useful later and we fix the notation here:

**Definition 2.5.** Given a set $S$, the vector space of finite *formal* linear combinations of elements in $S$, is called the *free vector space* on $S$, and denoted by $C(S)$.

**Definition 2.6.** Given a subset $T$ of a vector space $V$, the subspace of all linear combinations of vectors in $T$, is called the *space spanned by $T$*, and denoted by $\langle T \rangle$.

**Remark 2.7.** $S$ is automatically a basis of $C(S)$. $\langle T \rangle$ is the smallest subspace of $V$ containing $T$, and $T$ is not necessarily a basis of $\langle T \rangle$.

---

[5]A projection may also refer to an operator that squares to itself. $q_A p_A$ and $q_B p_B$ are projections in this sense. In this paper we use both meanings of projection and the precise meaning should be clear from the context. As the two meanings are closely related and both standard, we avoid inventing a new name.

The following notion is also useful in later analysis:

**Definition 2.8.** Given two finite dimensional Hilbert spaces $V$ and $W$, a linear map $U : V \to W$ is called *isometric* if

$$\forall\, v_1, v_2 \in V\,, \quad \langle v_1 | v_2 \rangle = \langle U v_1 | U v_2 \rangle\,, \tag{22}$$

which is equivalent to $U^\dagger U = \mathrm{id}_V$. On the other hand, $U : V \to W$ is called *partially isometric* if any one of the following equivalent conditions holds (denote by $\ker U^\perp$ the orthogonal complement of $\ker U$ in $V$ and by $\mathrm{Im}\, U$ the image of $U$):

1. The restriction $U : \ker U^\perp \to W$ is isometric;

2. The restriction $U^\dagger : \mathrm{Im}\, U \to V$ is isometric;

3. The restriction $U : \ker U^\perp \to \mathrm{Im}\, U$ is unitary;

4. The restriction $U^\dagger : \mathrm{Im}\, U \to \ker U^\perp$ is unitary;

5. $U^\dagger U$ is a projection, $(U^\dagger U)^2 = U^\dagger U$;

6. $U U^\dagger$ is a projection, $(U U^\dagger)^2 = U U^\dagger$.

**Remark 2.9.** For technical simplicity, in this paper we mainly work with the example $\mathrm{Rep}\, G$ whose objects and morphisms have underlying vector spaces and linear maps that can be calculated concretely. We would like to emphasize that the graphical calculus techniques here and below remain valid even if we consider a more general unitary fusion category (UFC). For a general fusion category, we continue to interpret objects as symmetry charges and morphisms as symmetric operators; however, we lose access to underlying vectors and linear maps, unless we are willing to deal with (weak) Hopf algebras and their (co-)modules.

**Convention 1.** *Throughout this paper, it is understood that $\mathcal{C} = \mathrm{Rep}\, G$, but when we write $\mathcal{C}$, we are making statements applicable to general unitary fusion categories, and when we write $\mathrm{Rep}\, G$ we are making statements specific to $\mathrm{Rep}\, G$. Note that we also need to assume that $G$ is finite for $\mathrm{Rep}\, G$ to be a unitary fusion category.*

**Remark 2.10.** $\mathcal{C}$ is local [4] or anomaly-free if there exists a fiber functor $\mathcal{C} \to \mathbf{Vec}$, in which case one can reconstruct $\mathcal{C}$ as the representation category of a Hopf algebra, and consider such Hopf algebra as the global symmetry of the system. The fiber functor $\mathcal{C} \to \mathbf{Vec}$ allows us to work still with vector spaces and linear maps. When there does not exist a fiber functor, the fusion category is a representation category of a weak Hopf algebra and describes an anomalous symmetry. It is anomalous in the sense that the local tensor product structure of the lattice system is no longer the usual one. Given two representations over a weak Hopf algebra, the usual vector space tensor product of them is no longer a representation. Instead, one needs to take the relative tensor product over a subalgebra of the weak Hopf algebra.

**Definition 2.11.** Let $(\mathcal{A}, \otimes, \alpha)$ be a monoidal category, its *Drinfeld center* $Z_1(\mathcal{A})$ is the braided monoidal category defined as follows:

1. An object of $Z_1(\mathcal{A})$ is a pair $(X, \beta_{X,-})$ where $X \in \mathcal{A}$ is an object in $\mathcal{A}$ and $\beta_{X,V} : X \otimes V \to V \otimes X$ is a collection of isomorphisms for each $V \in \mathcal{A}$ such that

   (a) $\beta_{X,-}$ is natural: for any $f : V \to W$,

$$(f \otimes \mathrm{id}_X)\beta_{X,V} = \beta_{X,W}(\mathrm{id}_X \otimes f)\,. \tag{23}$$

(b) $\beta_{X,-}$ satisfies the hexagon equation

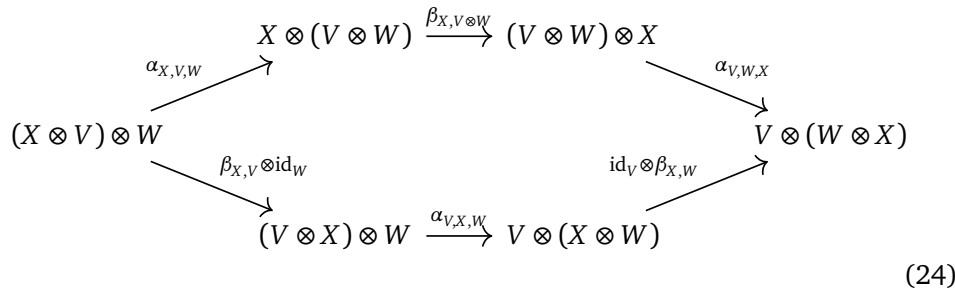

$$(24)$$

2. A morphism $g$ from $(X, \beta_{X,-})$ to $(Y, \beta_{Y,-})$[6] is a morphism $g : X \to Y$ in $\mathcal{A}$ satisfying $\beta_{Y,V}(g \otimes \mathrm{id}_V) = (\mathrm{id}_V \otimes g)\beta_{X,V}$ for any $V \in \mathcal{A}$.

3. The tensor product of $(X, \beta_{X,-})$ and $(Y, \beta_{Y,-})$ is given by $(X \otimes Y, \beta_{X \otimes Y,-})$ where

$$\beta_{X \otimes Y,V} = \alpha_{V,X,Y}(\beta_{X,V} \otimes \mathrm{id}_Y)\alpha_{X,V,Y}^{-1}(\mathrm{id}_X \otimes \beta_{Y,V})\alpha_{X,Y,V} \,. \tag{25}$$

4. The braiding is $c_{(X,\beta_{X,-}),(Y,\beta_{Y,-})} = \beta_{X,Y}$.

**Example 2.12 (Permutation group $S_3$).** $S_3$ is the simplest non-Abelian group with 6 elements:

$$S_3 = \left\{ 1, a, ab, ab^2, b, b^2 \mid a^2 = 1, b^3 = 1, aba = b^2 \right\}, \tag{26}$$

where $a, b$ are called two generators of $S_3$. $S_3$ has three irreducible representations $\lambda_0, \lambda_1, \Lambda$ listed below:

|  | $1$ | $a$ | $ab = b^2 a$ | $ab^2 = ba$ | $b$ | $b^2$ |
|---|---|---|---|---|---|---|
| $\lambda_0$ | $1$ | $1$ | $1$ | $1$ | $1$ | $1$ |
| $\lambda_1$ | $1$ | $-1$ | $-1$ | $-1$ | $1$ | $1$ |
| $\Lambda$ | $\begin{pmatrix} 1 & 0 \\ 0 & 1 \end{pmatrix}$ | $\begin{pmatrix} 0 & 1 \\ 1 & 0 \end{pmatrix}$ | $\begin{pmatrix} 0 & \omega^* \\ \omega & 0 \end{pmatrix}$ | $\begin{pmatrix} 0 & \omega \\ \omega^* & 0 \end{pmatrix}$ | $\begin{pmatrix} \omega & 0 \\ 0 & \omega^* \end{pmatrix}$ | $\begin{pmatrix} \omega^* & 0 \\ 0 & \omega \end{pmatrix}$ |

where $\omega = e^{2\pi i/3}$. Let $\mathrm{Irr}(\mathcal{C})$ denote for the set of isomorphism classes of simple objects in semisimple category $\mathcal{C}$. We have $\mathrm{Irr}(\mathrm{Rep}\, S_3) = \{\lambda_0, \lambda_1, \Lambda\}$. The dual representation $\Lambda^*$ of $\Lambda$ is listed as

|  | $1$ | $a$ | $ab = b^2 a$ | $ab^2 = ba$ | $b$ | $b^2$ |
|---|---|---|---|---|---|---|
| $\Lambda^*$ | $\begin{pmatrix} 1 & 0 \\ 0 & 1 \end{pmatrix}$ | $\begin{pmatrix} 0 & 1 \\ 1 & 0 \end{pmatrix}$ | $\begin{pmatrix} 0 & \omega \\ \omega^* & 0 \end{pmatrix}$ | $\begin{pmatrix} 0 & \omega^* \\ \omega & 0 \end{pmatrix}$ | $\begin{pmatrix} \omega^* & 0 \\ 0 & \omega \end{pmatrix}$ | $\begin{pmatrix} \omega & 0 \\ 0 & \omega^* \end{pmatrix}$ |

Denote the basis of $\Lambda$ as $\{\mathbf{0}, \mathbf{1}\}$. Correspondingly $\Lambda^*$ has dual basis $\{\delta_\mathbf{0}, \delta_\mathbf{1}\}$. Then $\Lambda^*$ is isomorphic to $\Lambda$ through the isomorphism $\mathbf{0} \mapsto \delta_\mathbf{1}, \mathbf{1} \mapsto \delta_\mathbf{0}$.

We also list the data in $Z_1(\mathrm{Rep}\, S_3)$ (See the calculation of $Z_1(\mathrm{Rep}\, G)$ for general $G$ in Appendix A):

| Centralizer subgroups | $N_1 \cong S_3$ | $N_a = \{1, a\} \cong N_{ab} = \{1, ab\}$ $\cong N_{ab^2} = \{1, ab^2\} \cong \mathbb{Z}_2$ | $N_b = N_{b^2}$ $= \{1, b, b^2\} \cong \mathbb{Z}_3$ |
|---|---|---|---|
| Conjugacy classes | $C_1 \cong G/N_1 \cong \{1\}$ | $C_a \cong C_{ab} \cong C_{ab^2}$ $\cong G/N_a \cong \{a, ab, ab^2\}$ | $C_b \cong C_{b^2} \cong G/N_b$ $\cong \{b, b^2\}$ |
| Representation of centralizer subgroup | $\mathrm{Rep}\, S_3$ | $\mathrm{Rep}\, \mathbb{Z}_2$ $\mathrm{Irr}(\mathrm{Rep}\, \mathbb{Z}_2) = \{+, -\}$ | $\mathrm{Rep}\, \mathbb{Z}_3$ $\mathrm{Irr}(\mathrm{Rep}\, \mathbb{Z}_3) = \{1, \omega, \omega^2\}$ |

---

[6]We follow the usual convention and abuse the same notation $\beta_{X,-}$ and $\beta_{Y,-}$ for different pairs $(X, \beta_{X,-})$ and $(Y, \beta_{Y,-})$. The reader is reminded that the pairs should be understood as a whole; the half-braiding $\beta_{X,-}$ depends on the pair $(X, \beta)$ instead of only the object $X$. Indeed, the same object $X$ may be equipped with different half-braidings to form different objects in the Drinfeld center.

Therefore, there are 8 simple objects in $Z_1(\text{Rep}\,S_3)$ labelled as:

$$\text{Irr}(Z_1(\text{Rep}\,S_3)) = \left\{ (C_1, \lambda_0), (C_1, \lambda_1), (C_1, \Lambda), (C_a, +), (C_a, -), (C_b, 1), (C_b, \omega), (C_b, \omega^2) \right\} . \quad (27)$$

**Example 2.13** (Quaternion group $\boldsymbol{Q_8}$). $Q_8$ is a non-Abelian group with 8 elements:

$$Q_8 = \left\{ \pm 1, \pm i, \pm j, \pm k \,|\, i^2 = j^2 = k^2 = ijk = -1, \, ikj = 1 \right\} . \quad (28)$$

$Q_8$ has five irreducible representations $\gamma_0, \gamma_1, \gamma_2, \gamma_3, \Gamma$ listed below:

|            | $\pm 1$ | $\pm i$ | $\pm j$ | $\pm k$ |
|------------|---------|---------|---------|---------|
| $\gamma_0$ | 1       | 1       | 1       | 1       |
| $\gamma_1$ | 1       | 1       | $-1$    | $-1$    |
| $\gamma_2$ | 1       | $-1$    | 1       | $-1$    |
| $\gamma_3$ | 1       | $-1$    | $-1$    | 1       |
| $\Gamma$   | $\pm I$ | $\mp i\sigma_x$ | $\mp i\sigma_y$ | $\mp i\sigma_z$ |

where $\sigma_x, \sigma_y, \sigma_z$ are Pauli matrices. The dual representation $\Gamma^*$ of $\Gamma$ is listed as

|            | $\pm 1$ | $\pm i$ | $\pm j$ | $\pm k$ |
|------------|---------|---------|---------|---------|
| $\Gamma^*$ | $\pm I$ | $\pm i\sigma_x$ | $\mp i\sigma_y$ | $\pm i\sigma_z$ |

Denote the basis of $\Gamma$ as $\{|0\rangle, |1\rangle\}$. Correspondingly $\Gamma^*$ has dual basis $\{\delta_{|0\rangle}, \delta_{|1\rangle}\}$. Then $\Gamma^*$ is isomorphic to $\Gamma$ through the isomorphism $|0\rangle \mapsto \delta_{|1\rangle}, |1\rangle \mapsto -\delta_{|0\rangle}$.

**Example 2.14** (Special unitary group $\boldsymbol{SU(2)}$). $SU(2)$ is a non-Abelian Lie group with infinite elements:

$$SU(2) = \left\{ \begin{pmatrix} \alpha & -\overline{\beta} \\ \beta & \overline{\alpha} \end{pmatrix} \middle| \alpha, \beta \in \mathbb{C}, |\alpha|^2 + |\beta|^2 = 1 \right\} , \quad (29)$$

where $\overline{\alpha}$ denotes the complex conjugate of $\alpha$. $SU(2)$ has infinite numbers of irreps labelled by $l$, which are non-negative integers and half-integers. The dimension of irrep labelled by $l$ is $2l+1$. We list generators for irreps $l = 0, 1/2, 1$ below:

|                   | $J_x$ | $J_y$ | $J_z$ |
|-------------------|-------|-------|-------|
| $l = 0$           | 0     | 0     | 0     |
| $l = \frac{1}{2}$ | $\frac{1}{2}\sigma_x$ | $\frac{1}{2}\sigma_y$ | $\frac{1}{2}\sigma_z$ |
| $l = 1$           | $\begin{pmatrix} 0 & 0 & 0 \\ 0 & 0 & -1 \\ 0 & 1 & 0 \end{pmatrix}$ | $\begin{pmatrix} 0 & 0 & 1 \\ 0 & 0 & 0 \\ -1 & 0 & 0 \end{pmatrix}$ | $\begin{pmatrix} 0 & -1 & 0 \\ 1 & 0 & 0 \\ 0 & 0 & 0 \end{pmatrix}$ |

For each $l$, group elements in $SU(2)$ represented by $l$ are

$$\left\{ e^{i\theta(n_x J_x + n_y J_y + n_z J_z)} \,|\, \theta, n_x, n_y, n_z \in \mathbb{R}, n_x^2 + n_y^2 + n_z^2 = 1 \right\} .$$

The dual representation $l = \frac{1}{2}^*$ of $l = \frac{1}{2}$ is listed as

|                     | $J_x$ | $J_y$ | $J_z$ |
|---------------------|-------|-------|-------|
| $l = \frac{1}{2}^*$ | $-\frac{1}{2}\sigma_x$ | $\frac{1}{2}\sigma_y$ | $-\frac{1}{2}\sigma_z$ |

Denote the basis of $l = \frac{1}{2}$ as $\left\{ \left|\frac{1}{2}, \frac{1}{2}\right\rangle, \left|\frac{1}{2}, -\frac{1}{2}\right\rangle \right\}$. Correspondingly $l = \frac{1}{2}^*$ has dual basis $\left\{ \delta_{|\frac{1}{2}, \frac{1}{2}\rangle}, \delta_{|\frac{1}{2}, -\frac{1}{2}\rangle} \right\}$. Then $\frac{1}{2}^*$ is isomorphic to $\frac{1}{2}$ through the isomorphism $\left|\frac{1}{2}, \frac{1}{2}\right\rangle \mapsto \delta_{|\frac{1}{2}, -\frac{1}{2}\rangle}, \left|\frac{1}{2}, -\frac{1}{2}\right\rangle \mapsto -\delta_{|\frac{1}{2}, \frac{1}{2}\rangle}$.

**Definition 2.15.** Fix a set $\mathbf{L}$ (whose elements are viewed as lattice sites) and a group $G$. *A quantum system with onsite symmetry $G$, on $\mathbf{L}$, or just a system for short, consists of the following data*

1. For each subset $\mathbf{K} \subset \mathbf{L}$, there is a Hilbert space $\mathcal{H}_{\mathbf{K}}$ which carries a group representation $(\mathcal{H}_{\mathbf{K}}, \rho^{\mathbf{K}})$;

2. A Hermitian operator (the total Hamiltonian) $H$ on $\mathcal{H}_{\mathbf{L}}$,

such that

1. For any two disjoint subsets $\mathbf{K}_1$ and $\mathbf{K}_2$, the representation associated to their disjoint union is the tensor product of those associated to $\mathbf{K}_1$ and $\mathbf{K}_2$

$$(\mathcal{H}_{\mathbf{K}_1 \coprod \mathbf{K}_2}, \rho^{\mathbf{K}_1 \coprod \mathbf{K}_2}) = (\mathcal{H}_{\mathbf{K}_1}, \rho^{\mathbf{K}_1}) \otimes (\mathcal{H}_{\mathbf{K}_2}, \rho^{\mathbf{K}_2}). \tag{30}$$

2. The total Hamiltonian has the form

$$H = \sum_{\mathbf{K} \subset \mathbf{L}} H_{\mathbf{K}}, \tag{31}$$

where $H_{\mathbf{K}} = \tilde{H}_{\mathbf{K}} \otimes \mathrm{id}_{\mathcal{H}_{\mathbf{L} \backslash \mathbf{K}}}$ and $\tilde{H}_{\mathbf{K}}$ is a symmetric operator on $\mathcal{H}_{\mathbf{K}}$, i.e., $\rho_g^{\mathbf{K}} \tilde{H}_{\mathbf{K}} (\rho_g^{\mathbf{K}})^{-1} = \tilde{H}_{\mathbf{K}}$, $\forall g \in G$.

We denote such a quantum system by $(\mathbf{L}, \mathcal{H}_{\mathbf{K}}, H)$.

**Remark 2.16.** The empty set $\emptyset \subset \mathbf{L}$ is necessarily associated with the trivial group representation. For the singleton subset $\{i\} \subset \mathbf{L}$ ($i$ is just a lattice site), we will simply write the associated representation as $(\mathcal{H}_i, \rho^i)$. It is clear that

$$\mathcal{H}_{\mathbf{K}} = \underset{i \in \mathbf{K}}{\otimes} \mathcal{H}_i, \tag{32}$$

$$\rho_g^{\mathbf{K}} = \underset{i \in \mathbf{K}}{\otimes} \rho_g^i. \tag{33}$$

An operator on $\mathcal{H}_{\mathbf{L}}$ of the form $O_{\mathbf{K}} = \tilde{O}_{\mathbf{K}} \otimes \mathrm{id}_{\mathcal{H}_{\mathbf{L} \backslash \mathbf{K}}}$ is called *supported on* $\mathbf{K}$. When no confusion arises, we will abuse notation and do not distinguish $O_{\mathbf{K}}$ from $\tilde{O}_{\mathbf{K}}$.

# 3 Symmetric operators as graphs

In this section we set up the formulation that any symmetric operator in a system with onsite symmetry $G$ can be represented by graphical calculus in $\mathrm{Rep}\,G$. In the language of tensor network, any symmetric operator can be represented by a $G$-invariant tensor network.

Each local Hilbert space can be decomposed to irreducible representations.

**Definition 3.1.** A *charge decomposition* consists of the following choices:

1. A representative $X_a$ for each isomorphic class $a$ of irreps, $a \in \mathrm{Irr}(\mathrm{Rep}\,G)$.

2. For each local Hilbert space or more generally each representation $V$, the projection maps to and the embedding maps from the representative irreps: $p_V^{a;n} : V \to X_a, q_{a;n}^V : X_a \to V$ where $n$ goes from 1 to the multiplicity of $X_a$ in $V$.

3. In particular, for each tensor product of two representative irreps, the projection maps to and the embedding maps from the representative irreps:

$$p_{ab}^{c;n} : X_a \otimes X_b \to X_c, q_{c;n}^{ab} : X_c \to X_a \otimes X_b. \tag{34}$$

Tree graphs decorated by $X_a$ and $p, q$'s in a charge decomposition serve as bases of intertwiner spaces. We may draw any graph in $\mathrm{Rep}\, G$ and interpret it as an intertwiner between several representations. More specifically, let $V_1, \ldots, V_n$ and $W_1, \ldots, W_m$ be representations, an intertwiner $f \in \mathrm{Hom}(V_1 \otimes \ldots \otimes V_n, W_1 \otimes \ldots \otimes W_m)$ can be represented by the graph:

$$f = \qquad\qquad\qquad\qquad \tag{35}$$

With a chosen charge decomposition, the intertwiner above can be expanded in terms of "basis" graphs. As an illustrative example, consider the intertwiner space $\mathrm{Hom}(V_1 \otimes V_2, W_1 \otimes W_2)$ with four external legs $V_1, V_2, W_1, W_2$. One first decompose all external legs to irreps, and then fuse the irreps one by one in a chosen order. One choice of basis graphs is

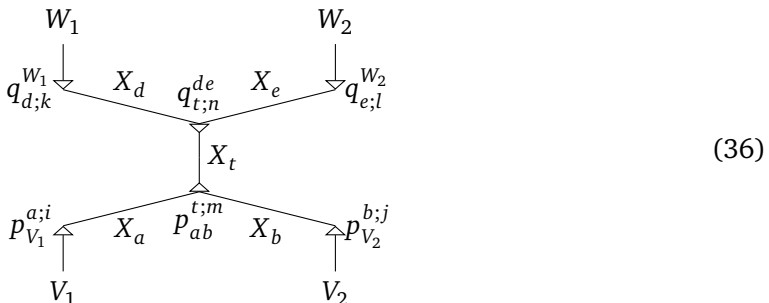

$$\tag{36}$$

where $a, b, t, d, e, i, j, k, l, m, n$ runs over all possible values. Another choice is

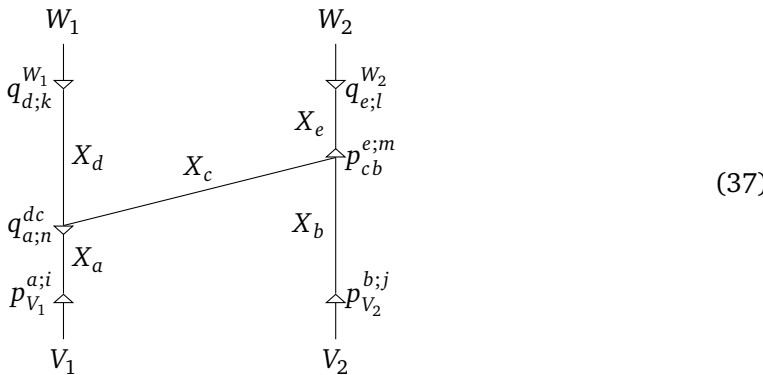

$$\tag{37}$$

where $a, b, c, d, e, i, j, k, l, m, n$ runs over all possible values. The only remaining third choice is left as an exercise for the reader.

In general, a basis of the intertwiner space $\mathrm{Hom}(V_1 \otimes \ldots \otimes V_n, W_1 \otimes \ldots \otimes W_m)$ can be obtained by decorating a tree graph, which is bivalent between external and internal legs and trivalent between internal legs:

- The external legs are fixed and decorated by $V_1, \ldots, V_n, W_1, \ldots, W_m$;

- The internal legs are decorated by representative irreps;

- The vertices are decorated by projection and embedding maps from a chosen charge decomposition.

Different orders to fuse internal legs lead to different tree graphs and thus different sets of basis intertwiners. Basis intertwiners on different tree graphs are related to each other by the F-moves.

After fixing basis graphs, every symmetric operator has a unique linear expansion and concrete calculation is possible. However, for the purpose to read out the total symmetry charge being transported by symmetric operators, we are going to use a slightly more effective method to rewrite the symmetric operators.

**Definition 3.2.** Consider an intertwiner $f \in \mathrm{Hom}(V, W)$. An *image decomposition* $(\bar{l}, \bar{r})$ of $f$ is a factorization of $f$ through its image

$$f = V \xrightarrow{\bar{l}} \mathrm{Im} f \xrightarrow{\bar{r}} W \,, \tag{38}$$

where $\mathrm{Im} f$ is the image of $f$ (Definition B.8). Graphically,

$$\boxed{f} \quad = \quad \mathrm{Im} f \,. \tag{39}$$

In more concrete words, we can think the image decomposition as a singular value decomposition performed on symmetric tensors, which we call *symmetric singular value decomposition* (SSVD). Explicitly, there are two steps in an SSVD of $f$:

1. Direct sum decomposition of $V$ and $W$. Suppose the direct sum decomposition of $V, W$ is given by the projection maps $p_V^{a;n}, p_W^{a;m}, n = 1, \ldots, n_a, m = 1, \ldots, m_a$, then $f$ can be represented by a block diagonal matrix, whose blocks correspond to irreps:

$$M_{mn}^a := p_W^{a;m} f (p_V^{a;n})^\dagger, \quad f = \sum_{a,mn} (p_W^{a;m})^\dagger M_{mn}^a p_V^{a;n} \,. \tag{40}$$

2. The usual SVD performed on the $m_a \times n_a$ matrices $M_{mn}^a$ for all representative irreps $X_a$, leads to a SSVD of $f$:

$$\bigoplus_a M^a = \bigoplus_a (w^a)^\dagger \Sigma^a v^a \,, \tag{41}$$

where $v^a$ and $w^a$ are $n_a \times n_a$ and $m_a \times m_a$ unitary matrices, and $\Sigma^a$ is a $m_a \times n_a$ rectangular diagonal matrix with non-negative real entries. The number of non-zero singular values of $M^a$ tells us how many copies of $X_a$ are in $\mathrm{Im} f$. Graphically, for each isomorphism class $a$ of irreps,

$$\boxed{M_{mn}^a} = \sum_k \boxed{\Sigma_{kk}^a} \,. \tag{42}$$

And therefore,

$$
f = \sum_{a,mn} \quad \boxed{M^a_{mn}} \quad = \sum_{a,k} \quad \boxed{\Sigma^a_{kk}} \quad ,
\tag{43}
$$

where $v_V^{a;k} := \sum_n v_{kn}^a p_V^{a;n}$ and $w_W^{a;k} := \sum_m w_{km}^a p_W^{a;m}$ are another two sets of projections of the direct sum decomposition of $V, W$.

Here it is easy to see that the ambiguity of $\bar{l}$ and $\bar{r}$ arises from choices of block diagonal unitary matrices in the SSVD and how one separate the non-zero singular values of $M^a$.

**Remark 3.3.** In the factorization $f = \bar{r}\bar{l}$, $\bar{r}$ is a monomorphism. And in any abelian category (in this paper all categories used are moreover semisimple), $\bar{l}$ is an epimorphism, proof of which is omitted.

**Remark 3.4.** The pair $(\bar{l}, \bar{r})$ is clearly not unique. Such ambiguity is related to the sectors of quantum currents and morphisms between quantum currents. We will come to this point later.

**Remark 3.5.** We introduce a trial-and-error method to compute the image, which is more straightforward (if succeed). Given two representations $(V, \rho), (W, \tau)$, note that we can endow the operator space $\mathrm{Hom}_{\mathbf{Vec}}(V, W)$ with a natural group action by post-composing $\tau_g$. For an intertwiner $f : V \to W$, we can then consider the cyclic sub-representation of $\mathrm{Hom}_{\mathbf{Vec}}(V, W)$ generated by $f$, $C(G)f := \langle \tau_g f, g \in G \rangle$. Pick any $v \in V$, there is an intertwiner:

$$
\begin{aligned}
\xi_v : C(G)f &\to W, \\
O &\mapsto O(v).
\end{aligned}
\tag{44}
$$

$O \in C(G)f$ has the form $O = \sum_{h \in G} c_h \tau_h f = \sum_{h \in G} c_h f \rho_h$, with which it is easy to check that $\xi_v$ is symmetric. If $C(G)f$ happens to be isomorphic to $\mathrm{Im} f$ for some $v$, the computation is completed. For this method to work, the necessary and sufficient conditions are

1. $\mathrm{Im} f \cong C(G)f$ is a cyclic sub-representation of $W$. $v$ should be chosen as a preimage of a cyclic vector in $\mathrm{Im} f$. In this case $\xi_v$ automatically maps onto $\mathrm{Im} f$.

2. $\xi_v$ must also be injective, which means that $\ker \xi_v = 0$, i.e., $\xi_v(O) = O(v) = 0$ implies that $O$ is the zero operator $O(w) = 0$, $\forall w \in V$.

There are simple cases that these conditions are satisfied, for example, $V$ is the regular representation $C(G)$ of Abelian group $G$, where one can take $v = e$ the identity element of $G$. However, in practice, the second condition is difficult to check, and also we can not know in advance whether $\mathrm{Im} f$ is a cyclic representation or not. Such method can only be used with trial-and-error.

**Example 3.6.** Let $G = S_3$ (Example 2.12). Consider the direct sum decomposition $\Lambda \otimes \Lambda \cong \Lambda \oplus \lambda_0 \oplus \lambda_1$. We take the basis of $\Lambda$ to be $\{\mathbf{0}, \mathbf{1}\}$, and the tensor basis of $\Lambda \otimes \Lambda$ to be $\{\mathbf{00}, \mathbf{01}, \mathbf{10}, \mathbf{11}\}$ (similarly for later examples involving 2-dimensional representations). Then the basis change between $\Lambda \otimes \Lambda$ and $\Lambda \oplus \lambda_0 \oplus \lambda_1$ is

$$
p_{\Lambda\otimes\Lambda}^{\Lambda} = \begin{pmatrix} 1 & 0 & 0 & 0 \\ 0 & 0 & 0 & 1 \end{pmatrix}, \quad p_{\Lambda\otimes\Lambda}^{\lambda_0} = \begin{pmatrix} 0 & \frac{1}{\sqrt{2}} & \frac{1}{\sqrt{2}} & 0 \end{pmatrix}, \quad p_{\Lambda\otimes\Lambda}^{\lambda_1} = \begin{pmatrix} 0 & \frac{1}{\sqrt{2}} & -\frac{1}{\sqrt{2}} & 0 \end{pmatrix}.
\tag{45}
$$

Consider an intertwiner $f \in \mathrm{Hom}(\Lambda \otimes \Lambda, \Lambda \otimes \Lambda)$ taking the following form:

$$
f = \begin{pmatrix} 1 & 0 & 0 & 0 \\ 0 & 0 & 0 & 0 \\ 0 & 0 & 0 & 0 \\ 0 & 0 & 0 & 1 \end{pmatrix} = \underbrace{\begin{pmatrix} 1 & 0 & 0 & 0 \\ 0 & 0 & \frac{1}{\sqrt{2}} & \frac{1}{\sqrt{2}} \\ 0 & 0 & \frac{1}{\sqrt{2}} & -\frac{1}{\sqrt{2}} \\ 0 & 1 & 0 & 0 \end{pmatrix}}_{\substack{\text{Changing basis from} \\ \Lambda \oplus \lambda_0 \oplus \lambda_1 \text{ to } \Lambda \otimes \Lambda}} \overset{M^\Lambda \oplus M^{\lambda_0} \oplus M^{\lambda_1}}{\begin{pmatrix} 1 & 0 & 0 & 0 \\ 0 & 1 & 0 & 0 \\ 0 & 0 & 0 & 0 \\ 0 & 0 & 0 & 0 \end{pmatrix}} \underbrace{\begin{pmatrix} 1 & 0 & 0 & 0 \\ 0 & 0 & 0 & 1 \\ 0 & \frac{1}{\sqrt{2}} & \frac{1}{\sqrt{2}} & 0 \\ 0 & \frac{1}{\sqrt{2}} & -\frac{1}{\sqrt{2}} & 0 \end{pmatrix}}_{\substack{\text{Changing basis from} \\ \Lambda \otimes \Lambda \text{ to } \Lambda \oplus \lambda_0 \oplus \lambda_1}}
$$

$$
= \begin{pmatrix} p^\Lambda_{\Lambda \otimes \Lambda} \\ p^{\lambda_0}_{\Lambda \otimes \Lambda} \\ p^{\lambda_1}_{\Lambda \otimes \Lambda} \\ p^{\lambda_1}_{\Lambda \otimes \Lambda} \end{pmatrix}^\dagger \begin{pmatrix} 1 & 0 & 0 & 0 \\ 0 & 1 & 0 & 0 \\ 0 & 0 & 0 & 0 \\ 0 & 0 & 0 & 0 \end{pmatrix} \begin{pmatrix} p^\Lambda_{\Lambda \otimes \Lambda} \\ p^{\lambda_0}_{\Lambda \otimes \Lambda} \\ p^{\lambda_1}_{\Lambda \otimes \Lambda} \\ p^{\lambda_1}_{\Lambda \otimes \Lambda} \end{pmatrix} = (p^\Lambda_{\Lambda \otimes \Lambda})^\dagger p^\Lambda_{\Lambda \otimes \Lambda} = M^\Lambda (p^\Lambda_{\Lambda \otimes \Lambda})^\dagger p^\Lambda_{\Lambda \otimes \Lambda}, \tag{46}
$$

where the similarity transformation can be thought as changing to the basis with fixed symmetry charge, i.e., changing $f \in \mathrm{End}(\Lambda \otimes \Lambda)$ to $\bigoplus_a M^a \in \mathrm{End}(\Lambda \oplus \lambda_0 \oplus \lambda_1)$. Since $M^\Lambda = 1$, there is only one non-zero singular value in $M^\Lambda$,[7] we see $\mathrm{Im}\, f \cong \Lambda$. Graphically,

$$
f = \quad \begin{array}{c} \Lambda \quad\quad \Lambda \\ \boxed{\bar{r}} \\ \Lambda \\ \boxed{\bar{l}} \\ \Lambda \quad\quad \Lambda \end{array} \quad , \tag{47}
$$

where by requiring $\bar{l}$ and $\bar{r}$ to be intertwiners, the image decomposition is solved as

$$
\bar{l} = \begin{pmatrix} 0 & 0 & 0 & c_1 \\ c_1 & 0 & 0 & 0 \end{pmatrix}, \qquad \bar{r} = \begin{pmatrix} 0 & c_2 \\ 0 & 0 \\ 0 & 0 \\ c_2 & 0 \end{pmatrix}, \tag{48}
$$

where $c_1, c_2$ are non-zero complex numbers such that $c_1 c_2 = 1$. A different choice of $(\bar{l}, \bar{r})$ correspond to changing $c_1$ and $c_2$ to $c_1'$ and $c_2'$ such that $c_1' c_2' = 1$. Here since $\dim \mathrm{Hom}(\Lambda \otimes \Lambda, \Lambda) = 1$, choices of $(\bar{l}, \bar{r})$ only differ by a scalar.

Similarly, a general intertwiner $f \in \mathrm{Hom}(\Lambda \otimes \Lambda, \Lambda \otimes \Lambda)$ reads

$$
f = c_1 (p^\Lambda_{\Lambda \otimes \Lambda})^\dagger p^\Lambda_{\Lambda \otimes \Lambda} + c_2 (p^{\lambda_0}_{\Lambda \otimes \Lambda})^\dagger p^{\lambda_0}_{\Lambda \otimes \Lambda} + c_3 (p^{\lambda_1}_{\Lambda \otimes \Lambda})^\dagger p^{\lambda_1}_{\Lambda \otimes \Lambda}
$$

$$
= \begin{pmatrix} p^\Lambda_{\Lambda \otimes \Lambda} \\ p^{\lambda_0}_{\Lambda \otimes \Lambda} \\ p^{\lambda_1}_{\Lambda \otimes \Lambda} \\ p^{\lambda_1}_{\Lambda \otimes \Lambda} \end{pmatrix}^\dagger \begin{pmatrix} c_1 & 0 & 0 & 0 \\ 0 & c_1 & 0 & 0 \\ 0 & 0 & c_2 & 0 \\ 0 & 0 & 0 & c_3 \end{pmatrix} \begin{pmatrix} p^\Lambda_{\Lambda \otimes \Lambda} \\ p^{\lambda_0}_{\Lambda \otimes \Lambda} \\ p^{\lambda_1}_{\Lambda \otimes \Lambda} \\ p^{\lambda_1}_{\Lambda \otimes \Lambda} \end{pmatrix}
$$

$$
= \begin{pmatrix} 1 & 0 & 0 & 0 \\ 0 & 0 & \frac{1}{\sqrt{2}} & \frac{1}{\sqrt{2}} \\ 0 & 0 & \frac{1}{\sqrt{2}} & -\frac{1}{\sqrt{2}} \\ 0 & 1 & 0 & 0 \end{pmatrix} \begin{pmatrix} c_1 & 0 & 0 & 0 \\ 0 & c_1 & 0 & 0 \\ 0 & 0 & c_2 & 0 \\ 0 & 0 & 0 & c_3 \end{pmatrix} \begin{pmatrix} 1 & 0 & 0 & 0 \\ 0 & 0 & 0 & 1 \\ 0 & \frac{1}{\sqrt{2}} & \frac{1}{\sqrt{2}} & 0 \\ 0 & \frac{1}{\sqrt{2}} & -\frac{1}{\sqrt{2}} & 0 \end{pmatrix}.
$$

$$
= \begin{pmatrix} c_1 & 0 & 0 & 0 \\ 0 & \frac{c_2+c_3}{2} & \frac{c_2-c_3}{2} & 0 \\ 0 & \frac{c_2-c_3}{2} & \frac{c_2+c_3}{2} & 0 \\ 0 & 0 & 0 & c_1 \end{pmatrix}. \tag{49}
$$

---

[7]Note that the matrix $M^\Lambda$ is not indexed by the basis of $\Lambda$. By Definition (40), $M^\Lambda$ is a $1 \times 1$ matrix.

**Example 3.7.** Let $G = Q_8$ (Example 2.13). Consider the direct sum decomposition $\Gamma^* \otimes \Gamma \cong \Gamma \otimes \Gamma^* \cong \gamma_0 \oplus \gamma_1 \oplus \gamma_2 \oplus \gamma_3$, and the basis change between them is[8]

$$
p^{\gamma_0}_{\Gamma^* \otimes \Gamma} = \begin{pmatrix} \frac{1}{\sqrt 2} & 0 & 0 & \frac{1}{\sqrt 2} \end{pmatrix}, \qquad p^{\gamma_1}_{\Gamma^* \otimes \Gamma} = \begin{pmatrix} 0 & \frac{1}{\sqrt 2} & \frac{1}{\sqrt 2} & 0 \end{pmatrix},
$$
$$
p^{\gamma_2}_{\Gamma^* \otimes \Gamma} = \begin{pmatrix} 0 & \frac{1}{\sqrt 2} & -\frac{1}{\sqrt 2} & 0 \end{pmatrix}, \qquad p^{\gamma_3}_{\Gamma^* \otimes \Gamma} = \begin{pmatrix} \frac{1}{\sqrt 2} & 0 & 0 & -\frac{1}{\sqrt 2} \end{pmatrix}, \tag{50}
$$
$$
p^{\gamma_0}_{\Gamma \otimes \Gamma^*} = \begin{pmatrix} \frac{1}{\sqrt 2} & 0 & 0 & \frac{1}{\sqrt 2} \end{pmatrix}, \qquad p^{\gamma_1}_{\Gamma \otimes \Gamma^*} = \begin{pmatrix} 0 & \frac{1}{\sqrt 2} & \frac{1}{\sqrt 2} & 0 \end{pmatrix},
$$
$$
p^{\gamma_2}_{\Gamma \otimes \Gamma^*} = \begin{pmatrix} 0 & \frac{1}{\sqrt 2} & -\frac{1}{\sqrt 2} & 0 \end{pmatrix}, \qquad p^{\gamma_3}_{\Gamma \otimes \Gamma^*} = \begin{pmatrix} \frac{1}{\sqrt 2} & 0 & 0 & -\frac{1}{\sqrt 2} \end{pmatrix}. \tag{51}
$$

Consider an intertwiner $f \in \mathrm{Hom}(\Gamma^* \otimes \Gamma, \Gamma \otimes \Gamma^*)$ taking the following form:

$$
f = \begin{pmatrix} 0 & 0 & 0 & 0 \\ 0 & 1 & 1 & 0 \\ 0 & 1 & 1 & 0 \\ 0 & 0 & 0 & 0 \end{pmatrix} = \begin{pmatrix} \frac{1}{\sqrt 2} & 0 & 0 & \frac{1}{\sqrt 2} \\ 0 & \frac{1}{\sqrt 2} & \frac{1}{\sqrt 2} & 0 \\ 0 & \frac{1}{\sqrt 2} & -\frac{1}{\sqrt 2} & 0 \\ \frac{1}{\sqrt 2} & 0 & 0 & -\frac{1}{\sqrt 2} \end{pmatrix} \begin{pmatrix} 0 & 0 & 0 & 0 \\ 0 & 2 & 0 & 0 \\ 0 & 0 & 0 & 0 \\ 0 & 0 & 0 & 0 \end{pmatrix} \begin{pmatrix} \frac{1}{\sqrt 2} & 0 & 0 & \frac{1}{\sqrt 2} \\ 0 & \frac{1}{\sqrt 2} & \frac{1}{\sqrt 2} & 0 \\ 0 & \frac{1}{\sqrt 2} & -\frac{1}{\sqrt 2} & 0 \\ \frac{1}{\sqrt 2} & 0 & 0 & -\frac{1}{\sqrt 2} \end{pmatrix}
$$
$$
= \begin{pmatrix} p^{\gamma_0}_{\Gamma \otimes \Gamma^*} \\ p^{\gamma_1}_{\Gamma \otimes \Gamma^*} \\ p^{\gamma_2}_{\Gamma \otimes \Gamma^*} \\ p^{\gamma_3}_{\Gamma \otimes \Gamma^*} \end{pmatrix}^{\dagger} \begin{pmatrix} 0 & 0 & 0 & 0 \\ 0 & 2 & 0 & 0 \\ 0 & 0 & 0 & 0 \\ 0 & 0 & 0 & 0 \end{pmatrix} \begin{pmatrix} p^{\gamma_0}_{\Gamma \otimes \Gamma^*} \\ p^{\gamma_1}_{\Gamma \otimes \Gamma^*} \\ p^{\gamma_2}_{\Gamma \otimes \Gamma^*} \\ p^{\gamma_3}_{\Gamma \otimes \Gamma^*} \end{pmatrix} = 2 (p^{\gamma_1}_{\Gamma \otimes \Gamma^*})^{\dagger} p^{\gamma_1}_{\Gamma^* \otimes \Gamma}, \tag{52}
$$

where the similarity transformation can be thought as changing to the basis with fixed symmetry charge in $\mathrm{Hom}(\gamma_0 \oplus \gamma_1 \oplus \gamma_2 \oplus \gamma_3, \gamma_0 \oplus \gamma_1 \oplus \gamma_2 \oplus \gamma_3)$. Since there is only one non-zero singular value in $M^{\gamma_1}$, we see $\mathrm{Im} f \cong \gamma_1$. Graphically,

$$
f = \quad \begin{array}{c} \Gamma \qquad\qquad \Gamma^* \\[2pt] \boxed{\bar r} \\ \gamma_1 \\ \boxed{\bar l} \\[2pt] \Gamma^* \qquad\qquad \Gamma \end{array} \quad , \tag{53}
$$

where the image decomposition is solved as

$$
\bar l = c_1 \begin{pmatrix} 0 & 1 & 1 & 0 \end{pmatrix}, \qquad \bar r = c_2 \begin{pmatrix} 0 \\ 1 \\ 1 \\ 0 \end{pmatrix}, \tag{54}
$$

where $c_1, c_2$ are non-zero complex numbers such that $c_1 c_2 = 1$.

# 4 Quantum current

In this section we motivate the definition of quantum current. We would like to define a quantum current as *a collection of symmetric operators that can transport symmetry charge all over the space*. This physical idea will be put into precise mathematical definition later. For concreteness we will first focus on lattice system in one spatial dimension with onsite symmetry.

---

[8]Denote the basis of $\Gamma$ to be $\{|0\rangle, |1\rangle\}$ and the dual basis of $\Gamma^*$ to be $\{\delta_{|0\rangle}, \delta_{|1\rangle}\}$. If we choose the isomorphism between $\Gamma^* \otimes \Gamma$ and $\Gamma \otimes \Gamma^*$ to be $\delta_{|0\rangle}|0\rangle \mapsto |0\rangle \delta_{|0\rangle}$, $\delta_{|0\rangle}|1\rangle \mapsto |0\rangle \delta_{|1\rangle}$, $\delta_{|1\rangle}|0\rangle \mapsto |1\rangle \delta_{|0\rangle}$, $\delta_{|1\rangle}|1\rangle \mapsto |1\rangle \delta_{|1\rangle}$, the matrix transformations changing their basis to $\gamma_0 \oplus \gamma_1 \oplus \gamma_2 \oplus \gamma_3$ happen to be the same.

We will show that a quantum current carries two important quantities: one is the symmetry charge being transported, and the other is the *half-braiding* that determines how the current extends over the space without leaving things behind along its path. Together, a quantum current is identified with an object in the Drinfeld center $Z_1(\mathcal{C})$.

## 4.1 Every symmetric operator carries a symmetry charge

To begin with, let's consider a bipartite system,

$$\mathcal{H} = \mathcal{H}_{\mathbf{A}} \otimes \mathcal{H}_{\mathbf{B}}, \tag{55}$$

with symmetry actions $\rho_g^i \in GL(\mathcal{H}_i), i = \mathbf{A}, \mathbf{B}$ and $\rho_g = \rho_g^{\mathbf{A}} \otimes \rho_g^{\mathbf{B}} =: \rho_g^{\mathbf{A}} \rho_g^{\mathbf{B}}$ where we abuse the notation $\rho_g^{\mathbf{A}} = \rho_g^{\mathbf{A}} \otimes \mathrm{id}_{\mathbf{B}}, \rho_g^{\mathbf{B}} = \mathrm{id}_{\mathbf{A}} \otimes \rho_g^{\mathbf{B}}$. A symmetric operator acting on the total space, is by definition an operator $O \in \mathrm{Hom}_{\mathbf{Vec}}(\mathcal{H}, \mathcal{H})$ that commutes with symmetry actions

$$\rho_g O \rho_g^{-1} = O, \quad \forall g \in G. \tag{56}$$

In general, $O$ does not commute with "partial" group actions $\rho^{\mathbf{A}}$ or $\rho^{\mathbf{B}}$. Indeed, it can transport symmetry charge between $\mathbf{A}$ and $\mathbf{B}$. We are tempted to represent $O$ by a trivalent tree graph

$$O = \quad \begin{array}{c} \mathcal{H}_{\mathbf{A}} \qquad \mathcal{H}_{\mathbf{B}} \\ \\ \boxed{l} \\ \\ \mathcal{H}_{\mathbf{A}} \qquad \mathcal{H}_{\mathbf{B}} \end{array} \quad . \tag{57}$$

Then we interpret $X$ as the symmetry charge transported by $O$ from subsystem $\mathbf{A}$ to subsystem $\mathbf{B}$ and the intertwiners $l, r$ as describing how the charge $X$ leaves $\mathbf{A}$ and arrives as $\mathbf{B}$. However, a large enough representation $X$ can always do the job to represent $O$. We need to find the smallest $X$.

Note that there is a natural isomorphism $\epsilon$ by "bending legs":

$$\mathrm{Hom}(\mathcal{H}_{\mathbf{A}} \otimes \mathcal{H}_{\mathbf{B}}, \mathcal{H}_{\mathbf{A}} \otimes \mathcal{H}_{\mathbf{B}}) \overset{\epsilon}{\cong} \mathrm{Hom}(\mathcal{H}_{\mathbf{A}}^* \otimes \mathcal{H}_{\mathbf{A}}, \mathcal{H}_{\mathbf{B}} \otimes \mathcal{H}_{\mathbf{B}}^*)$$

$$\begin{array}{c} \mathcal{H}_{\mathbf{A}} \quad \mathcal{H}_{\mathbf{B}} \qquad\qquad \mathcal{H}_{\mathbf{B}} \quad \mathcal{H}_{\mathbf{B}}^* \\ \boxed{O} \quad \overset{\epsilon}{\mapsto} \quad \boxed{O} \qquad =: \bar{O}. \\ \mathcal{H}_{\mathbf{A}} \quad \mathcal{H}_{\mathbf{B}} \qquad \mathcal{H}_{\mathbf{A}}^* \quad \mathcal{H}_{\mathbf{A}} \end{array} \tag{58}$$

The smallest $X$ is nothing but $\mathrm{Im}\,\bar{O}$.

**Remark 4.1.** If we choose bases of $\mathcal{H}_{\mathbf{A}}$ and $\mathcal{H}_{\mathbf{B}}$, $O$ can be represented by a tensor with four indices: $O_{ab}^{a'b'}$. Under the isomorphism $\epsilon$, while choosing the corresponding dual bases of $\mathcal{H}_{\mathbf{A}}^*$ and $\mathcal{H}_{\mathbf{B}}^*$, $\bar{O}$ can be represented by the tensor $\bar{O}_{a'a}^{b'b} = O_{ab}^{a'b'}$. In short, bending the legs on the graph corresponds to rotating the positions of indices of the tensors (in the corresponding bases and dual bases). Readers familiar with tensor network may find that the above is just doing SSVD for the tensor $O$ while viewing two $\mathcal{H}_{\mathbf{A}}$ legs $a', a$ as input and two $\mathcal{H}_{\mathbf{B}}$ legs $b', b$ as output.

A more physical way to understand the above is to note that the dual group representation is the anti symmetry charge. Thus, $\mathcal{H}_\mathbf{A}^* \otimes \mathcal{H}_\mathbf{A}$ may be thought as calculating the initial charge minus the final charge in **A**, i.e., the charge decrease in **A**, while $\mathcal{H}_\mathbf{B} \otimes \mathcal{H}_\mathbf{B}^*$ may be thought as calculating the final charge minus the initial charge in **B**, i.e., the charge increase in **B**. The result $\text{Im}\,\bar{O}$ is the transported charge. We conclude the above discussion in the following definition.

**Definition 4.2.** Given a symmetric operator $O$ acting on a bipartite system $\mathcal{H}_\mathbf{A} \otimes \mathcal{H}_\mathbf{B}$, *the symmetry charge transported by* $O$ from **A** to **B** is $\text{Im}\,\bar{O}$, denoted by $O\!\uparrow_\mathbf{A}^\mathbf{B} := \text{Im}\,\bar{O}$, where $\bar{O}$ is given by the "bending leg" isomorphism (58).

Now we have

$$O = \quad\raisebox{-2em}{\text{(diagram)}}\quad , \tag{59}$$

where $O = (\text{id}_{\mathcal{H}_\mathbf{A}} \otimes r)(l \otimes \text{id}_{\mathcal{H}_\mathbf{B}})$, and $(l, r)$ is a pair of intertwiners coming from the image decomposition $(\bar{l}, \bar{r})$ of $\bar{O}$. Explicitly, given a symmetric operator $O \in \text{Hom}(\mathcal{H}_\mathbf{A} \otimes \mathcal{H}_\mathbf{B}, \mathcal{H}_\mathbf{A} \otimes \mathcal{H}_\mathbf{B})$, the symmetry charge transported by $O$ is extracted by the following transformations:

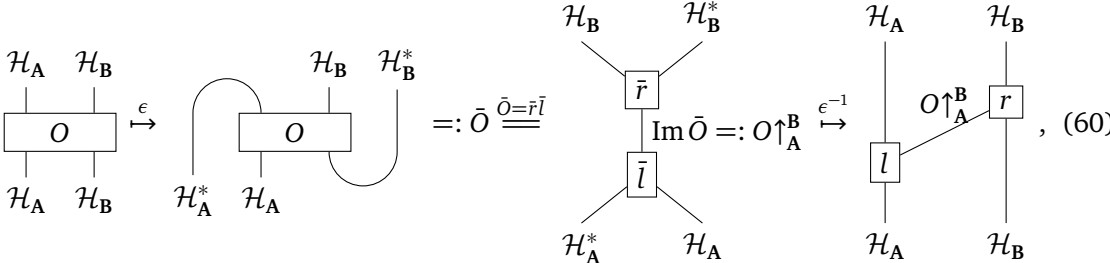

$$\tag{60}$$

where $\epsilon$ is the natural isomorphism (58), and there are further natural isomorphism $\text{Hom}(\mathcal{H}_\mathbf{A}, \mathcal{H}_\mathbf{A} \otimes O\!\uparrow_\mathbf{A}^\mathbf{B}) \cong \text{Hom}(\mathcal{H}_\mathbf{A}^* \otimes \mathcal{H}_\mathbf{A}, O\!\uparrow_\mathbf{A}^\mathbf{B})$ giving a bijection between $\bar{l}$ and $l$, and $\text{Hom}(O\!\uparrow_\mathbf{A}^\mathbf{B} \otimes \mathcal{H}_\mathbf{B}, \mathcal{H}_\mathbf{B}) \cong \text{Hom}(O\!\uparrow_\mathbf{A}^\mathbf{B}, \mathcal{H}_\mathbf{B} \otimes \mathcal{H}_\mathbf{B}^*)$ giving a bijection between $\bar{r}$ and $r$. These three steps are

1. Natural isomorphism $\epsilon$: We apply the rotation of symmetric tensor $\bar{O}_{a'a}^{b'b} = O_{ab}^{a'b'}$;

2. Image decomposition $\bar{O} = \bar{r}\bar{l}$: SSVD in Definition 3.2;

3. Natural isomorphism $\epsilon^{-1}$: We apply the inverse rotation of tensors $l_b^{ac} = \bar{l}_{ab}^c$ and $r_{ca}^b = \bar{r}_c^{ba}$.

**Remark 4.3.** The symmetry charge $O\!\uparrow_\mathbf{B}^\mathbf{A}$ transported by $O$ from **B** to **A** is similarly defined, and one have $O\!\uparrow_\mathbf{B}^\mathbf{A} = (O\!\uparrow_\mathbf{A}^\mathbf{B})^*$; the total symmetry charge is conserved.

**Example 4.4.** Let $G = S_3$ (Example 2.12). Consider a symmetric operator $O \in \text{Hom}(\Lambda \otimes \Lambda, \Lambda \otimes \Lambda)$ taking the following form:

$$\raisebox{-2em}{\text{(diagram)}} = \frac{1}{2}(\sigma_x^\mathbf{A}\sigma_x^\mathbf{B} + \sigma_y^\mathbf{A}\sigma_y^\mathbf{B}) = \begin{pmatrix} 0 & 0 & 0 & 0 \\ 0 & 0 & 1 & 0 \\ 0 & 1 & 0 & 0 \\ 0 & 0 & 0 & 0 \end{pmatrix}. \tag{61}$$

We go through the process in Diagram (60):

1. After rotation of symmetric tensor, we have $\bar{O}^{01}_{10} = O^{10}_{01} = 1$ and $\bar{O}^{10}_{01} = O^{01}_{10} = 1$.

2. Consider the direct sum decomposition $\Lambda^* \otimes \Lambda \cong \Lambda \otimes \Lambda^* \cong \Lambda \oplus \lambda_0 \oplus \lambda_1$ and the basis change between them is

$$p^{\Lambda}_{\Lambda^* \otimes \Lambda} = \begin{pmatrix} 0 & 1 & 0 & 0 \\ 0 & 0 & 1 & 0 \end{pmatrix}, \quad p^{\lambda_0}_{\Lambda^* \otimes \Lambda} = \begin{pmatrix} \frac{1}{\sqrt{2}} & 0 & 0 & \frac{1}{\sqrt{2}} \end{pmatrix}, \quad p^{\lambda_1}_{\Lambda^* \otimes \Lambda} = \begin{pmatrix} \frac{1}{\sqrt{2}} & 0 & 0 & -\frac{1}{\sqrt{2}} \end{pmatrix},$$

$$p^{\Lambda}_{\Lambda \otimes \Lambda^*} = \begin{pmatrix} 0 & 0 & 1 & 0 \\ 0 & 1 & 0 & 0 \end{pmatrix}, \quad p^{\lambda_0}_{\Lambda \otimes \Lambda^*} = \begin{pmatrix} \frac{1}{\sqrt{2}} & 0 & 0 & \frac{1}{\sqrt{2}} \end{pmatrix}, \quad p^{\lambda_1}_{\Lambda \otimes \Lambda^*} = \begin{pmatrix} \frac{1}{\sqrt{2}} & 0 & 0 & -\frac{1}{\sqrt{2}} \end{pmatrix}. \tag{62}$$

Then we have

$$\bar{O} = \begin{pmatrix} 0 & 0 & 0 & 0 \\ 0 & 0 & 1 & 0 \\ 0 & 1 & 0 & 0 \\ 0 & 0 & 0 & 0 \end{pmatrix} = \begin{pmatrix} \frac{1}{\sqrt{2}} & 0 & 0 & \frac{1}{\sqrt{2}} \\ 0 & 0 & 1 & 0 \\ 0 & 1 & 0 & 0 \\ \frac{1}{\sqrt{2}} & 0 & 0 & -\frac{1}{\sqrt{2}} \end{pmatrix} \begin{pmatrix} 0 & 0 & 0 & 0 \\ 0 & 1 & 0 & 0 \\ 0 & 0 & 1 & 0 \\ 0 & 0 & 0 & 0 \end{pmatrix} \begin{pmatrix} \frac{1}{\sqrt{2}} & 0 & 0 & \frac{1}{\sqrt{2}} \\ 0 & 1 & 0 & 0 \\ 0 & 0 & 1 & 0 \\ \frac{1}{\sqrt{2}} & 0 & 0 & -\frac{1}{\sqrt{2}} \end{pmatrix}$$

$$= \begin{pmatrix} p^{\lambda_0}_{\Lambda \otimes \Lambda^*} \\ p^{\Lambda}_{\Lambda \otimes \Lambda^*} \\ p^{\lambda_1}_{\Lambda \otimes \Lambda^*} \end{pmatrix}^{\dagger} \begin{pmatrix} 0 & 0 & 0 & 0 \\ 0 & 1 & 0 & 0 \\ 0 & 0 & 1 & 0 \\ 0 & 0 & 0 & 0 \end{pmatrix} \begin{pmatrix} p^{\lambda_0}_{\Lambda^* \otimes \Lambda} \\ p^{\Lambda}_{\Lambda^* \otimes \Lambda} \\ p^{\lambda_1}_{\Lambda^* \otimes \Lambda} \end{pmatrix} = (p^{\Lambda}_{\Lambda \otimes \Lambda^*})^{\dagger} p^{\Lambda}_{\Lambda^* \otimes \Lambda}, \tag{63}$$

where we find $O{\uparrow}^{\mathbf{B}}_{\mathbf{A}} = \operatorname{Im} \bar{O} = \Lambda$. And we solve $\bar{l}$ and $\bar{r}$ as the following form:

$$\bar{O} = c_1 \underbrace{\begin{pmatrix} 0 & 0 \\ 0 & 1 \\ 1 & 0 \\ 0 & 0 \end{pmatrix}}_{\bar{r}} c_2 \underbrace{\begin{pmatrix} 0 & 1 & 0 & 0 \\ 0 & 0 & 1 & 0 \end{pmatrix}}_{\bar{l}} = \quad \begin{array}{c} \Lambda \qquad\qquad \Lambda^* \\ \diagdown \qquad \diagup \\ \boxed{\bar{r}} \\ \big| \Lambda \\ \boxed{\bar{l}} \\ \diagup \qquad \diagdown \\ \Lambda^* \qquad\qquad \Lambda \end{array}, \tag{64}$$

where $c_1$ and $c_2$ are non-zero complex numbers such that $c_1 c_2 = 1$.

3. After inverse rotation of symmetric tensors, we find $l^{00}_1 = \bar{l}^0_{01} = 1$, $l^{11}_0 = \bar{l}^1_{10} = 1$, $r^0_{11} = \bar{r}^{01}_1 = 1$, $r^1_{00} = \bar{r}^{10}_0 = 1$, i.e.,

$$O = \underbrace{\left( I \otimes c_1 \begin{pmatrix} 0 & 0 & 0 & 1 \\ 1 & 0 & 0 & 0 \end{pmatrix} \right)}_{\mathrm{id}_{\Lambda} \otimes r} \underbrace{\left( c_2 \begin{pmatrix} 0 & 1 \\ 0 & 0 \\ 0 & 0 \\ 1 & 0 \end{pmatrix} \otimes I \right)}_{l \otimes \mathrm{id}_{\Lambda}} = \quad \begin{array}{c} \Lambda \qquad\quad \Lambda \\ \big| \qquad\quad \big| \\ \big| \quad \Lambda \; \boxed{r} \\ \boxed{l} \qquad\quad \big| \\ \big| \qquad\quad \big| \\ \Lambda \qquad\quad \Lambda \end{array}. \tag{65}$$

**Example 4.5.** Let $G = Q_8$ (Example 2.13). Consider a symmetric operator $O \in \mathrm{Hom}(\Gamma \otimes \Gamma, \Gamma \otimes \Gamma)$ taking the following form:

$$
\begin{aligned}
&\overset{\Gamma \quad \Gamma}{\underset{\Gamma \quad \Gamma}{\boxed{O}}} = \sigma_x^{\mathbf{A}} \sigma_x^{\mathbf{B}} = \begin{pmatrix} 0 & 0 & 0 & 1 \\ 0 & 0 & 1 & 0 \\ 0 & 1 & 0 & 0 \\ 1 & 0 & 0 & 0 \end{pmatrix} \\[2ex]
&\overset{\epsilon}{\mapsto} \underbrace{\begin{pmatrix} 0 & 0 & 0 & 0 \\ 0 & 1 & 1 & 0 \\ 0 & 1 & 1 & 0 \\ 0 & 0 & 0 & 0 \end{pmatrix}}_{\bar{O}} = \underbrace{c_1 \begin{pmatrix} 0 & 1 & 1 & 0 \end{pmatrix}}_{\bar{r}} \underbrace{c_2 \begin{pmatrix} 0 \\ 1 \\ 1 \\ 0 \end{pmatrix}}_{\bar{l}} = \\[2ex]
&\overset{\epsilon^{-1}}{\mapsto} \underbrace{\left( I \otimes c_1 \begin{pmatrix} 0 & 1 \\ 1 & 0 \end{pmatrix} \right)}_{\mathrm{id}_\Gamma \otimes r} \underbrace{\left( c_2 \begin{pmatrix} 0 & 1 \\ 1 & 0 \end{pmatrix} \otimes I \right)}_{l \otimes \mathrm{id}_\Gamma} = ,
\end{aligned}
\tag{66}
$$

where we go through the process in Diagram (60) and apply the result in Example 3.7 (recall that $O\uparrow_{\mathbf{A}}^{\mathbf{B}} = \mathrm{Im}\,\bar{O} = \gamma_1$, $c_1 c_2 = 1$).

**Example 4.6.** Let $G = SU(2)$ (Example 2.14). Suppose that each subsystem $\mathbf{A}, \mathbf{B}$ carries a spin $\frac{1}{2}$. Consider a symmetric operator $O \in \mathrm{Hom}(\frac{1}{2} \otimes \frac{1}{2}, \frac{1}{2} \otimes \frac{1}{2})$ in the form of Heisenberg interaction:

$$
\overset{\frac{1}{2} \quad \frac{1}{2}}{\underset{\frac{1}{2} \quad \frac{1}{2}}{\boxed{O}}} = \frac{1}{3} \vec{\sigma}^{\mathbf{A}} \cdot \vec{\sigma}^{\mathbf{B}} = \frac{1}{3} \begin{pmatrix} 1 & 0 & 0 & 0 \\ 0 & -1 & 2 & 0 \\ 0 & 2 & -1 & 0 \\ 0 & 0 & 0 & 1 \end{pmatrix}.
\tag{67}
$$

We go through the first two steps in Diagram (60):

1. After rotation of symmetric tensor, we have $\bar{O}_{00}^{00} = \bar{O}_{11}^{11} = \frac{1}{3}$, $\bar{O}_{00}^{11} = \bar{O}_{11}^{00} = -\frac{1}{3}$ and $\bar{O}_{10}^{01} = \bar{O}_{01}^{10} = \frac{2}{3}$.

2. Consider the direct sum decomposition $\frac{1}{2}^* \otimes \frac{1}{2} \cong \frac{1}{2} \otimes \frac{1}{2}^* \cong 1 \oplus 0$, and the basis change between them is

$$
\begin{aligned}
p^1_{\frac{1}{2}^* \otimes \frac{1}{2}} &= \begin{pmatrix} 0 & 0 & -1 & 0 \\ \frac{1}{\sqrt{2}} & 0 & 0 & -\frac{1}{\sqrt{2}} \\ 0 & 1 & 0 & 0 \end{pmatrix}, \qquad p^0_{\frac{1}{2}^* \otimes \frac{1}{2}} = \begin{pmatrix} -\frac{1}{\sqrt{2}} & 0 & 0 & -\frac{1}{\sqrt{2}} \end{pmatrix}, \\[2ex]
p^1_{\frac{1}{2} \otimes \frac{1}{2}^*} &= \begin{pmatrix} 0 & -1 & 0 & 0 \\ \frac{1}{\sqrt{2}} & 0 & 0 & -\frac{1}{\sqrt{2}} \\ 0 & 0 & 1 & 0 \end{pmatrix}, \qquad p^0_{\frac{1}{2} \otimes \frac{1}{2}^*} = \begin{pmatrix} \frac{1}{\sqrt{2}} & 0 & 0 & \frac{1}{\sqrt{2}} \end{pmatrix}.
\end{aligned}
\tag{68}
$$

Then we have

$$
\bar{O} = \frac{1}{3} \begin{pmatrix} 1 & 0 & 0 & -1 \\ 0 & 0 & 2 & 0 \\ 0 & 2 & 0 & 0 \\ -1 & 0 & 0 & 1 \end{pmatrix}
$$

$$
= \begin{pmatrix} 0 & \frac{1}{\sqrt{2}} & 0 & \frac{1}{\sqrt{2}} \\ -1 & 0 & 0 & 0 \\ 0 & 0 & 1 & 0 \\ 0 & -\frac{1}{\sqrt{2}} & 0 & \frac{1}{\sqrt{2}} \end{pmatrix} \frac{1}{3} \begin{pmatrix} 2 & 0 & 0 & 0 \\ 0 & 2 & 0 & 0 \\ 0 & 0 & 2 & 0 \\ 0 & 0 & 0 & 0 \end{pmatrix} \begin{pmatrix} 0 & 0 & -1 & 0 \\ \frac{1}{\sqrt{2}} & 0 & 0 & -\frac{1}{\sqrt{2}} \\ 0 & 1 & 0 & 0 \\ -\frac{1}{\sqrt{2}} & 0 & 0 & -\frac{1}{\sqrt{2}} \end{pmatrix}
$$

$$
= \begin{pmatrix} p^1_{\frac{1}{2} \otimes \frac{1}{2}^*} \\ p^0_{\frac{1}{2} \otimes \frac{1}{2}^*} \end{pmatrix}^{\dagger} \frac{1}{3} \begin{pmatrix} 2 & 0 & 0 & 0 \\ 0 & 2 & 0 & 0 \\ 0 & 0 & 2 & 0 \\ 0 & 0 & 0 & 0 \end{pmatrix} \begin{pmatrix} p^1_{\frac{1}{2}^* \otimes \frac{1}{2}} \\ p^0_{\frac{1}{2}^* \otimes \frac{1}{2}} \end{pmatrix} = \frac{2}{3} (p^1_{\frac{1}{2} \otimes \frac{1}{2}^*})^{\dagger} p^1_{\frac{1}{2}^* \otimes \frac{1}{2}} , \tag{69}
$$

where we find $O{\uparrow}^{\mathbf{B}}_{\mathbf{A}} = \mathrm{Im}\, \bar{O} = 1$.

At this stage, we find that $O$ transports an angular momentum of spin 1.

$$
\frac{1}{3} \vec{\sigma}^{\mathbf{A}} \cdot \vec{\sigma}^{\mathbf{B}} = \quad \text{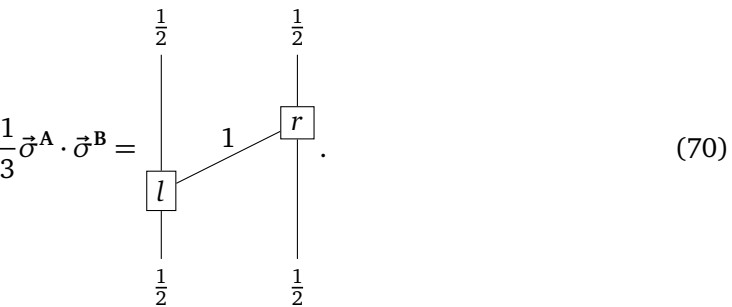} . \tag{70}
$$

Denote the basis of spin $\frac{1}{2}$ as $\{|\frac{1}{2}, \frac{1}{2}\rangle, |\frac{1}{2}, -\frac{1}{2}\rangle\}$ and the basis of spin 1 as $\{|1,1\rangle, |1,0\rangle, |1,-1\rangle\}$. One possible choice of $l, r$ can be obtained from the Clebsch-Gordan coefficients:

$$
l \left|\frac{1}{2}, \frac{1}{2}\right\rangle = \sqrt{\frac{2}{3}} \left|\frac{1}{2}, -\frac{1}{2}\right\rangle |1,1\rangle - \sqrt{\frac{1}{3}} \left|\frac{1}{2}, \frac{1}{2}\right\rangle |1,0\rangle ,
$$

$$
l \left|\frac{1}{2}, -\frac{1}{2}\right\rangle = \sqrt{\frac{1}{3}} \left|\frac{1}{2}, -\frac{1}{2}\right\rangle |1,0\rangle - \sqrt{\frac{2}{3}} \left|\frac{1}{2}, \frac{1}{2}\right\rangle |1,-1\rangle ,
$$

$$
r|1,1\rangle \left|\frac{1}{2}, -\frac{1}{2}\right\rangle = \sqrt{\frac{2}{3}} \left|\frac{1}{2}, \frac{1}{2}\right\rangle , \qquad r|1,0\rangle \left|\frac{1}{2}, \frac{1}{2}\right\rangle = -\sqrt{\frac{1}{3}} \left|\frac{1}{2}, \frac{1}{2}\right\rangle ,
$$

$$
r|1,0\rangle \left|\frac{1}{2}, -\frac{1}{2}\right\rangle = \sqrt{\frac{1}{3}} \left|\frac{1}{2}, -\frac{1}{2}\right\rangle , \qquad r|1,-1\rangle \left|\frac{1}{2}, \frac{1}{2}\right\rangle = -\sqrt{\frac{2}{3}} \left|\frac{1}{2}, -\frac{1}{2}\right\rangle . \tag{71}
$$

Other choices of $l, r$ are again only up to a scalar.

**Proposition 4.7.** When the conditions in Remark 3.5 are satisfied, we can simply define the space $O{\uparrow}^{\mathbf{B}}_{\mathbf{A}}$ to be the operator space spanned by $\rho^{\mathbf{B}}_g O \rho^{\mathbf{B}}_{g^{-1}}, g \in G$:

$$
O{\uparrow}^{\mathbf{B}}_{\mathbf{A}} = \langle \rho^{\mathbf{B}}_g O \rho^{\mathbf{B}}_{g^{-1}}, g \in G \rangle = \left\{ X \in \mathrm{Hom}(\mathcal{H}, \mathcal{H}) | X = \sum_{g \in G} a_g \rho^{\mathbf{B}}_g O \rho^{\mathbf{B}}_{g^{-1}}, a_g \in \mathbb{C} \right\} . \tag{72}
$$

In other words, $O{\uparrow}^{\mathbf{B}}_{\mathbf{A}}$ is the closure of $O$ under the "partial" conjugation group actions on operators. Note that

1. $\rho_g^{\mathbf{B}} O \rho_{g^{-1}}^{\mathbf{B}} = \rho_{g^{-1}}^{\mathbf{A}} O \rho_g^{\mathbf{A}}$.

2. $\rho_g^{\mathbf{B}} O \rho_{g^{-1}}^{\mathbf{B}}$ is in general no longer symmetric,

$$\rho_h \rho_g^{\mathbf{B}} O \rho_{g^{-1}}^{\mathbf{B}} \rho_h^{-1} = \rho_{hgh^{-1}}^{\mathbf{B}} O (\rho_{hgh^{-1}}^{\mathbf{B}})^{-1}. \tag{73}$$

3. $\rho_g^{\mathbf{B}} O \rho_{g^{-1}}^{\mathbf{B}}$ are in general not linearly independent for $g \in G$.

4. The group action on $O\!\uparrow_{\mathbf{A}}^{\mathbf{B}}$ is the "partial" conjugation by $\rho^{\mathbf{B}}$, with respect to which the tensors $l, r$ in Eq. (59) are intertwiners. More precisely, one can associate the group representation $\tau^{\mathbf{B}} : G \to GL(O\!\uparrow_{\mathbf{A}}^{\mathbf{B}})$ to $O\!\uparrow_{\mathbf{A}}^{\mathbf{B}}$ where

$$\tau_h^{\mathbf{B}}(\rho_g^{\mathbf{B}} O \rho_{g^{-1}}^{\mathbf{B}}) = \rho_h^{\mathbf{B}} \rho_g^{\mathbf{B}} O \rho_{g^{-1}}^{\mathbf{B}} \rho_{h^{-1}}^{\mathbf{B}} = \rho_{hg}^{\mathbf{B}} O \rho_{(hg)^{-1}}^{\mathbf{B}}. \tag{74}$$

In other words, as a group representation, $O\!\uparrow_{\mathbf{A}}^{\mathbf{B}}$ is understood as $(\langle \rho_g^{\mathbf{B}} O \rho_{g^{-1}}^{\mathbf{B}}, g \in G \rangle, \tau^{\mathbf{B}})$. But, it is easy to see that one can also associate this operator space with the group representation $\tau^{\mathbf{A}} : G \to GL(O\!\uparrow_{\mathbf{A}}^{\mathbf{B}})$ where

$$\tau_h^{\mathbf{A}}(\rho_g^{\mathbf{B}} O \rho_{g^{-1}}^{\mathbf{B}}) = \rho_h^{\mathbf{A}} \rho_g^{\mathbf{B}} O \rho_{g^{-1}}^{\mathbf{B}} \rho_{h^{-1}}^{\mathbf{A}} = \rho_{hg^{-1}}^{\mathbf{A}} O \rho_{h^{-1}g}^{\mathbf{A}} = \rho_{gh^{-1}}^{\mathbf{B}} O \rho_{g^{-1}h}^{\mathbf{B}}. \tag{75}$$

It is easy to see that $(\langle \rho_g^{\mathbf{B}} O \rho_{g^{-1}}^{\mathbf{B}}, g \in G \rangle, \tau^{\mathbf{A}}) = O\!\uparrow_{\mathbf{B}}^{\mathbf{A}}$ represents the symmetry charge transported by $O$ from $\mathbf{B}$ to $\mathbf{A}$. It is clear that $O\!\uparrow_{\mathbf{B}}^{\mathbf{A}}$ and $O\!\uparrow_{\mathbf{A}}^{\mathbf{B}}$ are dual to each other; the total symmetry charge of $\mathbf{A}, \mathbf{B}$ together is conserved.

*Proof.* When the conditions in Remark 3.5 are satisfied by $\bar{O}$,

$$O\!\uparrow_{\mathbf{A}}^{\mathbf{B}} := \operatorname{Im} \bar{O} = C(G)\bar{O} = \langle \rho_g^{\mathbf{B}} \otimes (\rho^{\mathbf{B}})_g^{\star} \bar{O}, g \in G \rangle. \tag{76}$$

Then graphically,

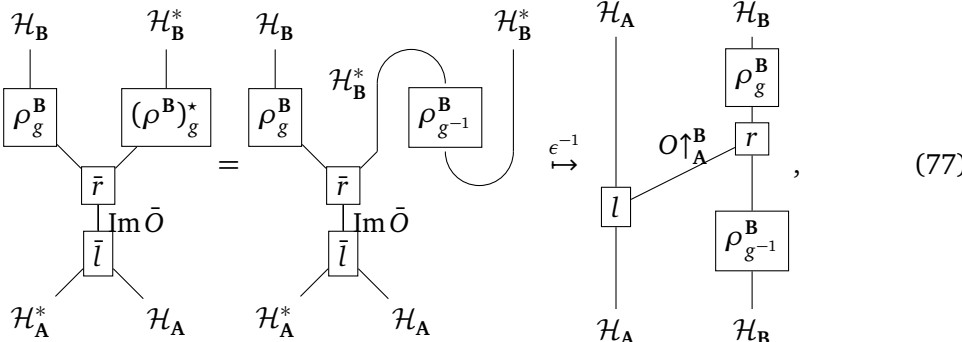

$$\tag{77}$$

whereby $O\!\uparrow_{\mathbf{A}}^{\mathbf{B}} = \langle \rho_g^{\mathbf{B}} O \rho_{g^{-1}}^{\mathbf{B}}, g \in G \rangle \cong \langle \rho_{g^{-1}}^{\mathbf{B}} O \rho_g^{\mathbf{B}}, g \in G \rangle = \langle \rho_g^{\mathbf{A}} O \rho_{g^{-1}}^{\mathbf{A}}, g \in G \rangle.$ $\qquad \square$

**Remark 4.8.** For the method in Remark 3.5 to work, the image of $\bar{O}$, i.e., the transported symmetry charge, is necessarily a cyclic representation. In Proposition 4.7, given an arbitrary symmetric operator $O \in \operatorname{End}(\mathcal{H})$, we actually do not know in advance whether the transported symmetry charge $O\!\uparrow_{\mathbf{A}}^{\mathbf{B}}$ is a cyclic representation or not. It is just a convenient method that we can try at first. If $O\!\uparrow_{\mathbf{A}}^{\mathbf{B}}$ is not cyclic, we will fail to obtain the correct result, i.e., there is no solution for $l$ and $r$ in Eq. (57) if we wrongly put $X = (\langle \rho_g^{\mathbf{B}} O \rho_{g^{-1}}^{\mathbf{B}}, g \in G \rangle, \tau^{\mathbf{B}})$. In this situation, we should extract the transported symmetry charge by the original method in Definition 4.2.

**Remark 4.9.** The regular representation and all irreducible representations of $G$ are cyclic representations of $G$.

**Example 4.10.** Let $G = \mathbb{Z}_2 := \{1, \zeta\}$. $\text{Irr}(\text{Rep}\,\mathbb{Z}_2) = \{(\alpha_+, \rho^+), (\alpha_-, \rho^-)\}$, where $\alpha_+$ is the trivial representation and $\alpha_-$ is the sign representation. Consider the regular representation on a qubit $R := C(|1\rangle, |\zeta\rangle)$, where we identify $|1\rangle$ with the logical 0 and $|\zeta\rangle$ with the logical 1, and write the nontrivial action $\rho_\zeta = \sigma_x$. Then we have $R \cong \alpha_+ \oplus \alpha_-$ where $\alpha_+ \cong \langle|+\rangle := \frac{1}{\sqrt{2}}(|1\rangle + |\zeta\rangle)\rangle$ and $\alpha_- \cong \langle|-\rangle := \frac{1}{\sqrt{2}}(|1\rangle - |\zeta\rangle)\rangle$. Consider two regions **A** and **B** both carrying the regular representation $R$, and a symmetric operator $O = \sigma_z^{\mathbf{A}}\sigma_z^{\mathbf{B}} \in \text{Hom}(R \otimes R, R \otimes R)$. Let us try the method in Proposition 4.7 to extract the symmetry charge transported by $O$:

$$O\!\uparrow_{\mathbf{A}}^{\mathbf{B}} = (\langle \sigma_z^{\mathbf{A}}\sigma_z^{\mathbf{B}}, -\sigma_z^{\mathbf{A}}\sigma_z^{\mathbf{B}}\rangle, \tau^{\mathbf{B}}) = (\langle \sigma_z^{\mathbf{A}}\sigma_z^{\mathbf{B}}\rangle, \tau^{\mathbf{B}}) \cong (\alpha_-, \rho^-). \tag{78}$$

We find no contradiction in solving $l, r$. Therefore,

$$\sigma_z^{\mathbf{A}}\,\sigma_z^{\mathbf{B}} = \quad\begin{array}{c} \alpha_+ \oplus \alpha_- \quad \alpha_+ \oplus \alpha_- \\[4pt] \end{array} \quad, \tag{79}$$

where one possible choice of $l, r$ is

$$\begin{aligned} l|1\rangle &= |1\rangle|-\rangle\,, & l|\zeta\rangle &= -|\zeta\rangle|-\rangle\,, \\ r|-\rangle|1\rangle &= |1\rangle\,, & r|-\rangle|\zeta\rangle &= -|\zeta\rangle\,, \end{aligned} \tag{80}$$

or in more compact tensor notation:

$$l^{|1\rangle|-\rangle}_{|1\rangle} = -l^{|\zeta\rangle|-\rangle}_{|\zeta\rangle} = r^{|1\rangle}_{|-\rangle|1\rangle} = -r^{|\zeta\rangle}_{|-\rangle|\zeta\rangle} = 1\,, \tag{81}$$

with other entries being zero. Other choices of $l$ and $r$ are only up to a scalar.

This result is physically easy to understand: the operator $\sigma_z$ flips the $\mathbb{Z}_2$ charge, and $O = \sigma_z^{\mathbf{A}}\sigma_z^{\mathbf{B}}$ flips the $\mathbb{Z}_2$ charges of **A** and **B** together. Therefore, $O$ indeed moves an odd $\mathbb{Z}_2$ charge from **A** to **B**.

**Example 4.11.** Let $G = S_3$ (Example 2.12).

$$\sigma_z^{\mathbf{A}}\sigma_z^{\mathbf{B}} = \quad\begin{array}{c} \Lambda \quad\quad \Lambda \\[4pt] \end{array}\quad. \tag{82}$$

Denote the basis of $\Lambda$ as $\{\mathbf{0}, \mathbf{1}\}$ and the basis of $\lambda_1$ as $\{\mathbf{o}\}$. One possible choice of $l, r$ is

$$\begin{aligned} l(\mathbf{0}) &= \mathbf{0o}\,, & l(\mathbf{1}) &= -\mathbf{1o}\,, \\ r(\mathbf{o0}) &= \mathbf{0}\,, & r(\mathbf{o1}) &= -\mathbf{1}\,. \end{aligned} \tag{83}$$

Here since the intertwiner spaces $\text{Hom}(\Lambda \otimes \lambda_1, \Lambda), \text{Hom}(\lambda_1 \otimes \Lambda, \Lambda)$ are all of dimension 1, other choices of $l$ and $r$ are only up to a scalar.

**Example 4.12.** Let $G = Q_8$ (Example 2.13).

$$\sigma_y^A \sigma_y^B = \text{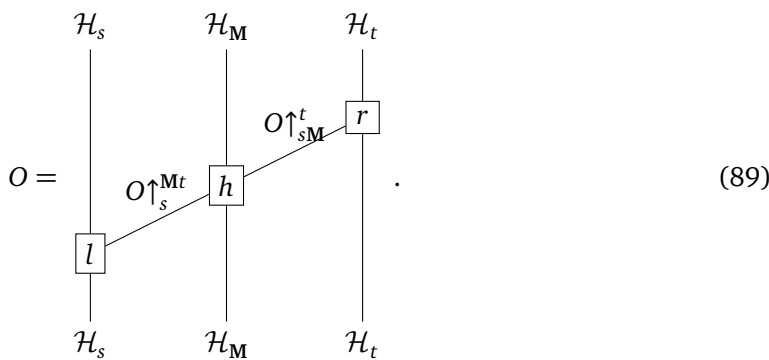}$$ 
(84)

Denote the basis of $\Gamma$ as $\{|0\rangle, |1\rangle\}$ and the basis of $\gamma_2$ as $\{|x_2\rangle\}$. One possible choice of $l, r$ is

$$
\begin{aligned}
l|0\rangle &= |1\rangle|x_2\rangle, & l|1\rangle &= -|0\rangle|x_2\rangle, \\
r|x_2\rangle|0\rangle &= -|1\rangle, & r|x_2\rangle|1\rangle &= |0\rangle.
\end{aligned}
$$
(85)

Other choices of $l$ and $r$ are only up to a scalar. And

$$\sigma_z^A \sigma_z^B = \qquad.$$
(86)

Denote the basis of $\gamma_3$ as $\{|x_3\rangle\}$. One possible choice of $l, r$ is

$$
\begin{aligned}
l|0\rangle &= |0\rangle|x_3\rangle, & l|1\rangle &= -|1\rangle|x_3\rangle, \\
r|x_3\rangle|0\rangle &= |0\rangle, & r|x_3\rangle|1\rangle &= -|1\rangle.
\end{aligned}
$$
(87)

Other choices of $l$ and $r$ are only up to a scalar.

## 4.2 Symmetry charge flows via half-braiding

Now consider a tripartite subsystem

$$\mathcal{H}_{\{s\} \sqcup \mathbf{M} \sqcup \{t\}} = \mathcal{H}_s \otimes \mathcal{H}_\mathbf{M} \otimes \mathcal{H}_t.$$
(88)

For symmetric operator $O$ acting on this subsystem, we can similarly extract the symmetry charge transported from $s\mathbf{M} := \{s\} \cup \mathbf{M}$ to $t$, or from $s$ to $\mathbf{M}t := \mathbf{M} \cup \{t\}$. By repeatedly performing the SSVD process in Diagram (60), $O$ can be represented by the following symmetric tensor

$$O = \qquad.$$
(89)

With such graphical representation, we can read out two transported charges $O\uparrow_s^{\mathbf{M}t}$ and $O\uparrow_{s\mathbf{M}}^t$, together with a symmetric tensor $h$ telling us how the charge flows through $\mathbf{M}$. Instead of such general form, here we would like to focus on the most simple form of charge transport. Suppose that we want the operator $O$ to only transport charge from $s$ to $t$ with no other effect on the intermediate region $\mathbf{M}$, and moreover, the intermediate region $\mathbf{M}$ can be arbitrary; in other words, we want to study how a constant symmetry charge can be freely transported all over the space. We arrive at the following definition of quantum current:

**Definition 4.13** (Quantum current). In a lattice system with onsite symmetry, $(\mathbf{L}, \mathcal{H}_{\mathbf{K}})$, (the Hamiltonian can be arbitrary), a *quantum current* $(Q, \beta)$ is the collection of symmetric operators of the following form

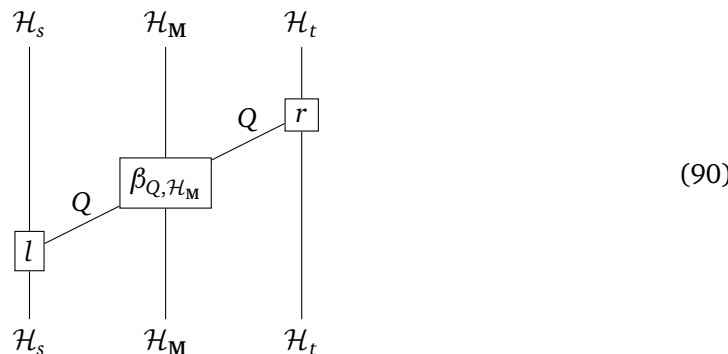

$$(90)$$

where

1. $s, t \in \mathbf{L}$ are called the source and target sites;[9]

2. $\mathbf{M} \subset \mathbf{L}$ is an arbitrary intermediate subregion, $\mathcal{H}_{\mathbf{M}} = \underset{i \in \mathbf{M}}{\otimes} \mathcal{H}_i$;

3. $Q$ is the symmetry charge transported from $s$ to $t$;

4. $l, r$ are arbitrary intertwiners in $\mathrm{Hom}(\mathcal{H}_s, \mathcal{H}_s \otimes Q)$ and $\mathrm{Hom}(Q \otimes \mathcal{H}_t, \mathcal{H}_t)$ respectively, and called source and target intertwiners;

5. $\beta$ is a collection of invertible intertwiners $\{\beta_{Q,V} : Q \otimes V \to V \otimes Q\}$ for any $V \in \mathcal{C}$,

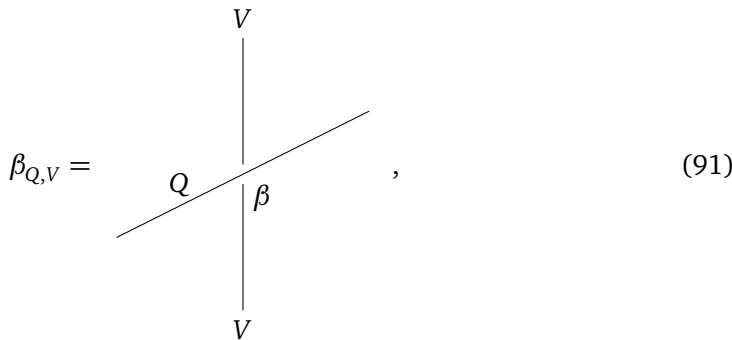

$$(91)$$

where for simplicity we omitted the subscript of $\beta$ in the graph which can be unambiguously read out from the decorations on the legs. We also draw the $\beta$ node intuitively as a crossing-over. $\beta$ must be compatible with the arbitrary choice of $\mathbf{M}$, which turns out to be the following conditions:

---

[9]To emphasize the locality of the charge transport, we use two sites instead of two regions as the source and target of the quantum current. Under renormalization, $s$ or $t$ can be the combination of several neighbouring sites in the original lattice.

(a) $\beta$ commutes with local symmetric operators (naturality): for any $f : V \to W$ in $\mathcal{C}$

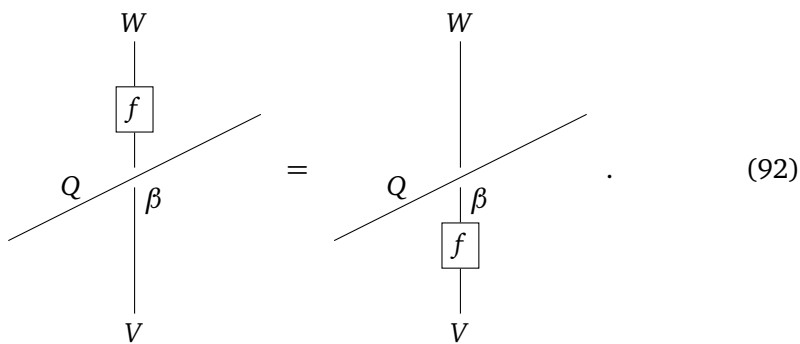

$$\tag{92}$$

(b) $\beta$ is compatible with any bipartition of the intermediate region: for any $V, W \in \mathcal{C}$,

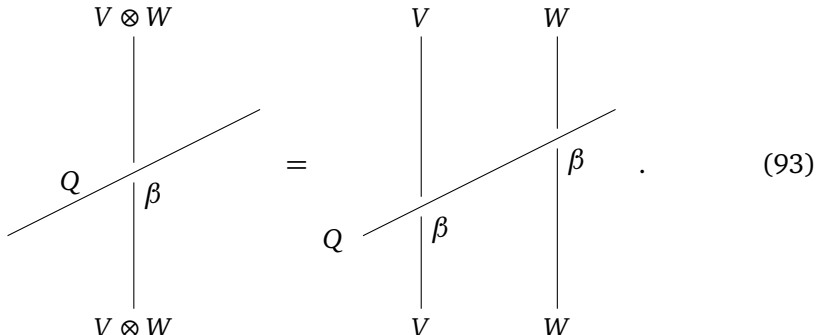

$$\tag{93}$$

**Remark 4.14.** Reducing to classical systems, the constant current in 1+1D consists of only one datum $Q$. In the quantum setting, we see that although the transported charge $Q$ is a constant, we still need the datum $\beta$ to describe how quantum mechanically the charge $Q$ flows through the local Hilbert spaces. Moreover, even if $Q$ is the same, different ways of flowing $\beta$ should be regarded as different quantum currents $(Q, \beta)$. Such $\beta$ is exactly the *half-braiding*. The above two conditions are exactly Eq. (23) and (24). Due to these conditions, not all charge $Q$ can flow over an arbitrarily long distance. Our convention for $\beta$ is also justified.

It is then clear that a quantum current is associated with the pair $(Q, \beta_{Q,-})$ which is an object in the Drinfeld center $Z_1(\mathcal{C})$. Conversely, given an object $(X, \beta_{X,-}) \in Z_1(\mathcal{C})$ we can construct quantum current operators by pasting $\beta_{X,-}$ in the intermediate sites and fusing $X$ to the source and target sites with some choice of source and target intertwiners.

**Remark 4.15.** Similar notions as *transparent patch operator* [6,7] and *unconfined (non-local) symmetric operator* [5] are defined elsewhere. Our definition here contrast with them in two aspects: (1) our definition is model independent; it only cares about the symmetry, or group representations, but not specific Hamiltonians. The condition (92), which is mathematically the naturality of half-braiding (23), requires a quantum current to commute with *any possible* local symmetric operators (including local Hamiltonians) that does not overlap with the source and the target. (2) We point out the condition (93), which is mathematically the hexagon equation for half-braiding (24), and physically the requirement for a quantum current to be able to consistently extend over an arbitrary long distance. This condition is overlooked in previous works [5–7].

**Definition 4.16.** A symmetric operator $O \in (Q, \beta)$ with a fixed choice of $s, \mathbf{M}, t, l, r$ is called a *realization* of the quantum current $(Q, \beta)$.

Let $s, \mathbf{M}, t, l$ be fixed and $r$ vary, all such realizations form a vector space, denoted by $(Q, \beta)_{l,-}^{s,\mathbf{M},t}$. This operator space admits a natural two-sided action of the local symmetric operator algebra at $t$, $\text{End}(\mathcal{H}_t) := \text{Hom}(\mathcal{H}_t, \mathcal{H}_t)$, by pre and post composition of operators. In other words, $(Q, \beta)_{l,-}^{s,\mathbf{M},t}$ is an $\text{End}(\mathcal{H}_t)$-bimodule. Similarly, $(Q, \beta)_{-,r}^{s,\mathbf{M},t}$ is an $\text{End}(\mathcal{H}_s)$-bimodule.

However, if one lets both $l, r$ vary, the set of operators $(Q, \beta)_{-,-}^{s,\mathbf{M},t}$ with fixed $s, \mathbf{M}, t$ is, in general, not even closed under addition.[10] This phenomenon is related to quantum entanglement.

**Example 4.17.** Let $G = \mathbb{Z}_2 = \{1, \zeta\}$ and $\text{Irr}(Z_1(\text{Rep}\,\mathbb{Z}_2)) = \{\mathbb{1}, m, e, \psi\}$, where we borrow the labelling of anyons in the well-known toric code model. Consider two qubits $\mathbf{A}$ and $\mathbf{B}$ both carrying the regular $\mathbb{Z}_2$ representation $R$. Denoting the simple $\mathbb{Z}_2$ charge by $(\alpha_+, \rho^+)$ and $(\alpha_-, \rho^-)$ and the charge basis of each qubit by $|+\rangle, |-\rangle$, we have $R \cong \alpha_+ \oplus \alpha_- = C(|+\rangle, |-\rangle)$ (we note that the basis we choose here is different from that in Example 4.10). We take $\beta$ to be trivial in this example ($\mathbb{1}$ and $e$ in $Z_1(\text{Rep}\,\mathbb{Z}_2)$ have trivial $\beta$) and thus omit the intermediate region $\mathbf{M}$. Let $P_i|j\rangle = \delta_{ij}|j\rangle, i, j = +, -$ be the projection operator to the fixed charge. Then

$$P_+ \otimes P_+ = \begin{pmatrix} 1 & 0 \\ 0 & 0 \end{pmatrix} \otimes \begin{pmatrix} 1 & 0 \\ 0 & 0 \end{pmatrix}, \qquad P_- \otimes P_- = \begin{pmatrix} 0 & 0 \\ 0 & 1 \end{pmatrix} \otimes \begin{pmatrix} 0 & 0 \\ 0 & 1 \end{pmatrix}. \tag{94}$$

By Proposition 4.7 and $\rho_\zeta = \sigma_z$ in basis $\{|+\rangle, |-\rangle\}$, $P_+ \otimes P_+$ and $P_- \otimes P_-$ both transport charge $\alpha_+$. However, if one adds them up,

$$O := P_+ \otimes P_+ + P_- \otimes P_- = \begin{pmatrix} 1 & 0 \\ 0 & 0 \end{pmatrix} \otimes \begin{pmatrix} 1 & 0 \\ 0 & 0 \end{pmatrix} + \begin{pmatrix} 0 & 0 \\ 0 & 1 \end{pmatrix} \otimes \begin{pmatrix} 0 & 0 \\ 0 & 1 \end{pmatrix}. \tag{95}$$

We extract the transported charge by $O$ by going through the process in Diagram (60). We have $R^* \otimes R \cong R \otimes R^* \cong \alpha_+ \oplus \alpha_+ \oplus \alpha_- \oplus \alpha_-$,[11] and the basis change between them is

$$
\begin{aligned}
p_{R^* \otimes R}^{\alpha_+;1} &= \begin{pmatrix} 1 & 0 & 0 & 0 \end{pmatrix}, & p_{R^* \otimes R}^{\alpha_-;1} &= \begin{pmatrix} 0 & 1 & 0 & 0 \end{pmatrix}, \\
p_{R^* \otimes R}^{\alpha_-;2} &= \begin{pmatrix} 0 & 0 & 1 & 0 \end{pmatrix}, & p_{R^* \otimes R}^{\alpha_+;2} &= \begin{pmatrix} 0 & 0 & 0 & 1 \end{pmatrix}, \\
p_{R \otimes R^*}^{\alpha_+;1} &= \begin{pmatrix} 1 & 0 & 0 & 0 \end{pmatrix}, & p_{R \otimes R^*}^{\alpha_-;1} &= \begin{pmatrix} 0 & 1 & 0 & 0 \end{pmatrix}, \\
p_{R \otimes R^*}^{\alpha_-;2} &= \begin{pmatrix} 0 & 0 & 1 & 0 \end{pmatrix}, & p_{R \otimes R^*}^{\alpha_+;2} &= \begin{pmatrix} 0 & 0 & 0 & 1 \end{pmatrix}.
\end{aligned}
\tag{96}
$$

Then we have

$$
\begin{aligned}
\bar{O} &= \begin{pmatrix} 1 & 0 & 0 & 0 \\ 0 & 0 & 0 & 0 \\ 0 & 0 & 0 & 0 \\ 0 & 0 & 0 & 1 \end{pmatrix} = \begin{pmatrix} 1 & 0 & 0 & 0 \\ 0 & 1 & 0 & 0 \\ 0 & 0 & 1 & 0 \\ 0 & 0 & 0 & 1 \end{pmatrix} \begin{pmatrix} 1 & 0 & 0 & 0 \\ 0 & 0 & 0 & 0 \\ 0 & 0 & 0 & 0 \\ 0 & 0 & 0 & 1 \end{pmatrix} \begin{pmatrix} 1 & 0 & 0 & 0 \\ 0 & 1 & 0 & 0 \\ 0 & 0 & 1 & 0 \\ 0 & 0 & 0 & 1 \end{pmatrix} \\
&= \begin{pmatrix} p_{R \otimes R^*}^{\alpha_+;1} \\ p_{R \otimes R^*}^{\alpha_-;1} \\ p_{R \otimes R^*}^{\alpha_-;2} \\ p_{R \otimes R^*}^{\alpha_+;2} \end{pmatrix}^\dagger \begin{pmatrix} 1 & 0 & 0 & 0 \\ 0 & 0 & 0 & 0 \\ 0 & 0 & 0 & 0 \\ 0 & 0 & 0 & 1 \end{pmatrix} \begin{pmatrix} p_{R^* \otimes R}^{\alpha_+;1} \\ p_{R^* \otimes R}^{\alpha_-;1} \\ p_{R^* \otimes R}^{\alpha_-;2} \\ p_{R^* \otimes R}^{\alpha_+;2} \end{pmatrix} = (p_{R \otimes R^*}^{\alpha_+;1})^\dagger p_{R^* \otimes R}^{\alpha_+;1} + (p_{R \otimes R^*}^{\alpha_+;2})^\dagger p_{R^* \otimes R}^{\alpha_+;2}, \tag{97}
\end{aligned}
$$

---

[10] For a quantum current $(Q, \beta)$, given two different realizations of it $(Q, \beta)_{r,l}^{s,\mathbf{M},t}$ and $(Q, \beta)_{r',l'}^{s,\mathbf{M},t}$ as two different symmetric operators, the addition of them may not still be in the set $(Q, \beta)_{-,-}^{s,\mathbf{M},t}$. This is what we mean by not *closed under addition*.

[11] Denote the dual basis of $R^*$ as $\{\delta_{|+\rangle}, \delta_{|-\rangle}\}$. $R$ is isomorphic to $R^*$ through the isomorphism $|+\rangle \mapsto \delta_{|+\rangle}, |-\rangle \mapsto \delta_{|-\rangle}$.

where we see the transported charge by $O$ is $O\!\uparrow_{\mathbf{A}}^{\mathbf{B}} = \alpha_+ \oplus \alpha_+$. Graphically,

$$
O = \quad
\begin{array}{cc}
\alpha_+ \oplus \alpha_- & \alpha_+ \oplus \alpha_- \\
\end{array}
\qquad . \tag{98}
$$

The explicit forms of $l, r$ are easy to compute and we omit them.

Therefore, we conclude that symmetric operators $P_+ \otimes P_+$, $P_- \otimes P_-$ transport the symmetry charge $\alpha_+$, different from $\alpha_+ \oplus \alpha_+$ transported by their addition, i.e., $(\alpha_+, \beta)_{-,-}^{\mathbf{A,M,B}}$ is not closed under addition.

Because of this subtlety, we introduce a finer notion, that instead of allowing arbitrary source and target intertwiners, we restrict to bimodules of $\mathrm{End}(\mathcal{H}_s) \otimes \mathrm{End}(\mathcal{H}_t)$.

**Definition 4.18.** Fix $s, \mathbf{M}, t$ to obtain a set of realizations of $(Q, \beta)$. Pick subspaces

$$
\mathfrak{L} \subset \mathrm{Hom}(\mathcal{H}_s, \mathcal{H}_s \otimes Q), \quad \mathfrak{R} \subset \mathrm{Hom}(Q \otimes \mathcal{H}_t, \mathcal{H}_t). \tag{99}
$$

We call the set of realizations with $s, \mathbf{M}, t, l \in \mathfrak{L}, r \in \mathfrak{R}$, a *sector* of $(Q, \beta)$, denoted by $(Q, \beta)_{\mathfrak{L},\mathfrak{R}}^{s,\mathbf{M},t}$, if $(Q, \beta)_{\mathfrak{L},\mathfrak{R}}^{s,\mathbf{M},t}$ forms a vector space, and moreover an $\mathrm{End}(\mathcal{H}_s) \otimes \mathrm{End}(\mathcal{H}_t)$-bimodule. We call a sector simple if this bimodule is simple .

**Definition 4.19.** Suppose a Hamiltonian $H = \sum_{\mathbf{K}} H_{\mathbf{K}}$ is given. A non-zero realization $O \in (Q, \beta)$, with non-empty $\mathbf{M}$, is called *superconducting* if $OH = HO$. A quantum current $(Q, \beta)$ is called *superconducting* if it has a realization that is superconducting.

**Remark 4.20.** A superconducting quantum current can transport charges over a long distance without costing any energy. In the earlier version of this work, we used the terminology that the quantum current is *condensed* instead of superconducting, because of the correspondence with anyon condensation in topological orders in one higher dimension. The terminology "superconducting" fits better with the idea of quantum current.

**Remark 4.21.** Alternatively, a realization $O \in (Q, \beta)_{-,-}^{s,\mathbf{M},t}$ automatically commutes with all terms $H_{\mathbf{K}}$, $\mathbf{K} \subset \mathbf{M}$, i.e. terms whose support is within $\mathbf{M}$. Therefore, $O$ can only cost energy, or create/annihilate excitations around $s$ or $t$. We can also consider whether a simple sector $(Q, \beta)_{\mathfrak{L},\mathfrak{R}}^{s,\mathbf{M},t}$ is superconducting. Suppose that $O', O \in (Q, \beta)_{\mathfrak{L},\mathfrak{R}}^{s,\mathbf{M},t}$ and that $O'$ is superconducting. By simpleness, the difference between $O$ and $O'$ must be local symmetric operators on $s$ and $t$. Thus, we know that excitations created/annihilated by $O$ around $s$ or $t$ must be of the trivial type. Therefore, a simple sector is superconducting if its realizations create/annihilate only excitations of the trivial type.

For non-simple quantum current, e.g., $(Q, \beta) = (Q_1, \beta_1) \oplus (Q_2, \beta_2)$, by definition, if any of $(Q_1, \beta_1)$ and $(Q_2, \beta_2)$ is superconducting, we will say $(Q, \beta)$ is superconducting. This definition is more natural than requiring that all realizations are superconducting, as can be seen in Section 6.1 and further explained in Remark 6.6. There, we will also elaborate more on the deep connection between the superconducting of quantum currents and the emergent symmetry of gapped quantum phases.

### 4.3 Morphisms between quantum currents

We will show that a morphism in the Drinfeld center defines a reasonable transformation, or morphism, between quantum currents.

A morphism in the Drinfeld center is an intertwiner that commutes with the half-braiding. Suppose we have a realization of quantum current with charge $Q$ and half-braiding $\beta_{Q,-}$. Pick a morphism $f : (Q, \beta_{Q,-}) \to (X, \beta_{X,-})$ in the Drinfeld center. Inserting $f$ in the graph we obtain a new realization

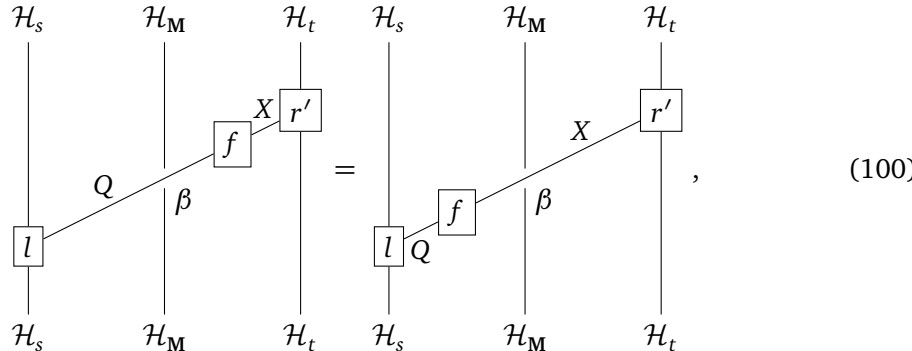

$$ \tag{100} $$

which is associated with charge $X$ and half-braiding $\beta_{X,-}$. We also see that the morphism $f$ in the Drinfeld center can change the intertwiners $l, r$ on the source and target sites. The entire process of dragging $f$ from $t$ to $s$ thus defines a map between $(Q, \beta)^{s,\mathbf{M},t}_{-,r'(f\otimes\mathrm{id}_{\mathcal{H}_t})}$ and $(X, \beta)^{s,\mathbf{M},t}_{-,r'}$. Clearly, such map, which happens on the $Q$ or $X$ leg, commutes with the action of local symmetric operators on $s$, which happens on the $\mathcal{H}_s$ legs. In other words, $f$ defines an $\mathrm{End}(\mathcal{H}_s)$-bimodule map. Meanwhile, the map $r' \mapsto r'(f \otimes \mathrm{id}_{\mathcal{H}_t})$ also commutes with action of $\mathrm{End}(\mathcal{H}_t)$.

**Remark 4.22.** It is desired to have a notion of morphisms between quantum currents, purely from the point of view of symmetric tensors, such that the category of quantum currents is equivalent to the Drinfeld center. We are unclear about such a definition for the moment. The above discussion suggests to define the bimodule maps over local operator algebras as the morphisms between quantum currents, which is the point of view taken in [5,8]. It should work for half-infinitely long quantum currents, but not work well for finitely long quantum currents.

**Conjecture 1.** The morphisms between $(Q, \beta_{Q,-})$ and $(X, \beta_{X,-})$ in the Drinfeld center are in bijection with the $\mathrm{End}(\mathcal{H}_s) \otimes \mathrm{End}(\mathcal{H}_t)$-bimodule maps between the sectors $(Q, \beta)^{s,\mathbf{M},t}_{\mathfrak{L},\mathfrak{R}}$ and $(X, \beta)^{s,\mathbf{M},t}_{\mathfrak{L}',\mathfrak{R}'}$ for large enough $\mathfrak{L}, \mathfrak{R}, \mathfrak{L}', \mathfrak{R}'$ and $\mathcal{H}_\mathbf{M}$.

## 5 Renormalization of 1+1D lattice system with symmetry

Physically, the renormalization fixed-points represent phases of matter and are thus of great interest. In this section we try to give a rigorous treatment of the renormalization process of 1+1D lattice model. Based on this, we give a general analysis of 1+1D gapped lattice model at fixed-point.

### 5.1 Renormalization fixed-point and Frobenius algebra

Recall Definition 2.15. For a 1D lattice, without losing generality, we may think the set **L** of lattice sites as a linearly ordered set.

**Definition 5.1.** Given two 1+1D quantum systems with symmetry $G$, $(\mathbf{L}, \mathcal{H}_{\mathbf{K}}, H)$ and $(\mathbf{L}', \mathcal{H}_{\mathbf{K}'}, H')$:

1. A *lattice renormalization* is an order preserving map $f : \mathbf{L} \to \mathbf{L}'$.

2. A *Hilbert space renormalization* is a pair $(f, \{U_i\})$, a lattice renormalization $f$ with a collection of intertwiners indexed by $\mathbf{L}'$, $\{U_i : \mathcal{H}_{f^{-1}(i)} \to \mathcal{H}_i\}_{i \in \mathbf{L}'}$ such that $U_i$ are partial isometries (recall Definition 2.8). $(f, \{U_i\})$ is called *proper* when $f$ is surjective. Graphically for example,

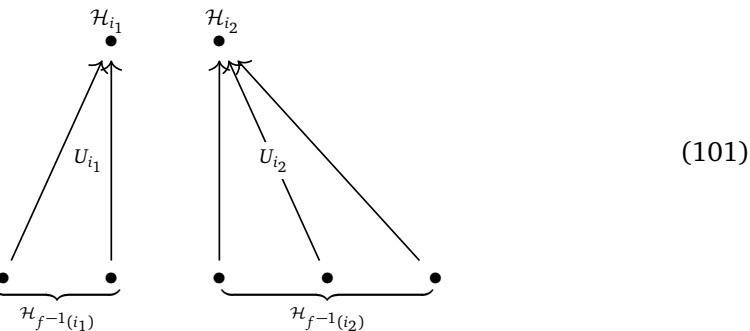

(101)

where we use ● to depict a lattice site, and $f : \mathbf{L} \to \mathbf{L}'$, $i_1, i_2 \subset \mathbf{L}'$, and $f^{-1}(i_1), f^{-1}(i_2) \subset \mathbf{L}$.

3. Let $U = \otimes_{i \in \mathbf{L}'} U_i$. When the following condition holds,

$$UHU^\dagger = UU^\dagger (H' + \Delta E) UU^\dagger,$$

(102)

where $\Delta E$ is some energy zero point shift, we call the pair $(f, \{U_i\})$ a *(system) renormalization*, and meanwhile call $(\mathbf{L}', \mathcal{H}_{\mathbf{K}'}, H')$ the *renormalized system* of $(\mathbf{L}, \mathcal{H}_{\mathbf{K}}, H)$ after $(f, \{U_i\})$.

**Remark 5.2.** For a proper Hilbert space renormalization $(f, \{U_i\})$, the image of $U_i$ should be viewed as the low energy degrees of freedom after renormalization, which justifies our definition: $U_i^\dagger U_i$ and $U_i U_i^\dagger$ are projections onto the low energy subspace, and the inner product is preserved within the low energy subspace. Equation (102) just says that the restrictions of $H'$ and $H$ in the low energy subspace are equal, and serve as the effective Hamiltonian. The energy shift can be equivalently moved to the left-hand side

$$U(H - \Delta E)U^\dagger = UU^\dagger H' UU^\dagger.$$

(103)

When $f$ is not surjective, for $f^{-1}(i) = \emptyset$, $U_i : \mathcal{H}_\emptyset = \mathbb{C} \to \mathcal{H}_i$ just fixes a state in $\mathcal{H}_i$. In other words, the non-proper Hilbert space renormalization adds sites in fixed states to the original system.

**Remark 5.3.** There is a natural composition of Hilbert space renormalizations

$$(g, \{W_i\}) \circ (f, \{U_j\}) := (gf, \{W_i(\otimes_{j \in g^{-1}(i)} U_j)\}).$$

(104)

Graphically for example,

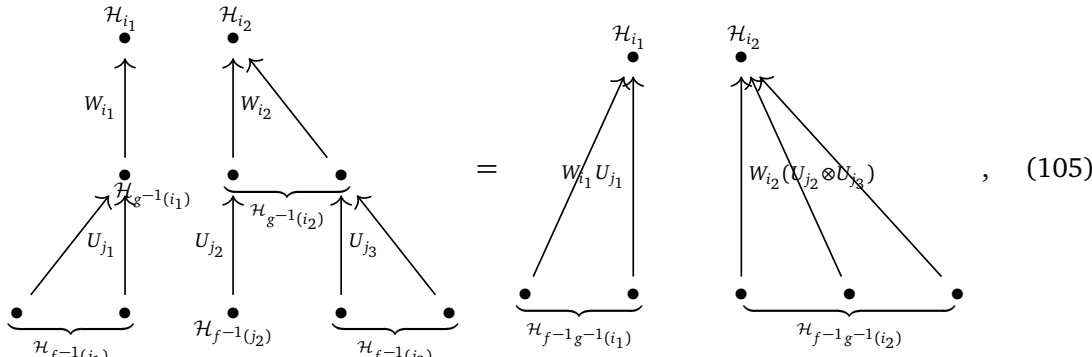

(105)

where $f : \mathbf{L} \to \mathbf{L}'$, $g : \mathbf{L}' \to \mathbf{L}''$, $i_1, i_2 \subset \mathbf{L}''$, $g^{-1}(i_1), g^{-1}(i_1) \subset \mathbf{L}'$ and $f^{-1}(j_1), f^{-1}(j_2), f^{-1}(j_3) \subset \mathbf{L}$.

We can speak of whether a system $(\mathbf{L}, \mathcal{H}_{\mathbf{K}}, H)$ is invariant with respect to a renormalization $(f, \{U_i\})$. However, in practice we expect to learn from renormalization some information of quantum phases, which are quantum systems at thermodynamic limit (infinite system size) whose interactions are local. To this end, we focus on the special case $\mathbf{L} = \mathbb{Z}$, i.e. infinite chain.

**Definition 5.4.** A quantum system $(\mathbb{Z}, \mathcal{H}_{\mathbf{K}}, H = \sum_{\mathbf{K}} H_{\mathbf{K}})$ is called *local* if all terms $H_{\mathbf{K}}$ are of finite support.

The quantum phases are also almost uniform with (at-least emergent) translation invariance.

**Definition 5.5.** A local quantum system $(\mathbb{Z}, \mathcal{H}_{\mathbf{K}}, H)$ is called *translation invariant* if $\mathcal{H}_i = A$ for all $i \in \mathbb{Z}$ and $H = \sum_i P_i$ where $P_i$ is supported on $\{i, \dots, i+n-1\}$ and has the form $P_i = P \otimes \mathrm{id}$ for some $P \in \mathrm{End}(A^{\otimes n})$. We may simply denote a translation invariant system by $(\mathbb{Z}, A, P)$.

For translation invariant systems, we hope to study the renormalizations that only depend on how many sites are combined into one, but not on positions.

**Definition 5.6.** Let $\{m_n : A^{\otimes n} \to A\}_{n \in \mathbb{N}}$ be partial isometries. A Hilbert space renormalization $(f, \{U_i\})$, between the translation invariant local systems $(\mathbb{Z}, A, P)$ and $(\mathbb{Z}, A, P')$, is called *(locally) generated* by $\{m_n\}$ if for any $i \in \mathbb{Z}$, $f^{-1}(i)$ is finite, and $U_i = m_{|f^{-1}(i)|}$.

Graphically, for example we sketch a Hilbert space renormalization generated by $\{m_0, m_1, m_2\}$ on the following local patch,

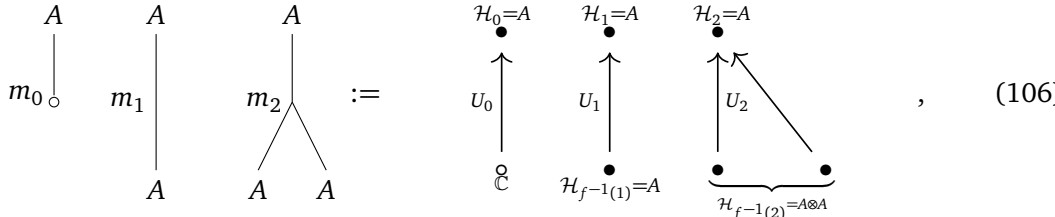

$$(106)$$

where we use $\circ$ to depict the empty subset of the lattice whose associated Hilbert space is $\mathbb{C}$.

Now we are ready to define renormalization fixed-point.

**Definition 5.7.** We call $\{m_n\}$ a *fixed-point local generator (of renormalization)*, if $m_1 = \mathrm{id}_A$ and for any two $(f, \{U_j\})$ and $(g, \{W_i\})$ generated by $\{m_n\}$, their composition $(gf, W_i(\otimes_{j \in g^{-1}(i)} U_j))$ is still generated by $\{m_n\}$.

**Theorem 5.8.** Given a fixed-point local generator $\{m_n\}$, $m := m_2$ is an associative binary operation on $A$, and $\eta := m_0$ is the unit of $m$. $(A, m, \eta)$ is an isometric associative unital algebra, $(A, m^\dagger, \eta^\dagger)$ is an associative unital coalgebra, and $(A, m, \eta, m^\dagger, \eta^\dagger)$ is a Frobenius algebra. Note that here algebra all means algebra object in the unitary fusion category $\mathcal{C}$.

*Proof.* Recall the composition of Hilbert space renormalization in Eq. (104), we have the following equations on renormalizations:

$$(107)$$

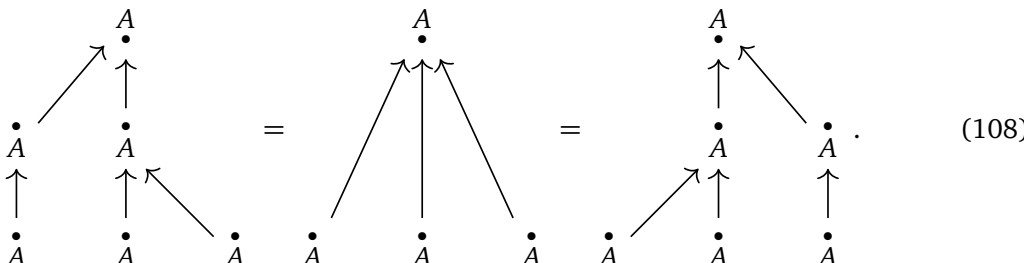

$$(108)$$

The above renormalizations are generated by $\{m_0, m_1, m_2\}$ as shown in Diagram (106). Therefore, we obtain

$$m_2(m_0 \otimes m_1) = m_1 = \mathrm{id}_A = m_2(m_1 \otimes m_0), \tag{109}$$

$$m_2(m_1 \otimes m_2) = m_3 = m_2(m_2 \otimes m_1), \tag{110}$$

which means that $(A, m = m_2, \eta = m_0)$ is an unital associative algebra. Taking Hermitian conjugate one has that $(A, m^\dagger, \eta^\dagger)$ is a unital associative coalgebra. By definition $m_n$ are partial isometries, while unitality implies that $m$ is an epimorphism[12] and $\mathrm{Im}\, m = A$, thus

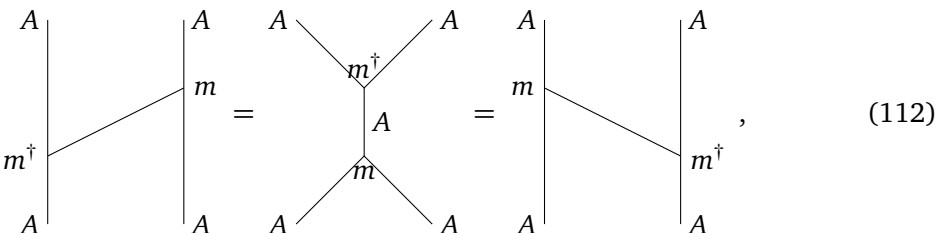

$$(111)$$

i.e., $m m^\dagger = \mathrm{id}_A$. To show the Frobenius condition

$$(112)$$

i.e., $(\mathrm{id}_A \otimes m)(m^\dagger \otimes \mathrm{id}_A) = m^\dagger m = (m \otimes \mathrm{id}_A)(\mathrm{id}_A \otimes m^\dagger)$, we introduce an auxiliary operator $\alpha = (\mathrm{id}_A \otimes m)(m^\dagger \otimes \mathrm{id}_A) - m^\dagger m$ and prove $\alpha = 0$. Direct calculation shows that

$$\alpha^\dagger \alpha = (m \otimes \mathrm{id}_A)(\mathrm{id}_A \otimes m^\dagger)(\mathrm{id}_A \otimes m)(m^\dagger \otimes \mathrm{id}_A) - m^\dagger m, \tag{113}$$

and

$$0 = (m \otimes \mathrm{id}_A)(\mathrm{id}_A \otimes \alpha^\dagger \alpha)(m^\dagger \otimes \mathrm{id}_A) = ((\mathrm{id}_A \otimes \alpha)(m^\dagger \otimes \mathrm{id}_A))^\dagger (\mathrm{id}_A \otimes \alpha)(m^\dagger \otimes \mathrm{id}_A). \tag{114}$$

Then by positive definiteness we conclude $(\mathrm{id}_A \otimes \alpha)(m^\dagger \otimes \mathrm{id}_A) = 0$ and thus

$$\alpha = (\eta^\dagger \otimes \mathrm{id}_{A \otimes A})(\mathrm{id}_A \otimes \alpha)(m^\dagger \otimes \mathrm{id}_A) = 0. \tag{115}$$

$\square$

---

[12] $f_1 m = f_2 m \implies f_1 m(\mathrm{id}_A \otimes \eta) = f_2 m(\mathrm{id}_A \otimes \eta) \implies f_1 = f_2$.

**Corollary 5.9.** For $n \geq 3$, $m_n$ is just the $n$-ary multiplication, generated by (any sequence of) the binary multiplication $m = m_2$:

$$
\begin{aligned}
m_n &= m(m \otimes \mathrm{id}_A)(m \otimes \mathrm{id}_{A \otimes A}) \cdots (m \otimes \mathrm{id}_{A^{\otimes(n-2)}}) \\
&= m(m \otimes \mathrm{id}_A)(m \otimes \mathrm{id}_{A \otimes A}) \cdots (\mathrm{id}_A \otimes m \otimes \mathrm{id}_{A^{\otimes(n-3)}}) \\
&= \cdots \\
&= m(\mathrm{id}_A \otimes m)(\mathrm{id}_{A \otimes A} \otimes m) \cdots (\mathrm{id}_{A^{\otimes(n-2)}} \otimes m).
\end{aligned}
\tag{116}
$$

Therefore, for any $n \geq 1$, $m_n$ is an epimorphism and $\mathrm{Im}\, m_n = A$. In particular, we have $m_n m_n^\dagger = \mathrm{id}_A$.

**Definition 5.10.** $(\mathbb{Z}, A, P)$ is called a (translation invariant local) *fixed-point* lattice model with respect to a fixed-point local generator $\{m_n\}$ if any *proper* Hilbert space renormalization generated by $\{m_n\}$ is a system renormalization between $(\mathbb{Z}, A, P)$ and itself. Spelling it out (recall Definition 5.1 and Corollary 5.9), $(\mathbb{Z}, A, P)$ is invariant under any renormalization $(f : \mathbb{Z} \to \mathbb{Z}, \{U_i = m_{|f^{-1}(i)|}\})$, where $f$ is surjective, $U U^\dagger = \otimes_i U_i U_i^\dagger = \mathrm{id}$ and Eq. (102) becomes

$$
U H U^\dagger = H + \Delta E.
\tag{117}
$$

In the following a Frobenius algebra is always assumed to be isometric.

**Remark 5.11.** For readers familiar with the abstract notions, we have established a connection between the renormalization process in 1+1D and the operad theory. We believe that renormalization fixed-points always have certain (weakly) associative algebraic structures.

## 5.2 Commuting projector Hamiltonian and ground states

**Theorem 5.12.** Given a Frobenius algebra $(A, m, \eta)$ in $\mathcal{C}$, $(\mathbb{Z}, A, -m^\dagger m)$ is a fixed-point lattice model. Moreover, $m^\dagger m$ is a projector and commutes with each other when acting on different sites, i.e., $(\mathbb{Z}, A, -m^\dagger m)$ is a commuting projector fixed-point lattice model.[13]

*Proof.* From the Frobenius condition (112) and the associativity of $m^\dagger$, we have

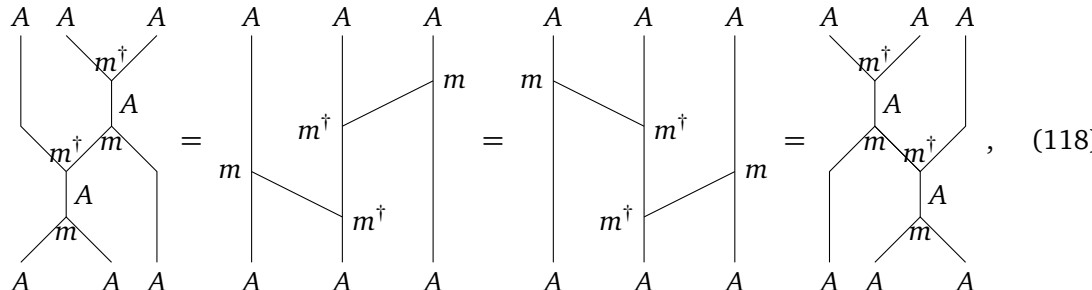

$$
\tag{118}
$$

i.e., $(\mathrm{id}_A \otimes m^\dagger m)(m^\dagger m \otimes \mathrm{id}_A) = (m^\dagger m \otimes \mathrm{id}_A)(\mathrm{id}_A \otimes m^\dagger m)$, which implies $m^\dagger m$ on neighboring sites commutes. And from Eq. (111), we have

$$
\tag{119}
$$

---

[13]The commuting-projector model is automatically gapped.

i.e., $m^\dagger m m^\dagger m = m^\dagger m$, which implies that $m^\dagger m$ is a projector.

Next, we show $(\mathbb{Z}, A, -m^\dagger m)$ is a fixed-point model. Recall Definitions 5.10 and 5.1. Let $j \in \mathbb{Z}$ be a site. Consider a special lattice renormalization

$$f_{j,n}(i) = \begin{cases} i, & i < j, \\ j, & j \le i < j+n, \\ i-(n-1), & i \ge j+n. \end{cases} \tag{120}$$

$f_{j,n}^{-1}(j) = \{j, j+1, \ldots, j+n-1\}$, $U_j = m_n$ and $U_i = \mathrm{id}_A$, $\forall i \ne j$. $(\mathbb{Z}, A, -m^\dagger m)$ is a fixed-point model with respect to fixed-point local generators $\{m_n\}$ if for any $j \in \mathbb{Z}$ and $n \ge 1$ we have

$$m_n H m_n^\dagger = H + \Delta E, \tag{121}$$

where the Hamiltonian is $H = -\sum_i (m^\dagger m)_i$. We check for

1. $n = 1$. Obviously Eq. (121) is satisfied for any $j \in \mathbb{Z}$ and $\Delta E = 0$.

2. $n = 2$ and $m = m_2$. We compute each term $(m^\dagger m)_i \otimes \mathrm{id}$ in the Hamiltonian wrapped by $m_2, m_2^\dagger$.[14] For term $i = j-1$, by Frobenius condition (112),

$$m_2 (m^\dagger m)_{j-1} m_2^\dagger = \quad = \quad = (m^\dagger m)_{j-1}. \tag{122}$$

For term $i = j$,

$$m_2 (m^\dagger m)_j m_2^\dagger = \quad = \quad = (\mathrm{id}_A)_j. \tag{123}$$

For term $i = j+1$,

$$m_2 (m^\dagger m)_{j+1} m_2^\dagger = \quad = \quad = (m^\dagger m)_j. \tag{124}$$

---

[14]We distinguish $m$ and $m_2$ here in order to emphasize $m^\dagger m$ is a term in Hamiltonian while $m_2$ is a Hilbert space renormalization local generator.

All other terms obviously commute with $m_2$. Therefore, $m_2 H m_2^\dagger = H - 1$.

3. $n \geq 3$. By similar calculations, we have $m_n H m_n^\dagger = H - (n-1)$. $\qquad\square$

**Remark 5.13.** Based on this example we explain why in Definition 5.10 we only require the fixed-point to be invariant under *proper* renormalizations. Consider the non-proper renormalization $m_0$, which adds an extra site (in the state of the unit of the algebra) to the model. We may try to include more interactions, such as $m_n^\dagger m_n$. Straightforward calculation shows that the interaction $m_n^\dagger m_n$ overlapping with this extra site gets renormalized to the interaction $m_{n-1}^\dagger m_{n-1}$. Moreover, to preserve translation invariance, we necessarily needs to add extra sites by non-proper renormalizations in a translation invariant way. Such fixed-point in a stronger sense is possible to be defined, but greatly complicates the analysis. We thus focus on the nearest neighbour interactions which is at fixed-point in a weaker sense (only invariant under proper renormalizations).

**Theorem 5.14.** The ground state subspace (of total Hilbert space) of the commuting projector fixed-point lattice model $(\mathbb{Z}, A, -m^\dagger m)$ (an infinite chain model with translation invariance) is $A$.

*Proof.* The Hamiltonian is $H = -\sum_i (m^\dagger m)_i$, where $(m^\dagger m)_i$ is supported on $\{i, i+1\}$ and has the form $(m^\dagger m)_i = m^\dagger m \otimes \mathrm{id}$ for $m^\dagger m \in \mathrm{End}(A \otimes A)$. Since all terms of $H$ are commuting projections, the ground states of $H$ are given by the common eigenstates of $(m^\dagger m)_i$ for all $i$ with eigenvalue $+1$, i.e., the ground state subspace is

$$\{|\psi\rangle \in \mathcal{H} | \forall i, (m^\dagger m)_i |\psi\rangle = |\psi\rangle\} = \{|\psi\rangle \in \mathcal{H} | \prod_i (m^\dagger m)_i |\psi\rangle = |\psi\rangle\} = \mathrm{Im} \prod_i (m^\dagger m)_i, \quad (125)$$

where we note that the equation holds as eigenvalues of a projector can only be 0 or $+1$. Graphically, from the Frobenius condition (112), we have

$$\prod_i (m^\dagger m)_i = \quad \text{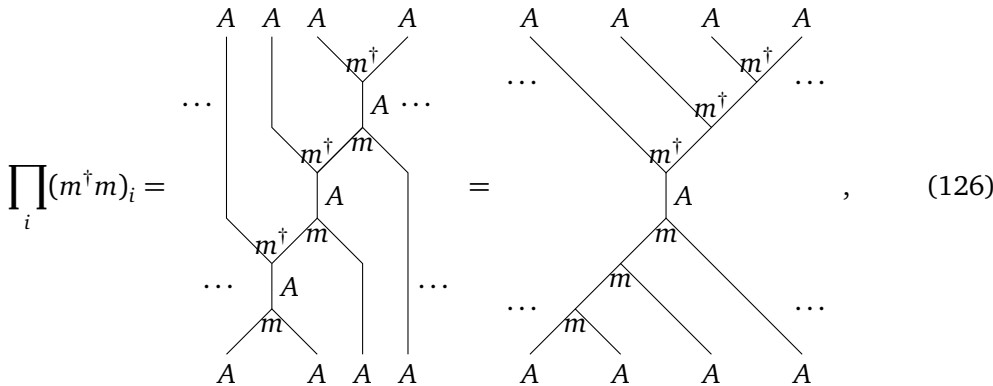} \quad , \quad (126)$$

where we see $\mathrm{Im} \prod_i (m^\dagger m)_i = A$ (we can check it satisfies Definition B.8). $\qquad\square$

## 5.3 Fixed-point boundary conditions

Now we extend the above discussion to include boundaries of 1D lattice. Without losing generality, let's consider the special case that the lattice is $\mathbf{L} = \mathbb{N}$.

**Definition 5.15.** A system $(\mathbb{N}, \mathcal{H}_{\mathbf{K}}, H)$ is called a translation invariant local system *with boundary condition* $(M, Q)$ if

- $\mathcal{H}_0 = M$, $\mathcal{H}_i = A$ for all $i \geq 1$;

- $H = Q + \sum_{i \geq 1} P_i$ where $P_i$ supported in $\{i, \dots, i+n-1\}$, $P_i = P \otimes \mathrm{id}$ for some $P \in \mathrm{End}(A^{\otimes n})$, and 0 is in the finite support of $Q$.

We may simply denote this system by $(\mathbb{N}, A, P, M, Q)$.

**Definition 5.16.** Let $\{\rho_n : M \otimes A^{\otimes n} \to M, m_n : A^{\otimes n} \to A\}_{n \in \mathbb{N}}$ be partial isometries. A Hilbert space renormalization $(f, \{U_i\})$, between the systems $(\mathbb{N}, A, P, M, Q)$ and $(\mathbb{N}, A, P', M, Q')$, is called *(locally) generated* by $\{\rho_n, m_n\}$ if for any $i \in \mathbb{N}$, $f^{-1}(i)$ is finite, $f^{-1}(0)$ is nonempty, $U_0 = \rho_{|f^{-1}(0)|-1}$, and $U_i = m_{|f^{-1}(i)|}$ for $i \geq 1$.

Graphically, for example we sketch a Hilbert space renormalization generated by $\{\rho_1, m_2\}$ on the following local patch,

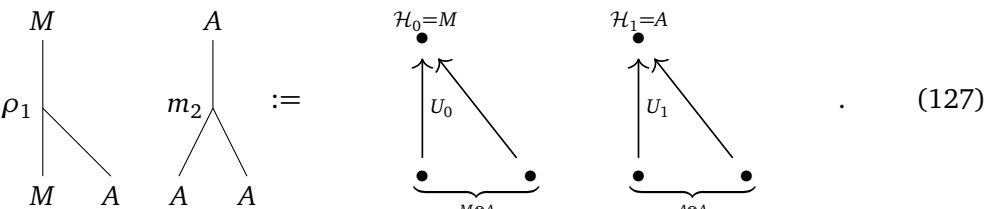

$$ \tag{127} $$

**Definition 5.17.** We call $\{\rho_n, m_n\}$ a *fixed-point local generator (with boundary)*, if

$$\rho_0 = \mathrm{id}_M, \quad m_1 = \mathrm{id}_A, \tag{128}$$

and for any two $(f, \{U_j\})$ and $(g, \{W_i\})$ generated by $\{\rho_n, m_n\}$, their composition is still generated by $\{\rho_n, m_n\}$. $(\mathbb{N}, A, P, M, Q)$ is called a (translation invariant local) *fixed-point* lattice model with respect to a fixed-point local generator $\{\rho_n, m_n\}$ if any proper Hilbert space renormalization generated by $\{\rho_n, m_n\}$ is a system renormalization between $(\mathbb{N}, A, P, M, Q)$ and itself.

**Theorem 5.18.** Given a fixed-point local generator $\{\rho_n, m_n\}$, $(A, m := m_2, \eta := m_0)$ is a Frobenius algebra as before, and $(M, \rho := \rho_1)$ is a right $A$-module (Definition C.3).

*Proof.* Similar to the proof of Theorem 5.8, by composition of Hilbert space renormalization, we have $\rho(\rho \otimes \mathrm{id}_A) = \rho(\mathrm{id}_M \otimes m)$ and $\rho(\mathrm{id}_M \otimes \eta) = \mathrm{id}_M$. $\qquad\square$

**Remark 5.19.** Similarly, one can consider a boundary condition on the right side of the chain, which is given by a left $A$-module.

**Theorem 5.20.** Let $(A, m, \eta)$ be a Frobenius algebra, $(M, \rho : M \otimes A \to M)$ a right $A$-module and $(N, \lambda : A \otimes N \to N)$ a left $A$-module, where $\rho$ and $\lambda$ are both partially isometric $\rho\rho^\dagger = \mathrm{id}_M$, $\lambda\lambda^\dagger = \mathrm{id}_N$. We have the followings:

(a) $(\rho \otimes \mathrm{id}_A)(\mathrm{id}_M \otimes m^\dagger) = \rho^\dagger\rho = (\mathrm{id}_M \otimes m)(\rho^\dagger \otimes \mathrm{id}_A)$, graphically,

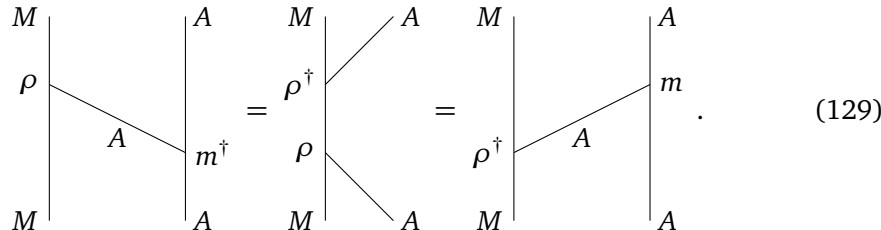

$$ \tag{129} $$

(b) $(m \otimes \mathrm{id}_N)(\mathrm{id}_A \otimes \lambda^\dagger) = \lambda^\dagger\lambda = (\mathrm{id}_A \otimes \lambda)(m^\dagger \otimes \mathrm{id}_N)$, graphically,

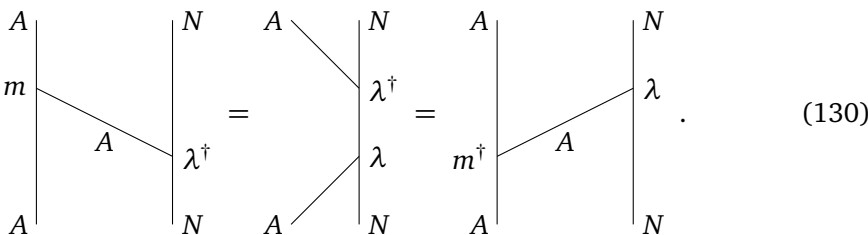

$$ \tag{130} $$

(c)  $(\rho \otimes \mathrm{id}_N)(\mathrm{id}_M \otimes \lambda^\dagger) = (\mathrm{id}_M \otimes \lambda)(\rho^\dagger \otimes \mathrm{id}_N)$
$= (\rho \otimes \lambda)(\mathrm{id}_M \otimes m^\dagger \eta \otimes \mathrm{id}_N) = (\mathrm{id}_M \otimes \eta^\dagger m \otimes \mathrm{id}_N)(\rho^\dagger \otimes \lambda^\dagger)$, graphically,

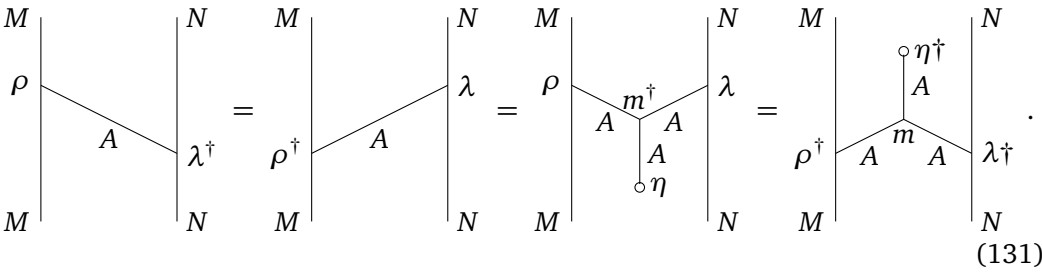

$$(131)$$

(d)  Denote by $P = (\rho \otimes \mathrm{id}_N)(\mathrm{id}_M \otimes \lambda^\dagger) : M \otimes N \to M \otimes N$ the morphism in (c). $P$ is a Hermitian projection $P^\dagger = P$ and $P^2 = P$.

(e)  Due to (d), one can take the isometric image decomposition of $P$, i.e., $r : M \otimes N \to \mathrm{Im}\, P$ such that $r^\dagger r = P$ and $r r^\dagger = \mathrm{id}_{\mathrm{Im}\, P}$. Graphically,

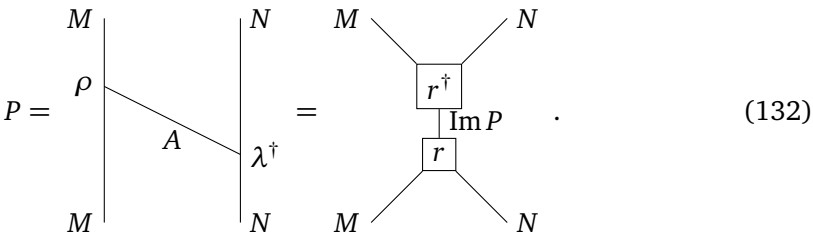

$$(132)$$

$r$ exhibits $\mathrm{Im}\, P$ as the relative tensor product $M \underset{A}{\otimes} N$.

*Proof.* (a)(b) are Frobenius-like conditions and the proof is similar to the proof of Theorem 5.8. (c) is an easy consequence of (a)(b). (d) $P$ is clearly Hermitian and $P^2 = P$ is straightforward to verify. We elaborate on (e) the universal property of $M \underset{A}{\otimes} N$ (Definitions C.5 and B.9) satisfied by $r : M \otimes N \to \mathrm{Im}\, P$. First, one can verify that

$$P(\rho \otimes \mathrm{id}_N) = P(\mathrm{id}_M \otimes \lambda), \tag{133}$$

and together with $r = r r^\dagger r = r P$ we know that

$$r(\rho \otimes \mathrm{id}_N) = r(\mathrm{id}_M \otimes \lambda). \tag{134}$$

Now suppose $f : L \otimes R \to X$ is a morphism satisfying

$$f(\rho \otimes \mathrm{id}_N) = f(\mathrm{id}_M \otimes \lambda). \tag{135}$$

We need to show that there exists a unique $\bar{f} : \mathrm{Im}\, P \to X$ such that $\bar{f} r = f$. Note that

$$f r^\dagger r = f P = f(\rho \otimes \mathrm{id}_N)(\mathrm{id}_M \otimes \lambda^\dagger) = f(\mathrm{id}_M \otimes \lambda)(\mathrm{id}_M \otimes \lambda^\dagger) = f. \tag{136}$$

The existence is guaranteed and we only need to prove the uniqueness. This is easy, since for any $\bar{f}$ satisfying $\bar{f} r = f$, one must have $\bar{f} = \bar{f} r r^\dagger = f r^\dagger$. Therefore, $(\mathrm{Im}\, P, r)$ is the coequalizer:

$$M \otimes A \otimes N \underset{\mathrm{id}_M \otimes \lambda}{\overset{\rho \otimes \mathrm{id}_N}{\rightrightarrows}} M \otimes N \overset{r}{\longrightarrow} \mathrm{Im}\, P = M \underset{A}{\otimes} N. \tag{137}$$

$\square$

**Corollary 5.21.** $(\mathbb{N}, A, -m^\dagger m, M, -\rho^\dagger \rho)$ (a half-infinite chain model) is a commuting projector fixed-point lattice model, and the ground state subspace is exactly $M$.

*Proof.* The Hamiltonian is $H = -(\rho^\dagger \rho)_0 - \sum_{i \geq 1}(m^\dagger m)_i$. The local terms commuting with each other and the model being at fixed-point is due to Theorem 5.20(a). Similarly to the infinite chain case in Theorem 5.14, the ground state subspace is $\mathrm{Im}(\rho^\dagger \rho)_0 \prod_i (m^\dagger m)_i = M$. $\qquad\square$

**Corollary 5.22.** Given an isometric algebra $(A, m, \eta)$ and right and left modules $(M, \rho)$ and $(N, \lambda)$ as in the theorem. One can define a fixed-point[15] lattice system on a finite open chain whose bulk sites have local Hilbert space $A$ with bulk local Hamiltonian terms $-m^\dagger m$, and boundary sites have Hilbert spaces $M$ and $N$ with boundary Hamiltonian $-\rho^\dagger \rho$ and $-\lambda^\dagger \lambda$. The ground state subspace of this system is $M \underset{A}{\otimes} N$.

*Proof.* Consider a finite chain model with two-side boundary conditions denoted as $(\mathbf{L} = \{0, \ldots, J\}, A, -m^\dagger m, M, -\rho^\dagger \rho, N, -\lambda^\dagger \lambda)$, where $(M, \rho)$ is a right $A$-module and $(N, \lambda)$ is a left $A$-module. Boundary Hilbert spaces are $\mathcal{H}_0 = M$ and $\mathcal{H}_J = N$. And the Hamiltonian is $H = -(\rho^\dagger \rho)_0 - (\lambda^\dagger \lambda)_{J-1} - \sum_{1 \leq i \leq J-2}(m^\dagger m)_i$. Similarly to Theorem 5.20(a), the ground state subspace is given by $\mathrm{Im}(\rho^\dagger \rho)_0 \prod_i (m^\dagger m)_i (\lambda^\dagger \lambda)_{J-1}$. By repeatedly applying the Frobenius conditions one can show that

$$\mathrm{Im}(\rho^\dagger \rho)_0 \prod_i (m^\dagger m)_i (\lambda^\dagger \lambda)_{J-1} = \mathrm{Im}(\rho \otimes \mathrm{id}_N)(\mathrm{id}_M \otimes \lambda^\dagger) = \mathrm{Im}\, P = M \underset{A}{\otimes} N. \tag{138}$$

$\qquad\square$

We may fix the bulk and consider the boundary change:

**Definition 5.23.** A *boundary change* is a system renormalization $(f, \{U_i\})$ between $(\mathbb{N}, A, P, M, Q)$ and $(\mathbb{N}, A, P, M', Q')$ such that $f = \mathrm{id}_\mathbb{N}$, and $U_i = \mathrm{id}_A$ for $i \geq 1$. In other words, a boundary change is a partial isometry $U_0 : M \to M'$ such that $U_0 U_0^\dagger (Q' + \Delta E) U_0 U_0^\dagger = U_0 Q U_0^\dagger$. Given two fixed-point local generators $\{\rho_n : M \otimes A^{\otimes n} \to M, m_n\}$ and $\{\rho_n' : M' \otimes A^{\otimes n} \to M', m_n\}$, a boundary change is called *between fixed-points* if

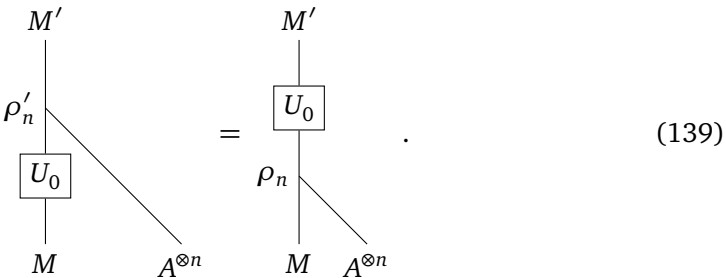

$$\tag{139}$$

**Theorem 5.24.** A boundary change $U_0 : M \to M'$ between fixed-points is an $A$-module map.

Now we consider fixing an Frobenius algebra $A$ and collect all possible fixed-point boundaries (i.e., all right $A$-modules, and $A$-module maps) to form a category, denoted by $\mathcal{C}_A$ (Definition C.9).

**Remark 5.25.** In the above analysis, the action $\rho = \rho_1$ and the boundary change $U_0$ are assumed to be partially isometric. By Proposition F.14, any $A$-module is isomorphic to a submodule of a free module $M \otimes A$ for some $M \in \mathcal{C}$. The action of the free module $M \otimes A$ is just $\mathrm{id}_M \otimes m$, which is partially isometric, $(\mathrm{id}_M \otimes m)(\mathrm{id}_M \otimes m)^\dagger = \mathrm{id}_{M \otimes A}$. Therefore, any $A$-module is

---

[15]There is no rigorous fixed-point for finite-size models. Here the fixed-point is in the sense that we borrow the local terms in infinite-size fixed-point models to define a finite-size model.

isomorphic to one with an partially isometric action. Also since $\mathcal{C}_A$ is semisimple and unitary, partially isometric $A$-module maps and general $A$-module maps differ by at most a scaling. Thus when considering all possible boundary conditions, we can safely drop the requirement for the actions and module maps to be partial isometries.

For any object $V \in \mathcal{C}$ and right $A$-module $M$, $V \otimes M$ has a natural structure of right $A$-module (Remark C.10). Thus $\mathcal{C}_A$ is automatically a left $\mathcal{C}$-module category. Such categorical action may be physically understood as enlarging the lattice of the system $(\mathbb{N}, A, P, M, Q)$ from $\mathbb{N}$ to, say, $\mathbb{N} \cup \{-1\}$, and putting a representation $V$ on site $-1$, $\mathcal{H}_{-1} = V$, and then renormalizing $\{-1, 0\}$ to $\{0\}$ so that the renormalized system is $(\mathbb{N}, A, P, V \otimes M, Q')$.

Conversely, we may ask for a pair of boundaries, $M, M'$, what is the difference between them, or in vague terms, whether there is a $V$ so that $V \otimes M \sim M'$. The precise answer is the *internal hom*, denoted by $[M, M']$, which is an object in $\mathcal{C}$ defined by the following adjunction

$$\mathrm{Hom}_{\mathcal{C}_A}(V \otimes M, M') \cong \mathrm{Hom}_{\mathcal{C}}(V, [M, M']), \tag{140}$$

or in more precise terms, $[M, M']$ is the object representing the functor $\mathrm{Hom}_{\mathcal{C}_A}(- \otimes M, M')$. We have a natural morphism $\mathrm{ev}_{M,M'} : [M, M'] \otimes M \to M'$, referred to as the evaluation, whose image under the adjunction is $\mathrm{id}_{[M,M']}$:

$$\mathrm{Hom}_{\mathcal{C}_A}([M, M'] \otimes M, M') \cong \mathrm{Hom}_{\mathcal{C}}([M, M'], [M, M']),$$
$$\mathrm{ev}_{M,M'} \longleftrightarrow \mathrm{id}_{[M,M']}. \tag{141}$$

$\mathrm{ev}_{M,M'}$ tells us how to renormalize $M$ to $M'$ by adding the representation $[M, M']$. It is the universal answer to the difference between $M$ and $M'$: possible renormalizations $V \otimes M \to M'$ are in natural bijections with intertwiners $V \to [M, M']$; as long as there is an intertwiner $V \to [M, M']$, we can do the renormalization

$$V \otimes M \to [M, M'] \otimes M \xrightarrow{\mathrm{ev}_{M,M'}} M'. \tag{142}$$

**Remark 5.26.** Using the internal hom, $\mathcal{C}_A$ can be promoted to the category enriched over $\mathcal{C}$, denoted by $^{\mathcal{C}}\mathcal{C}_A$. We will not elaborate on details of enriched categories. Technically, by the canonical construction [26, 27], we will not distinguish the pair $(\mathcal{C}, \mathcal{M})$ where $\mathcal{M}$ is a $\mathcal{C}$-module, from the enriched category $^{\mathcal{C}}\mathcal{M}$.

**Example 5.27.** If we take $A = \mathbf{1}$ the tensor unit in $\mathcal{C}$ ($A = \mathbb{C}$ the trivial representation in $\mathrm{Rep}\, G$), we have $\mathcal{C}_{\mathbf{1}} = \mathcal{C}$. When viewing $\mathcal{C}$ itself as $\mathcal{C}$-module, we have a self enriched category $^{\mathcal{C}}\mathcal{C}$ where the internal hom is $[V, W] = W \otimes V^*$. As we have explained before, tensor product is the "addition" of symmetry charge, and the dual representation is the anti charge, thus the (internal) hom is the "subtraction" of symmetry charge. For vector spaces, we have $\mathrm{Hom}_{\mathbf{Vec}}(V, W) = W \otimes V^*$; the $\mathrm{Hom}_{\mathcal{C}}$ in $\mathcal{C}$ can be viewed as an internal hom in $\mathbf{Vec}$. We may write such intuition as

$$V \otimes W \sim \text{``}V + W\text{''}, \quad [V, W] \sim \mathrm{Hom}(V, W) \sim \text{``}W - V\text{''}. \tag{143}$$

With such intuition, the internal hom adjunction Eq.(140) is "translated" to

$$\mathrm{Hom}_{\mathcal{C}}(V \otimes X, Y) \cong \mathrm{Hom}_{\mathcal{C}}(V, [X, Y]) \sim \text{``}Y - (X + V) = (Y - X) - V\text{''}. \tag{144}$$

This "translation" is a good way to understand the idea behind the internal hom adjunction.

However, there are different Hom's (in different categories) involved. Note that for the self enriched category $^{\mathcal{C}}\mathcal{C}$,

$$\mathrm{Hom}_{\mathcal{C}}(W, [V, [X, Y]) \cong \mathrm{Hom}_{\mathcal{C}}(W \otimes V, [X, Y]) \cong \mathrm{Hom}_{\mathcal{C}}(W \otimes V \otimes X, Y) \cong \mathrm{Hom}_{\mathcal{C}}(W, [V \otimes X, Y]). \tag{145}$$

By Yoneda Lemma we know that when using only internal hom, we have, rigorously

$$[V, [X, Y]] \cong [V \otimes X, Y]. \tag{146}$$

If we consider $\mathcal{C} = \mathrm{Rep}\, G$ as a usual category, $\mathrm{Hom}_{\mathcal{C}}(V, W) \neq W \otimes V^*$ does not subtract symmetry charge. To find the current and determine the local conservation of charge, we have to compute the difference of symmetry charge, and thus enriched category and internal hom is a must.

**Example 5.28.** The Drinfeld center $Z_1(\mathcal{C})$ has a natural action on $\mathcal{C}$, by forgetting the half-braiding and then taking the tensor product. Thus we can also talk about the enriched category $^{Z_1(\mathcal{C})}\mathcal{C}$ and the interal hom $[-, -]_{Z_1(\mathcal{C})}$ in $Z_1(\mathcal{C})$. In this case, the internal hom not only computes the charge difference, but also how the charge flows (i.e. the half-braiding). In other words, internal hom computes the quantum current. With such interpretation, we call the adjunction

$$\mathrm{Hom}_{\mathcal{C}}(Q \otimes X, Y) \cong \mathrm{Hom}_{Z_1(\mathcal{C})}((Q, \beta), [X, Y]_{Z_1(\mathcal{C})}), \tag{147}$$

where $(Q, \beta) \in Z_1(\mathcal{C})$ is a quantum current, $X, Y \in \mathcal{C}$, as the quantum version of equation of local conservation. It is computable in finite semisimple categories: Just let $(Q, \beta)$ run over simple objects, one has

$$\begin{aligned}
[X, Y]_{Z_1(\mathcal{C})} &\cong \oplus_{i \in \mathrm{Irr}(Z_1(\mathcal{C}))} \mathrm{Hom}(i, [X, Y]_{Z_1(\mathcal{C})}) \otimes i \\
&\cong \oplus_{i \in \mathrm{Irr}(Z_1(\mathcal{C}))} \mathrm{Hom}_{\mathcal{C}}(i \otimes X, Y) \otimes i.
\end{aligned} \tag{148}$$

Note that here we made use of the action of **Vec** on any semisimple category, that given an $n$-dimensional vector space $V \cong \mathbb{C}^{\oplus n} \in \mathbf{Vec}$, $V \otimes i \cong \mathbb{C}^{\oplus n} \otimes i \cong (\mathbb{C} \otimes i)^{\oplus n} \cong i^{\oplus n}$.

## 5.4 Fixed-point defects

We can further consider the fixed-point defects between two fixed-point models given by two Frobenius algebras $A$ and $A'$. In particular, excitations are viewed as defects in the same model (i.e., between $A$ and $A$). By similar arguments as our previous discussions on fixed-point boundaries, we have

**Definition 5.29.** A system $(\mathbb{Z}, \mathcal{H}_{\mathbf{K}}, H)$ is called a translation invariant local system *with fixed-point defect* $(B, D)$ if

- $\mathcal{H}_0 = B$, $\mathcal{H}_i = A$ for $i < 0$ and $\mathcal{H}_i = A'$ for $i > 0$;

- $H = D + \sum_{i \leq (-n)} P_i + \sum_{i \geq 1} P_i'$ where $0$ is in the finite support of $D$, $P_i$ are supported on $\{i, \ldots, i + n - 1\}$, $P_i = P \otimes \mathrm{id}$, for some $P \in \mathrm{End}(A^{\otimes n})$, and similar for $P_i'$.

We may simply denote this system by $(\mathbb{Z}, A, P, A', P', B, D)$.

Let $\{\rho_{k;n} : A^{\otimes k} \otimes B \otimes (A')^{\otimes n} \to B, m_n : A^{\otimes n} \to A, m_n' : (A')^{\otimes n} \to A'\}_{k, n \in \mathbb{N}}$ be partial isometries. A fixed-point local generator $\{\rho_{k;n}, m_n, m_n'\}$ is defined similarly as Definitions 5.16 and 5.17.

**Theorem 5.30.** Given a fixed-point local generator $\{\rho_{k;n}, m_n\}$, $(A, m := m_2, \eta := m_0)$ and $(A', m' := m_2', \eta' := m_0')$ are Frobenius algebras as before, and $(B, \rho := \rho_{0;1}, \lambda := \rho_{1;0})$ is an $A$-$A'$-bimodule (Definition C.4). We may also denote the bimodule by $_A B_{A'}$ for clarity.

*Proof.* Similar to the proof of Theorem 5.8, by composition of Hilbert space renormalization, we have $\rho(\rho \otimes \mathrm{id}_A) = \rho(\mathrm{id}_B \otimes m)$, $\rho(\mathrm{id}_B \otimes \eta) = \mathrm{id}_B$, $\lambda(\mathrm{id}_A \otimes \lambda) = \lambda(m \otimes \mathrm{id}_B)$, $\lambda(\eta \otimes \mathrm{id}_B) = \mathrm{id}_B$ and $\rho(\lambda \otimes \mathrm{id}_A) = \lambda(\mathrm{id}_A \otimes \rho)$. $\qquad \square$

**Theorem 5.31.** $(\mathbb{Z}, A, -m^\dagger m, A', -(m')^\dagger m', B, -(\lambda^\dagger \lambda)_{-1} - (\rho^\dagger \rho)_0)$ is a commuting-projector fixed-point model with fixed-point defect $B$. The ground state subspace of $(\mathbb{Z}, A, -m^\dagger m, A', -(m')^\dagger m', B, -(\lambda^\dagger \lambda)_{-1} - (\rho^\dagger \rho)_0)$ is exactly $B$.

*Proof.* The Hamiltonian is $H = -(\lambda^\dagger \lambda)_{-1} - (\rho^\dagger \rho)_0 - \sum_{i<-1}(m^\dagger m)_i - \sum_{i>0}((m')^\dagger m')_i$. By Theorem 5.20(a) and (b), the local terms commuting with each other, the model being at fixed-point, and the ground state subspace being $\mathrm{Im}(\lambda^\dagger \lambda)_{-1}(\rho^\dagger \rho)_0 \prod_{i<-1}(m^\dagger m)_i \prod_{i>0}((m')^\dagger m')_i = B$ can all be proved similarly as before. $\qquad\square$

**Theorem 5.32.** Moreover, consider three Frobenius algebras $A'', A, A'$ and two bimodules $_{A''}B'_A$, $_AB_{A'}$. One can construct a fixed-point model with two fixed-point defects $B'$ and $B$. The local terms of the Hamiltonian away from the fixed-point defects are given by the multiplication of the algebras $-(m'')^\dagger m'', -m^\dagger m, -(m')^\dagger m'$, to the left of $B'$, in between $B', B$ and to the right of $B$, respectively. The local terms around $B$ is still given by $-\lambda^\dagger \lambda - \rho^\dagger \rho$ and similar for $B'$. The ground state subspace of this system is $B' \underset{A}{\otimes} B$.

*Proof.* Similarly to Corollary 5.22, the ground state subspace is given by the image of the following intertwiner

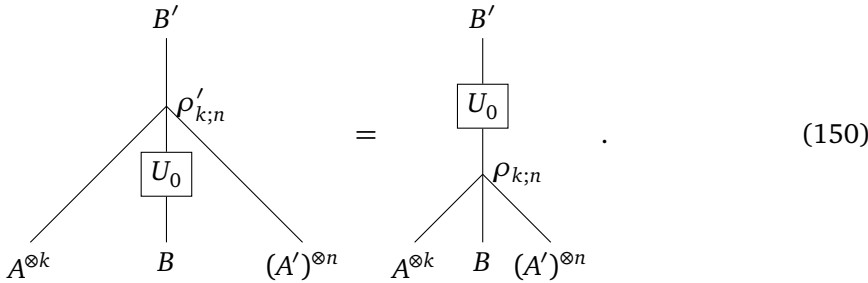

$$P' = \qquad (149)$$

which is $\mathrm{Im}\, P' = B' \underset{A}{\otimes} B$. In fact, Corollary 5.22 is a special case by taking $A'' = A' = \mathbf{1}$. $\qquad\square$

**Definition 5.33.** A *defect change* is a system renormalization $(f, \{U_i\})$ between $(\mathbb{Z}, A, P, A', P', B, D)$ and $(\mathbb{Z}, A, P, A', P', B', D')$ such that $f = \mathrm{id}_{\mathbb{Z}}$, and $U_i = \mathrm{id}_A$ or $U_i = \mathrm{id}_{A'}$ for $i \neq 0$. Given two fixed-point local generators $\{\rho_{k;n} : A^{\otimes k} \otimes B \otimes (A')^{\otimes n} \to B, m_n, m'_n\}$ and $\{\rho'_{k;n} : A^{\otimes k} \otimes B' \otimes (A')^{\otimes n} \to B', m_n, m'_n\}$, a defect change is called *between fixed-points* if

$$ \qquad (150) $$

**Theorem 5.34.** A defect change between fixed-points is an $A$-$A'$-bimodule map.

**Theorem 5.35.** The fixed-point defects and defect change between fixed-point models of the Frobenius algebras $A, A'$, form the category of $A$-$A'$-bimodules and bimodule maps, in $\mathcal{C}$, which is denoted by $_A\mathcal{C}_{A'}$. For similar reasons as explained in Remark 5.25, we do not require actions and bimodule maps in $_A\mathcal{C}_{A'}$ to be partial isometries. In particular, the excitations in the fixed-point model $(\mathbb{Z}, A, -m^\dagger m)$ form the category $_A\mathcal{C}_A$, where the fusion of excitations is given by $\underset{A}{\otimes}$.

Thus, although we build our model using symmetric operators in $\mathcal{C}$, after renormalization of $A$, we find that the category of excitations becomes $_A\mathcal{C}_A$. As symmetry is reflected by how

excitations are "added up" or "fused", we should say that there is a new emergent symmetry $_A\mathcal{C}_A$ at low energy.

Observe that in this model $(\mathbb{Z}, A, -m^\dagger m)$, the Hamiltonian $m^\dagger m$ is a symmetric operator in $\mathcal{C}$, while the emergent symmetry at fixed-point is no longer $\mathcal{C}$. Such phenomenon is in line with the usual spontaneous symmetry breaking (SSB) where the Hamiltonian has symmetry but the ground state breaks symmetry. Therefore, we consider our models as generalized spontaneous symmetry breaking phases; we call $_A\mathcal{C}_A$ connected to $\mathcal{C}$ by (generalized) SSB, i.e., by a model whose Hamiltonian has symmetry $\mathcal{C}$ but the ground state subspace is $A$ and excitations have symmetry $_A\mathcal{C}_A$. The ground state breaks symmetry unless $A$ is a Morita trivial algebra.

**Example 5.36.** The usual complete SSB is achieved by taking $\mathcal{C} = \operatorname{Rep} G$, $A = \operatorname{Fun}(G)$. In this case, $\mathcal{C}_A \cong \mathbf{Vec}$, $_A\mathcal{C}_A \cong \mathbf{Vec}_G$.

Physically, we expect that $_A\mathcal{C}_A$ should have the same quantum currents as $\mathcal{C}$. Indeed,

**Theorem 5.37** (EGNO15 [28]). As a unitary fusion category $_A\mathcal{C}_A \cong \operatorname{Fun}_\mathcal{C}(\mathcal{C}_A, \mathcal{C}_A)^{\operatorname{rev}}$, where $\operatorname{Fun}_\mathcal{C}(\mathcal{C}_A, \mathcal{C}_A)$ is the category of $\mathcal{C}$-module functors, and $^{\operatorname{rev}}$ means reversing the tensor product. $_A\mathcal{C}_A$ is (categorically) Morita equivalent to $\mathcal{C}$; they have the same Drinfeld center $Z_1(_A\mathcal{C}_A) \cong Z_1(\mathcal{C})$. Moreover, any unitary fusion category that is Morita equivalent to $\mathcal{C}$, is equivalent to $_{A'}\mathcal{C}_{A'}$ for some algebra $A' \in \mathcal{C}$.

**Corollary 5.38.** Connection by spontaneous symmetry breaking is just Morita equivalence.

# 6 The fixed-point model

The Levin-Wen or string-net models [29] in 2+1D can be understood as taking a gapped boundary condition (boundary excitations are described by a UFC) as input, and producing a lattice model for the bulk topological order [30, 31], which exhibits boundary-bulk correspondence. In the last section we have figured out the gapped fixed-point of 1+1D lattice model with a given symmetry, together with all the possible boundary conditions, as well as excitations. Below we will further analyse the fixed-point Hamiltonians given by Frobenius algebras in $\mathcal{C}$. In these models, we can verify the holographic principle: boundary determines bulk, or, bulk is the center of boundary [17–19], in the enriched setting [27, 32]:

$$Z_0(^\mathcal{C}\mathcal{C}_A) =^{Z_1(\mathcal{C})} \operatorname{Fun}_\mathcal{C}(\mathcal{C}_A, \mathcal{C}_A). \tag{151}$$

Here

- $A$ is a Frobenius algebra in $\mathcal{C}$;

- $\mathcal{C}_A$ is the category of boundary conditions, which is a $\mathcal{C}$-module, and by the canonical construction (see Remark 5.26), the boundary conditions form the enriched category $^\mathcal{C}\mathcal{C}_A$;

- $\operatorname{Fun}_\mathcal{C}(\mathcal{C}_A, \mathcal{C}_A) \cong {_A\mathcal{C}_A^{\operatorname{rev}}}$ describes excitations in the 1+1D lattice model;

- $Z_1(\mathcal{C})$, which we identify with quantum currents, are operators that transport the excitations;

- $Z_0(^\mathcal{C}\mathcal{C}_A)$ is the $E_0$-center of $^\mathcal{C}\mathcal{C}_A$ defined in the 2-category of enriched categories [27].

- $\operatorname{Fun}_\mathcal{C}(\mathcal{C}_A, \mathcal{C}_A)$ is the "relative" $E_0$-center of $\mathcal{C}_A$ with respect to $\mathcal{C}$-action: $\mathcal{C} \to \operatorname{Fun}(\mathcal{C}_A, \mathcal{C}_A)$.

Table 2: Holographic categorical symmetry viewed from $Z_0(^{\mathcal{C}}\mathcal{C}_A) =^{Z_1(\mathcal{C})} \mathrm{Fun}_{\mathcal{C}}(\mathcal{C}_A, \mathcal{C}_A)$. Here $[A,A]$ is the internal hom of Frobenius algebra $A$ (viewed as the regular $A$-$A$-bimodule), which is a Lagrangian algebra in $Z_1(\mathcal{C})$ describing superconducting quantum currents. We will introduce it in subsection 6.1 below.

| $\mathcal{C}$ | A 1+1D gapped quantum system with symmetry described by UFC $\mathcal{C}$ | A 2+1D topological order described by string-net model with input UFC $\mathcal{C}$ |
|---|---|---|
| $\mathcal{C}_A$ | 0+1D fixed-point boundary conditions | A 1+1D gapped boundary of the string-net model |
| $Z_1(\mathcal{C})$ | Quantum currents | 2+1D bulk excitations |
| $[A,A]$ | Superconducting quantum currents | Condensed bulk excitations on boundary described by $\mathcal{C}_A$ |
| $\mathrm{Fun}_{\mathcal{C}}(\mathcal{C}_A, \mathcal{C}_A)^{\mathrm{rev}}$ $\cong {}_A\mathcal{C}_A$ $\cong Z_1(\mathcal{C})_{[A,A]}$ | Fixed-point excitations | 1+1D boundary excitations |

$^{\mathcal{C}}\mathcal{C}_A$ is the data describing boundary conditions while $^{Z_1(\mathcal{C})}\mathrm{Fun}_{\mathcal{C}}(\mathcal{C}_A, \mathcal{C}_A)$ is the data describing the bulk. Again by the canonical construction, what we need to verify is that the excitations (i.e., $\mathrm{Fun}_{\mathcal{C}}(\mathcal{C}_A, \mathcal{C}_A)$) naturally form a *monoidal* [27] module over the quantum currents $Z_1(\mathcal{C})$. We further list the holographic categorical symmetry correspondence between our 1+1D model with symmetry and 2+1D topological order with gapped boundaries described by the string-net model, both of which can be characterized by Eq. (151) that taking the $E_0$-center of an enriched category, in Table 2.

**Remark 6.1.** A finite semisimple left $\mathcal{C}$-module $\mathcal{M}$ is equivalent to $\mathcal{C}_A$ for some algebra $A \in \mathcal{C}$. [28, 33]

**Remark 6.2.** Similar constructions of 1+1D lattice model can be found in Refs. [8, 23, 34]. We motivate the construction from the idea of renormalization and give analysis on excitations and quantum currents.

## 6.1 Superconducting quantum currents

In the following we show that, in the fixed-point model $(\mathbb{Z}, A, -m^{\dagger}m)$ determined by the Frobenius algebra $(A, m, \eta)$, the superconducting quantum currents form a Lagrangian algebra in the Drinfeld center $Z_1(\mathcal{C})$. Mathematically, the category of modules over the Lagrangian algebra in $Z_1(\mathcal{C})$ is automatically a monoidal module over $Z_1(\mathcal{C})$. As we will see soon, this category is also exactly the category of excitations.

Recall Definition 4.19. Supposing that the quantum current $(Q, \beta)$ is superconducting in the model $(\mathbb{Z}, A, -m^{\dagger}m)$, then there exists a realization $O \in (Q, \beta)$ whose target intertwiner commutes with the two terms of the form $m^{\dagger}m$ supported around the target site. That is to say, there is $r \in \mathrm{Hom}(Q \otimes A, A)$ such that

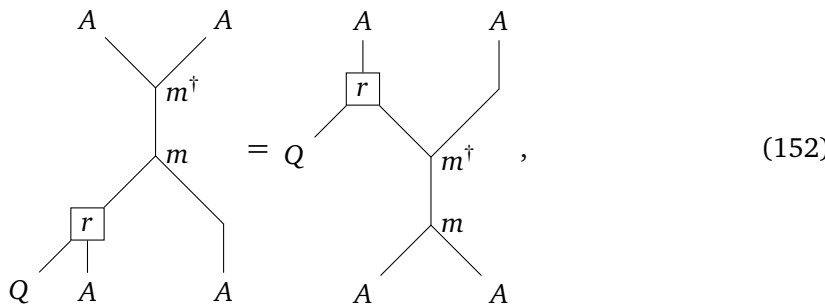

$$\tag{152}$$

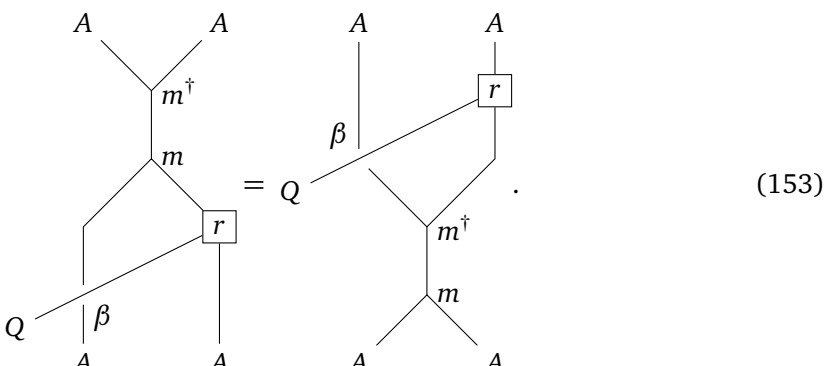

$$(153)$$

Note that when such $r$ exists, we can choose the source intertwiner $l$ of a realization to be

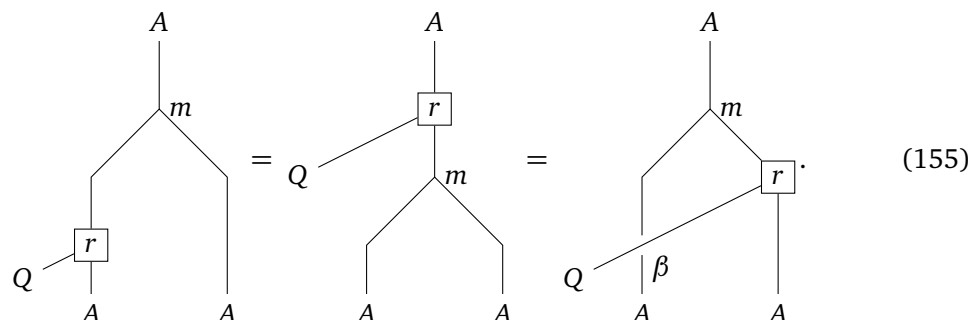

$$(154)$$

such that this realization commutes with the Hamiltonian terms around its source site. Therefore, we have the following theorem

**Theorem 6.3.** the quantum current $(Q, \beta)$ is superconducting in $(\mathbb{Z}, A, -m^\dagger m)$ if and only if the following equivalent conditions hold:

(1) There exists non-zero $r \in \text{Hom}_\mathcal{C}(Q \otimes A, A)$ satisfying Eq. (152) and (153).

(2) There exists non-zero $r \in \text{Hom}_\mathcal{C}(Q \otimes A, A)$ satisfying

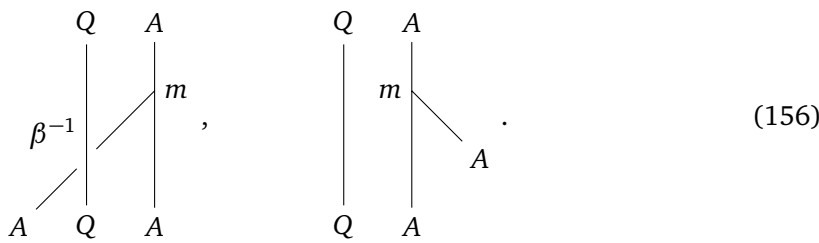

$$(155)$$

(3) There exists non-zero $r \in \text{Hom}_\mathcal{C}(Q \otimes A, A)$ which is an $A$-$A$-bimodule map, with respect to the free $A$-$A$-bimodule structure on $Q \otimes A$.

*Proof.* The left and right actions of $A$ on $Q \otimes A$ are respectively

$$(156)$$

Then, $r$ being a bimodule map simply means that

$$\tag{157}$$

$$\tag{158}$$

Then, the equivalence between (1)(2)(3) is an easy exercise using the unitality and Frobenius condition of $A$ and the naturality of $\beta$. □

**Remark 6.4.** This result is physically reasonable, that a realization of quantum current is superconducting if and only if its source and target intertwiners in $\mathcal{C}$ are also morphisms or symmetric operators (i.e., $A$-$A$-bimodule maps) in the new emergent symmetry $_A\mathcal{C}_A$.

Note that the free bimodule functor

$$- \otimes A : Z_1(\mathcal{C}) \to {}_A\mathcal{C}_A,$$
$$(Q, \beta) \mapsto Q \otimes A, \tag{159}$$

with bimodule structures given above, is moreover a central functor [35] which can be identified with $Z_1(\mathcal{C}) \cong Z_1({}_A\mathcal{C}_A) \xrightarrow{\text{Forget}} {}_A\mathcal{C}_A$. Let $[A, -]$ be the right adjoint of $- \otimes A$, by results in [35], $[A, A]$ has a canonical structure of a Lagrangian algebra in $Z_1(\mathcal{C})$, and $_A\mathcal{C}_A$ is identified with the category of right $[A, A]$-modules in $Z_1(\mathcal{C})$. We now rephrase the results of [35] in the language of internal hom.

Note that given a bimodule $B \in {}_A\mathcal{C}_A$ and an object $(Q, \beta) \in Z_1(\mathcal{C})$, $Q \otimes B$ has a natural structure of $A$-$A$-bimodule, defined similarly as that of $Q \otimes A$. In other words, $_A\mathcal{C}_A$ is a left $Z_1(\mathcal{C})$-module with module action $(Q, B) \mapsto Q \otimes B : Z_1(\mathcal{C}) \times {}_A\mathcal{C}_A \to {}_A\mathcal{C}_A$. It is not hard to check that this action is moreover a monoidal functor, and thus $_A\mathcal{C}_A$ is a monoidal module over $Z_1(\mathcal{C})$. This justifies our notation of internal hom for the right adjoint of $- \otimes A$. We now have the internal hom adjunction for any quantum current $(Q, \beta)$ and $A$-$A$-bimodules $B, B'$:

$$\text{Hom}_{{}_A\mathcal{C}_A}(Q \otimes B, B') \cong \text{Hom}_{Z_1(\mathcal{C})}((Q, \beta), [B, B']). \tag{160}$$

Be reminded that when computing this internal hom, $B, B'$ are viewed as $A$-$A$-bimodules, instead of objects in $\mathcal{C}$.[16] Given three bimodules $B, B', B''$, there is a canonical associative composition morphism

$$[B', B''] \otimes [B, B'] \to [B, B''], \tag{161}$$

defined as the image of the following morphism under the internal hom adjunction (160)

$$[B', B''] \otimes [B, B'] \otimes B \xrightarrow{\text{id}_{[B',B'']} \otimes \text{ev}_{B,B'}} [B', B''] \otimes B' \xrightarrow{\text{ev}_{B',B''}} B''. \tag{162}$$

---

[16]Do not confuse with Example 5.28. Example 5.28 may be thought as a special case with trivial algebra $A = \mathbf{1}$.

In particular, the composition $[A,A] \otimes [A,A] \to [A,A]$ is exactly the multiplication of the Lagrangian algebra $[A,A]$. Since $[A,A]$ is a commutative algebra, a right $[A,A]$-module is automatically an $[A,A]$-$[A,A]$-bimodule. The category of right $[A,A]$-modules $Z_1(\mathcal{C})_{[A,A]}$ is thus a monoidal category with monoidal structure $\underset{[A,A]}{\otimes}$. Finally, the monoidal equivalence between ${}_A\mathcal{C}_A$ and $Z_1(\mathcal{C})_{[A,A]}$ is given by the functors [33,35]

$$
{}_A\mathcal{C}_A \cong Z_1(\mathcal{C})_{[A,A]}, 
$$
$$
B \mapsto [A,B], \tag{163}
$$
$$
M \underset{[A,A]}{\otimes} A \hookleftarrow M, \tag{164}
$$

where the right action $[A,B] \otimes [A,A] \to [A,B]$ of $[A,A]$ on $[A,B]$ is given by the composition of internal hom, and the left action of $[A,A]$ on $A$ is given by the evaluation of internal hom $\mathrm{ev}_{A,A} : [A,A] \otimes A \to A$. The functor $M \mapsto M \underset{[A,A]}{\otimes} A$ is obviously monoidal, and so is its inverse $B \mapsto [A,B]$.

Coming back to the regular $A$-$A$-bimodule $A$, the adjunction $(- \otimes A) \dashv [A,-]$ reads

$$
\mathrm{Hom}_{{}_A\mathcal{C}_A}(Q \otimes A, A) \cong \mathrm{Hom}_{Z_1(\mathcal{C})}((Q,\beta), [A,A]), \tag{165}
$$

by which we know that the $A$-$A$-bimodule maps between $Q \otimes A$ and $A$ are in natural bijection with the morphisms between $(Q,\beta)$ and $[A,A]$ in $Z_1(\mathcal{C})$. Physically, $\mathrm{Hom}_{{}_A\mathcal{C}_A}(Q \otimes A, A)$ is the ways how $(Q,\beta)$ can be superconducting, which is the same as the ways $(Q,\beta)$ can be mapped into $[A,A]$, i.e., $\mathrm{Hom}_{Z_1(\mathcal{C})}(Q, [A,A])$. In other words, the quantum current $[A,A]$ provides the universal answer to how an arbitrary quantum current $(Q,\beta)$ can be superconducting. Therefore

**Theorem 6.5.** The Lagrangian algebra $[A,A] \in Z_1(\mathcal{C})$ is the universal quantum current that is superconducting in $(\mathbb{Z}, A, -m^\dagger m)$. The excitations are related to the superconducting quantum currents via

$$
\mathrm{Fun}_{\mathcal{C}}(\mathcal{C}_A, \mathcal{C}_A)^{\mathrm{rev}} \cong {}_A\mathcal{C}_A \cong Z_1(\mathcal{C})_{[A,A]}. \tag{166}
$$

The first equivalence is given by Remark E.4.

**Remark 6.6.** Consider a simple quantum current $i \in Z_1(\mathcal{C})$. If $i$ is superconducting, it is necessarily a direct summand of $[A,A]$. Moreover, the number of copies of $i$ in $[A,A]$ is the number of ways how $i$ can be superconducting. Therefore, intuitively, $[A,A]$ is the "maximal" superconducting quantum current, in the sense that $[A,A]$ contains all superconducting simple quantum currents. However, $[A,A]$ is not maximal in the literal sense; an arbitrarily large quantum current $(Q,\beta)$ is superconducting as long as it shares a simple object with $[A,A]$. Mathematically, we have to define the superconducting of quantum currents this way (Definition 4.19), such that any non-zero $A$-$A$-bimodule map $Q \otimes A \to A$ is a way of superconducting, and for any non-zero morphism $(Q',\beta') \to (Q,\beta)$, the composition $Q' \otimes A \to Q \otimes A \to A$ is still a way of superconducting. In other words, the notion of superconducting needs to be compatible with the composition of morphisms between quantum currents. Then the ways of superconducting is computed by the functor $\mathrm{Hom}_{{}_A\mathcal{C}_A}(- \otimes A, A)$ and we can extract the representing object $[A,A]$. The requirement for all direct summands of $(Q,\beta)$ to be superconducting, is equivalently restricting to the subspace spanned by monomorphisms in $\mathrm{Hom}_{Z_1(\mathcal{C})}((Q,\beta), [A,A])$, which is quite unnatural.

**Remark 6.7.** The Lagrangian algebra $[A,A]$ is the *full center* [36] of $A$.

**Remark 6.8.** Conversely, suppose that $L$ is a Lagrangian algebra in $Z_1(\mathcal{C})$. $Z_1(\mathcal{C})_L$ is then a unitary fusion category, that is Morita equivalent to $\mathcal{C}$ [35,37], and there must exist an algebra $A \in \mathcal{C}$ such that ${}_A\mathcal{C}_A \cong Z_1(\mathcal{C})_L$. In fact, such $A$ can be chosen as an indecomposable sub-algebra of the image of $L$ in $\mathcal{C}$ under the forgetful functor $Z_1(\mathcal{C}) \to \mathcal{C}$.

Based on the above analysis, we see that the excitations in the fixed-point model indeed form a monoidal module category over the quantum currents $Z_1(\mathcal{C})$. By the canonical construction, we have verified that the excitations are described by the enriched fusion category $^{Z_1(\mathcal{C})}\text{Fun}_{\mathcal{C}}(\mathcal{C}_A, \mathcal{C}_A)$.

## 6.2 The universal model

Let $\mathcal{C} = \text{Rep}\,G$. First, we consider Frobenius algebras $A := \text{Fun}(G/H)$ in $\text{Rep}\,G$ (Proposistion F.8 and Example F.9), which are $\mathbb{C}$-valued functions on cosets $G/H$. If we just take local Hilberts space to be $A$, the model $(\mathbb{Z}, A, -m^\dagger m)$ may be too small to for us to see all possible excitations. Thus, we prefer to take $(\mathbb{Z}, \text{Fun}(G), -(\iota_A \otimes \iota_A)m^\dagger m(\iota_A^\dagger \otimes \iota_A^\dagger))$, where the local Hilbert space is the large enough vector space $\text{Fun}(G)$, and

$$\iota_A : A \to \text{Fun}(G),$$
$$\sum_{gH} a_{gH} gH \mapsto \sum_{gH} a_{gH} \sum_{x \in gH} x, \tag{167}$$

is the embedding of $A$ into $\text{Fun}(G)$. Graphically, the local term of the Hamiltonian is

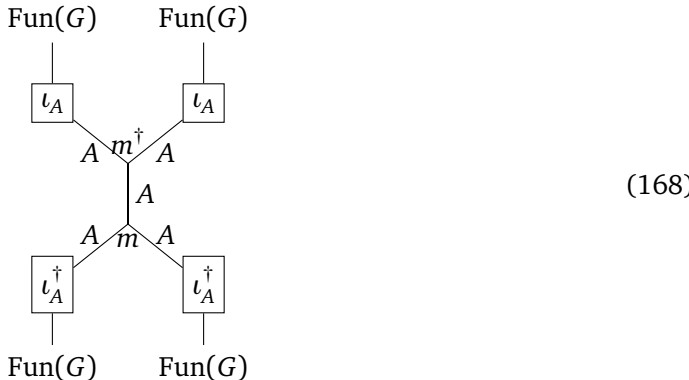

$$(168)$$

It is not hard to check that $(\mathbb{Z}, A, -m^\dagger m)$ is the renormalized system of $(\mathbb{Z}, \text{Fun}(G), -(\iota_A \otimes \iota_A)m^\dagger m(\iota_A^\dagger \otimes \iota_A^\dagger))$ after the Hilbert space renormalization $(\text{id}_{\mathbb{Z}}, \{U_i = \iota_A^\dagger, \forall i \in \mathbb{Z}\})$.

For the Frobenius algebras $(H, \omega_2)$ with nontrivial $\omega_2$ as in Example F.10, they can be embedded into two adjacent sites $\text{Fun}(G) \otimes \text{Fun}(G)$. In other words, one can similarly define a Hamiltonian on $(\mathbb{Z}, \text{Fun}(G))$ that renormalizes to $(\mathbb{Z}, A, -m^\dagger m)$ using the embedding in Example F.10. These pairs $(H, \omega_2)$ classify all 1+1D bosonic gapped phases with $G$-symmetry [38–40]. The physical meaning of the data $(H, \omega_2)$ is that the symmetry is spontaneously broken from $G$ to $H$, i.e., $H$ is the unbroken subgroup, and $\omega_2 \in H^2(H, U(1))$ labels the SPT (symmetry protected topological) order under the unbroken $H$. Therefore, the lattice $(\mathbb{Z}, \text{Fun}(G))$ can host all possible $G$-symmetric gapped phases.

Moreover, $\text{Fun}(G)$ as the regular group representation, contains all possible irreps of $G$

$$\text{Fun}(G) \cong \oplus_{i \in \text{Irr}(\text{Rep}(G))} i^{\oplus d_i}, \tag{169}$$

where $d_i$ is the dimension of irrep $i$. The simple fixed-point boundary conditions and fixed-point defects, can be embedded into a suitable free module $i \otimes A$ or free bimodule $A \otimes i \otimes A'$ (see Remark F.15) and then embedded into several (at most five) $\text{Fun}(G)$ sites. Therefore, we know in the models constructed on the lattice $(\mathbb{Z}, \text{Fun}(G))$, indeed all possible fixed-point boundary conditions and fixed-point defects/excitations can be seen.

Table 3: Frobenius algebra $A$, $\mathcal{C}_A$, $_A\mathcal{C}_A$ and Lagrangian algebra $[A,A]$ in $\mathrm{Rep}\,\mathbb{Z}_2$. Each $H$ is a subgroup of $G$. The ground state subspace is $A$. $\mathcal{C}_A \cong \mathrm{Rep}\,H$ is given in Proposition F.12. And in the column "unbroken symmetry" we list the left symmetry if there is a SSB, or $G$ if there is no SSB.

| $A = \mathrm{Fun}(G/H)$ | $\mathcal{C}_A \cong \mathrm{Rep}\,H$ | Unbroken symmetry | $_A\mathcal{C}_A$ | $[A,A]$ |
|---|---|---|---|---|
| $\mathrm{Fun}(\mathbb{Z}_2)$ | **Vec** | None | $\mathbf{Vec}_{\mathbb{Z}_2}$ | $\mathbb{1} \oplus e$ |
| $\mathrm{Fun}(\mathbb{Z}_2/\mathbb{Z}_2) = \alpha_+$ | $\mathrm{Rep}\,\mathbb{Z}_2$ | $\mathbb{Z}_2$ | $\mathrm{Rep}\,\mathbb{Z}_2$ | $\mathbb{1} \oplus m$ |

## 6.3 Example $G = \mathbb{Z}_2$

Let $G = \mathbb{Z}_2 = \{1, \zeta\}$ and $\mathrm{Irr}(\mathrm{Rep}\,\mathbb{Z}_2) = \{(\alpha_+, \rho^+), (\alpha_-, \rho^-)\}$. Simple objects in $Z_1(\mathrm{Rep}\,\mathbb{Z}_2)$ are

$$(C_1, \alpha_+) =: \mathbb{1}, (C_\zeta, \alpha_+) =: m, (C_1, \alpha_-) =: e, (C_\zeta, \alpha_-) =: \psi, \tag{170}$$

where $\mathbb{1}, m, e, \psi$ are labels of 2+1D toric code anyons (this is a holographic categorical symmetry point of view). We elaborate on all choices of Frobenius algebra $A$ in $\mathrm{Rep}\,\mathbb{Z}_2$, and compute the corresponding $\mathcal{C}_A$, $_A\mathcal{C}_A$ and $[A,A]$. We summarize the result in Table 3.[17]

The whole universal model for $G = \mathbb{Z}_2$ can be summarized by the transverse field Ising model with Hamiltonian

$$H = -\sum_i \sigma_z^i \sigma_z^{i+1} - h \sum_i \sigma_x^i, \tag{171}$$

where the nontrivial global symmetry action is $\prod_i \sigma_x^i$, and $h$ is the field strength. When $|h| < 1$, the model is in the $\mathbb{Z}_2$ SSB phase; when $|h| > 1$, the model is in the $\mathbb{Z}_2$ symmetric phase (trivial $\mathbb{Z}_2$ SPT phase). Below are the explicit calculations:

**Example 6.9.** Let the Frobenius algebra

$$A := \left\langle |1\rangle, |\zeta\rangle \right\rangle \cong \mathrm{Fun}(\mathbb{Z}_2). \tag{172}$$

The multiplication $m : A \otimes A \to A$ in the above basis is (denote $m(a,b)$ by $a \cdot b$)

$$|1\rangle \cdot |1\rangle = |1\rangle, \qquad |\zeta\rangle \cdot |\zeta\rangle = |\zeta\rangle, \qquad |1\rangle \cdot |\zeta\rangle = |\zeta\rangle \cdot |1\rangle = 0. \tag{173}$$

The group action of $\mathbb{Z}_2$ on $A$ is

$$1|1\rangle = |1\rangle, \qquad 1|\zeta\rangle = |\zeta\rangle, \qquad \zeta|1\rangle = |\zeta\rangle, \qquad \zeta|\zeta\rangle = |1\rangle. \tag{174}$$

We have $A \cong \langle |1\rangle + |\zeta\rangle, |1\rangle - |\zeta\rangle \rangle \cong \alpha_+ \oplus \alpha_-$. The embedding $\iota_A$ is trivial, and the universal model is just $(\mathbb{Z}, \mathrm{Fun}(\mathbb{Z}_2), m^\dagger m)$. The Hamiltonian is

$$H = -\sum_i (m^\dagger m)_i = -\sum_i \frac{1}{2}(\sigma_z^i \sigma_z^{i+1} + 1), \tag{175}$$

where each term assigns lower energy on subspace $\langle |1\rangle|1\rangle, |\zeta\rangle|\zeta\rangle \rangle$. It corresponds to the $\mathbb{Z}_2$ SSB phase up to an energy zero point shift.

We depict the $\mathbb{Z}_2$ action on $A$ by the following intuitive diagram:

$$|1\rangle \underset{\zeta}{\overset{\zeta}{\rightleftarrows}} |\zeta\rangle, \tag{176}$$

---

[17]For Abelian group $G$ we have $\mathbf{Vec}_G \cong \mathrm{Rep}\,G$. Here we distinguish them in order to be consistent with cases that $G$ is non-Abelian. Here $\mathbf{Vec}_{\mathbb{Z}_2} \cong \mathrm{Rep}\,\mathbb{Z}_2$, and therefore $Z_1(\mathrm{Rep}\,\mathbb{Z}_2)_{\mathbb{1}\oplus e} \cong Z_1(\mathrm{Rep}\,\mathbb{Z}_2)_{\mathbb{1}\oplus m}$. This is consistent with the fact that exchanging $e$ and $m$ is a braided auto-equivalence of $Z_1(\mathrm{Rep}\,\mathbb{Z}_2)$.

where the general rule and a more nontrivial example is explained in Convention 2

We now compute the simple right $A$-modules and $A$-$A$-bimodules in $\mathrm{Rep}\,\mathbb{Z}_2$. By Remark F.15, we only need to decompose the free (bi)modules. Denote the basis of irreps of $\mathbb{Z}_2$ by

$$\alpha_+ = \langle |+\rangle\rangle, \quad \text{where} \quad 1|+\rangle = \zeta|+\rangle = |+\rangle,$$
$$\alpha_- = \langle |-\rangle\rangle, \quad \text{where} \quad 1|-\rangle = |-\rangle, \quad \zeta|-\rangle = -|-\rangle. \tag{177}$$

We check the free right $A$-modules:

1. $\alpha_+ \otimes A = \langle |+\rangle|1\rangle, |+\rangle|\zeta\rangle \rangle$ is simple. In terms of diagram,

$$|+\rangle|1\rangle \; \underset{\zeta}{\overset{\zeta}{\rightleftarrows}} \; |+\rangle|\zeta\rangle. \tag{178}$$

The nontrivial $A$-module action of $\alpha_+ \otimes A$ is

$$\alpha_+ \otimes A \otimes A \rightarrow \alpha_+ \otimes A,$$
$$|+\rangle|1\rangle \otimes |1\rangle \mapsto |+\rangle|1\rangle,$$
$$|+\rangle|\zeta\rangle \otimes |\zeta\rangle \mapsto |+\rangle|\zeta\rangle. \tag{179}$$

2. $\alpha_- \otimes A = \langle |-\rangle|1\rangle, |-\rangle|\zeta\rangle \rangle$. In terms of diagram,

$$|-\rangle|1\rangle \; \underset{\zeta}{\overset{\zeta}{\rightleftarrows}} \; -|-\rangle|\zeta\rangle. \tag{180}$$

We see $\alpha_- \otimes A \cong \alpha_+ \otimes A$ through the isomorphism $|-\rangle|1\rangle \mapsto |+\rangle|1\rangle$, $-|-\rangle|\zeta\rangle \mapsto |+\rangle|\zeta\rangle$.

We conclude that $(\mathrm{Rep}\,\mathbb{Z}_2)_A \cong \mathbf{Vec}$.

Then we check the free $A$-$A$-bimodules:

1. $A \otimes \alpha_+ \otimes A = \langle |1\rangle|+\rangle|1\rangle, |1\rangle|+\rangle|\zeta\rangle, |\zeta\rangle|+\rangle|1\rangle, |\zeta\rangle|+\rangle|\zeta\rangle \rangle$
   $\cong \langle |1\rangle|+\rangle|1\rangle, |\zeta\rangle|+\rangle|\zeta\rangle \rangle \oplus \langle |1\rangle|+\rangle|\zeta\rangle, |\zeta\rangle|+\rangle|1\rangle \rangle = \bar{1} \oplus \bar{\zeta}$. In terms of diagram,

$$|1\rangle|+\rangle|1\rangle \; \underset{\zeta}{\overset{\zeta}{\rightleftarrows}} \; |\zeta\rangle|+\rangle|\zeta\rangle, \quad |1\rangle|+\rangle|\zeta\rangle \; \underset{\zeta}{\overset{\zeta}{\rightleftarrows}} \; |\zeta\rangle|+\rangle|1\rangle. \tag{181}$$

The nontrivial $A$-$A$-bimodule actions of $\bar{1}$ and $\bar{\zeta}$ are

$$A \otimes \bar{1} \otimes A \rightarrow \bar{1},$$
$$|1\rangle \otimes |1\rangle|+\rangle|1\rangle \otimes |1\rangle \mapsto |1\rangle|+\rangle|1\rangle,$$
$$|\zeta\rangle \otimes |\zeta\rangle|+\rangle|\zeta\rangle \otimes |\zeta\rangle \mapsto |\zeta\rangle|+\rangle|\zeta\rangle, \tag{182}$$

$$A \otimes \bar{\zeta} \otimes A \rightarrow \bar{\zeta},$$
$$|1\rangle \otimes |1\rangle|+\rangle|\zeta\rangle \otimes |\zeta\rangle \mapsto |1\rangle|+\rangle|\zeta\rangle, \tag{183}$$
$$|\zeta\rangle \otimes |\zeta\rangle|+\rangle|1\rangle \otimes |1\rangle \mapsto |\zeta\rangle|+\rangle|1\rangle.$$

Therefore, we see $\bar{1}$ and $\bar{\zeta}$ are not isomorphic, and $A \cong \bar{1}$ as $A$-$A$-bimodules through the isomorphism $|1\rangle|+\rangle|1\rangle \mapsto |1\rangle$, $|\zeta\rangle|+\rangle|\zeta\rangle \mapsto |\zeta\rangle$.

2. $A \otimes \alpha_- \otimes A = \langle |1\rangle|-\rangle|1\rangle, |1\rangle|-\rangle|\zeta\rangle, |\zeta\rangle|-\rangle|1\rangle, |\zeta\rangle|-\rangle|\zeta\rangle \rangle$
   $\cong \langle |1\rangle|-\rangle|1\rangle, |\zeta\rangle|-\rangle|\zeta\rangle \rangle \oplus \langle |1\rangle|-\rangle|\zeta\rangle, |\zeta\rangle|-\rangle|1\rangle \rangle \cong \bar{1} \oplus \bar{\zeta}$. In terms of diagram,

$$|1\rangle|-\rangle|1\rangle \; \underset{\zeta}{\overset{\zeta}{\rightleftarrows}} \; -|\zeta\rangle|-\rangle|\zeta\rangle, \quad |1\rangle|-\rangle|\zeta\rangle \; \underset{\zeta}{\overset{\zeta}{\rightleftarrows}} \; -|\zeta\rangle|-\rangle|1\rangle. \tag{184}$$

Next we compute the relative tensor product $\underset{A}{\otimes}$ (Definition C.5) between $A$-$A$-bimodules which serves as the monoidal structure in $_A\mathcal{C}_A$. Due to the special multiplication of $A$, whenever the $|1\rangle, |\zeta\rangle$ in the middle does not match, the tensor over $A$ is zero, for example, $|1\rangle|+\rangle|1\rangle\underset{A}{\otimes}|\zeta\rangle|+\rangle|1\rangle = 0$. We thus use the abbreviation $|1\rangle|+\rangle|\zeta\rangle|+\rangle|1\rangle := |1\rangle|+\rangle|\zeta\rangle\underset{A}{\otimes}|\zeta\rangle|+\rangle|1\rangle$:

1. $\bar{1}\underset{A}{\otimes}\bar{1} = \langle|1\rangle|+\rangle|1\rangle|+\rangle|1\rangle, |\zeta\rangle|+\rangle|\zeta\rangle|+\rangle|\zeta\rangle\rangle \cong \bar{1}$.

2. $\bar{1}\underset{A}{\otimes}\bar{\zeta} = \langle|1\rangle|+\rangle|1\rangle|+\rangle|\zeta\rangle, |\zeta\rangle|+\rangle|\zeta\rangle|+\rangle|1\rangle\rangle \cong \bar{\zeta}\underset{A}{\otimes}\bar{1} \cong \bar{\zeta}$.

3. $\bar{\zeta}\underset{A}{\otimes}\bar{\zeta} = \langle|1\rangle|+\rangle|\zeta\rangle|+\rangle|1\rangle, |\zeta\rangle|+\rangle|1\rangle|+\rangle|\zeta\rangle\rangle \cong \bar{1}$.

Therefore, we conclude that $_A(\operatorname{Rep}\mathbb{Z}_2)_A \cong \mathbf{Vec}_{\mathbb{Z}_2}$ as fusion categories.

In the end we compute the superconducting quantum currents. In order to compute the internal hom $[A, A]$ by the adjunction (165), we check the left and right $A$ actions on $Q \otimes A$ (Diagram (156)) as an $A$-$A$-bimodule for all simple $Q \in Z_1(\operatorname{Rep}\mathbb{Z}_2)$. The right $A$ actions are all simply the multiplication of $A$. The left $A$ actions for all $Q \otimes A$ are

1. For $m \otimes A = (C_\zeta, \alpha_+) \otimes A$, the half-braiding (recall Eq. (A.15)) is

$$
\begin{aligned}
\beta_{(C_\zeta, \alpha_+), A} : (C_\zeta, \alpha_+) \otimes A &\to A \otimes (C_\zeta, \alpha_+), \\
\zeta \otimes |1\rangle &\mapsto (\zeta|1\rangle) \otimes |\zeta\rangle = |\zeta\rangle \otimes \zeta, \\
\zeta \otimes |\zeta\rangle &\mapsto (\zeta|\zeta\rangle) \otimes |\zeta\rangle = |1\rangle \otimes \zeta.
\end{aligned}
\tag{185}
$$

The left action on $(C_\zeta, \alpha_+) \otimes A$ is

$$
\begin{array}{ccccc}
 & \xrightarrow{\beta^{-1}_{(C_\zeta, \alpha_+), A} \otimes \mathrm{id}_A} & & \xrightarrow{\mathrm{id}_{(C_\zeta, \alpha_+)} \otimes m} & \\
A \otimes (C_\zeta, \alpha_+) \otimes A & & (C_\zeta, \alpha_+) \otimes A \otimes A & & (C_\zeta, \alpha_+) \otimes A, \\
|1\rangle \otimes \zeta \otimes |\zeta\rangle & \longmapsto & \zeta \otimes |\zeta\rangle \otimes |\zeta\rangle & \longmapsto & \zeta \otimes |\zeta\rangle, \\
|\zeta\rangle \otimes \zeta \otimes |1\rangle & \longmapsto & \zeta \otimes |1\rangle \otimes |1\rangle & \longmapsto & \zeta \otimes |1\rangle,
\end{array}
\tag{186}
$$

where all other left actions are zero. We see $(C_\zeta, \alpha_+) \otimes A \cong \bar{\zeta}$ as $A$-$A$-bimodules as they have the same bimodule action.

2. For $e \otimes A = (C_1, \alpha_-) \otimes A$, the half-braiding is

$$
\begin{aligned}
\beta_{(C_1, \alpha_-), A} : (C_1, \alpha_-) \otimes A &\to A \otimes (C_1, \alpha_-), \\
1 \otimes |1\rangle &\mapsto (1|1\rangle) \otimes |1\rangle = |1\rangle \otimes 1, \\
1 \otimes |\zeta\rangle &\mapsto (1|\zeta\rangle) \otimes |1\rangle = |\zeta\rangle \otimes 1.
\end{aligned}
\tag{187}
$$

The left action on $(C_1, \alpha_-) \otimes A$ is

$$
\begin{array}{ccccc}
 & \xrightarrow{\beta^{-1}_{(C_1, \alpha_-), A} \otimes \mathrm{id}_A} & & \xrightarrow{\mathrm{id}_{(C_1, \alpha_-)} \otimes m} & \\
A \otimes (C_1, \alpha_-) \otimes A & & (C_1, \alpha_-) \otimes A \otimes A & & (C_1, \alpha_-) \otimes A, \\
|1\rangle \otimes 1 \otimes |1\rangle & \longmapsto & 1 \otimes |1\rangle \otimes |1\rangle & \longmapsto & 1 \otimes |1\rangle, \\
|\zeta\rangle \otimes 1 \otimes |\zeta\rangle & \longmapsto & 1 \otimes |\zeta\rangle \otimes |\zeta\rangle & \longmapsto & 1 \otimes |\zeta\rangle,
\end{array}
\tag{188}
$$

where all other left actions are zero. We see $(C_1, \alpha_-) \otimes A \cong \bar{1} \cong A$ as $A$-$A$-bimodules.

3. $\mathbb{1} \otimes A = (C_1, \alpha_+) \otimes A \cong \bar{1} \cong A$ as bimodules.

4. $\psi \otimes A = (C_\zeta, \alpha_-) \otimes A \cong \bar{\zeta}$ as bimodules.

Therefore using the adjunction

$$\text{Hom}_{A(\text{Rep}\,\mathbb{Z}_2)_A}(Q \otimes A, A) \cong \text{Hom}_{Z_1(\text{Rep}\,\mathbb{Z}_2)}(Q, [A,A]),\tag{189}$$

we conclude that

$$[A,A] \cong (C_1, \alpha_+) \oplus (C_1, \alpha_-) = \mathbb{1} \oplus e,\tag{190}$$

which corresponds to condensing $\mathbb{Z}_2$ charges.

**Example 6.10.** Let $A = \alpha_+$ the trivial algebra. The multiplication $m$ of $A$ is trivial, and the embedding $\iota_A : A = \alpha_+ \to \text{Fun}(\mathbb{Z}_2) \cong \alpha_+ \oplus \alpha_-$. The universal model is $(\mathbb{Z}, \text{Fun}(\mathbb{Z}_2), -(\iota_A \otimes \iota_A)(\iota_A^\dagger \otimes \iota_A^\dagger))$. The Hamiltonian is

$$
\begin{aligned}
H &= \sum_i -(\iota_A \otimes \iota_A)_i (\iota_A^\dagger \otimes \iota_A^\dagger)_i = -\sum_i (\iota_A \iota_A^\dagger)_i (\iota_A \iota_A^\dagger)_{i+1} \\
&= -\sum_i \frac{(\sigma_x^i + I)}{2} \frac{(\sigma_x^{i+1} + I)}{2} = -\frac{1}{4}\sum_i (1 + 2\sigma_x^i + \sigma_x^i \sigma_x^{i+1}).
\end{aligned}
\tag{191}
$$

It is clear that this Hamiltonian describes the same phase as the polarized Ising model (they share the same ground state and the same classification of excitations):

$$H = -\sum_i \sigma_x^i,\tag{192}$$

and they both correspond to the $\mathbb{Z}_2$ trivial SPT phase.

Moreover, we have $(\text{Rep}\,\mathbb{Z}_2)_A \cong \text{Rep}\,\mathbb{Z}_2$, $_A(\text{Rep}\,\mathbb{Z}_2)_A \cong \text{Rep}\,\mathbb{Z}_2$ and in this case

$$[A,A] \cong (C_1, \alpha_+) \oplus (C_\zeta, \alpha_+) =: \mathbb{1} \oplus m,\tag{193}$$

which corresponds to condensing $\mathbb{Z}_2$ fluxes.

## 6.4 Example $G = S_3$

Let $G = S_3 = \{1, a, b, b^2, ba, b^2a\}$ (we use notations in Example 2.12). Simple objects in $Z_1(\text{Rep}\,S_3)$ are

$$
\begin{aligned}
&(C_1 = \{1\}, \lambda_0), \quad (C_1, \lambda_1), \quad (C_1, \Lambda), \\
&(C_b = \{b, b^2\}, 1), \quad (C_b, \omega), \quad (C_b, \omega^2), \\
&(C_a = \{a, ba, b^2a\}, +), \quad (C_a, -),
\end{aligned}
\tag{194}
$$

where $1, \omega, \omega^2$ denote irreps of $\mathbb{Z}_3$ and $+, -$ denotes irreps of $\mathbb{Z}_2$. We elaborate on Frobenius algebras $A$ in $\text{Rep}\,S_3$, and compute the corresponding $\mathcal{C}_A$, $_A\mathcal{C}_A$ and $[A,A]$. We summarize the result in Table 4. As the second cohomology group of subgroups $(\{1\}, \mathbb{Z}_2, \mathbb{Z}_3, S_3)$ of $S_3$ are all trivial, the four examples exhaust all fixed-point models with $S_3$ symmetry.

Below are the explicit calculations:

**Example 6.11.** Let the Frobenius algebra

$$A := \langle 1 + a, b + ba, b^2 + b^2a \rangle \cong \text{Fun}(S_3/\mathbb{Z}_2),\tag{195}$$

which is a sub Frobenius algebra of $\text{Fun}(S_3)$, and we omit the Dirac notation in the following. For simplicity of later computation, we denote these three basis vectors in $A$ as

$$x := 1 + a, y := b + ba, z := b^2 + b^2a.\tag{196}$$

Table 4: Frobenius algebra $A$, $\mathcal{C}_A$, $_A\mathcal{C}_A$ and Lagrangian algebra $[A,A]$ in $\mathrm{Rep}\,S_3$. Each $H$ is a subgroup of $G$. The ground state subspace is $A$.

| $A = \mathrm{Fun}(G/H)$ | $\mathcal{C}_A \cong \mathrm{Rep}\,H$ | Unbroken symmetry | $_A\mathcal{C}_A$ | $[A,A]$ |
|---|---|---|---|---|
| $\mathrm{Fun}(S_3)$ | **Vec** | None | $\mathbf{Vec}_{S_3}$ | $(C_1,\lambda_0)\oplus(C_1,\lambda_1)\oplus(C_1,\Lambda)\oplus(C_1,\Lambda)$ |
| $\mathrm{Fun}(S_3/\mathbb{Z}_3)$ | $\mathrm{Rep}\,\mathbb{Z}_3$ | $\mathbb{Z}_3$ | $\mathbf{Vec}_{S_3}$ | $(C_1,\lambda_0)\oplus(C_1,\lambda_1)\oplus(C_b,1)\oplus(C_b,1)$ |
| $\mathrm{Fun}(S_3/\mathbb{Z}_2)$ | $\mathrm{Rep}\,\mathbb{Z}_2$ | $\mathbb{Z}_2$ | $\mathrm{Rep}\,S_3$ | $(C_1,\lambda_0)\oplus(C_1,\Lambda)\oplus(C_a,+)$ |
| $\mathrm{Fun}(S_3/S_3)=\lambda_0$ | $\mathrm{Rep}\,S_3$ | $S_3$ | $\mathrm{Rep}\,S_3$ | $(C_1,\lambda_0)\oplus(C_b,1)\oplus(C_a,+)$ |

The multiplication $m : A\otimes A \to A$ in the above basis is

$$x\cdot x = x, \quad y\cdot y = y, \quad z\cdot z = z, \quad x\cdot y = x\cdot z = y\cdot z = y\cdot x = z\cdot x = z\cdot y = 0. \tag{197}$$

The group action of $S_3$ on $A$ is (we list only $a,b$ as $S_3$ is generated by $a,b$)

$$ax = x, \quad ay = z, \quad az = y, \quad bx = y, \quad by = z, \quad bz = x. \tag{198}$$

We have $A \cong \lambda_0 \oplus \Lambda$ as $S_3$ representations[18] and $A$ is cyclic.

**Convention 2.** We introduce the following Cayley-like diagram to represent cyclic representations:

$$a\,\circlearrowright\, x \xrightarrow{\;b\;} y \xrightarrow{\;b\;} z, \tag{199}$$

where

- each node is a vector in the representation;

- each node has outgoing arrows labeled by generators of the group, which is $a,b$ of $S_3$ here;

- the number of outgoing arrows at each node is equal to the number of generators;

- for an arrow labeled by $g$, whose source node is vector $\boldsymbol{n}$, the target node is $g\boldsymbol{n}$.

Denote the bases of irreps of $S_3$ by

$$\begin{aligned}
\lambda_0 &= \langle \boldsymbol{e}\rangle, & \text{where} \quad & a\boldsymbol{e} = \boldsymbol{e} = b\boldsymbol{e}, \\
\lambda_1 &= \langle \boldsymbol{o}\rangle, & \text{where} \quad & a\boldsymbol{o} = -\boldsymbol{o}, b\boldsymbol{o} = \boldsymbol{o}, \\
\Lambda &= \langle \boldsymbol{0},\boldsymbol{1}\rangle, & \text{where} \quad & a\boldsymbol{0} = \boldsymbol{1}, \quad a\boldsymbol{1} = \boldsymbol{0}, \quad b\boldsymbol{0} = \omega\boldsymbol{0}, \quad b\boldsymbol{1} = \omega^2\boldsymbol{1}.
\end{aligned} \tag{200}$$

We check the free right $A$-modules:

1. $\lambda_0 \otimes A$ with basis (we omit tensor product between vectors in the following computations)

$$\boldsymbol{e}x, \boldsymbol{e}y, \boldsymbol{e}z, \tag{201}$$

is simple. This is easy to see from the multiplication of $A$, which is like delta-functions that picks out $x$ or $y,z$. Let $M$ be a non-zero submodule of $\lambda_0 \otimes A$, then there is $w := c_x \boldsymbol{e}x + c_y \boldsymbol{e}y + c_z \boldsymbol{e}z \in M$ where at least one of $c_x, c_y, c_z$ is non-zero. Suppose

---

[18]$A = \langle x,y,z\rangle \cong \langle x+y+z\rangle \oplus \langle x+\omega^2 y+\omega z, x+\omega y+\omega^2 z\rangle = \lambda_0 \oplus \Lambda$.

$c_x$ is non-zero, by multiplying $x$, one finds $w \cdot x = c_x ex \in M$ and thus $ex \in M$. Then $ey = b(ex) \in M$ and $ez = b(ey) \in M$. Thus $M = \lambda_0 \otimes A$.[19]

2. More generally, the special form of the multiplication of $A$ implies that any nonzero submodule of a free module $X \otimes A$ must contain $wx, b(w)y, b^2(w)z$ for some non-zero $w \in X$.

3. $\lambda_1 \otimes A = \langle ox, oy, oz \rangle$ is also simple.

4. $\Lambda \otimes A = \langle 0x, 0y, 0z, 1x, 1y, 1z \rangle$ is isomorphic to $(\lambda_0 \otimes A) \oplus (\lambda_1 \otimes A)$. The symmetric $A$-module maps are as follows:

$$
\begin{aligned}
\lambda_0 \otimes A &\to \Lambda \otimes A, \\
ex &\mapsto (0+1)x, \\
ey = b(ex) &\mapsto (\omega 0 + \omega^2 1)y = b((0+1)x), \\
ez = b(ey) &\mapsto (\omega^2 0 + \omega 1)z = b((\omega 0 + \omega^2 1)y), \\
\lambda_1 \otimes A &\to \Lambda \otimes A, \\
ox &\mapsto (0-1)x, \\
oy = b(ox) &\mapsto (\omega 0 - \omega^2 1)y = b((0-1)x), \\
oz = b(oy) &\mapsto (\omega^2 0 - \omega 1)z = b((\omega 0 - \omega^2 1)y).
\end{aligned}
$$

$$(202)$$

$$(203)$$

The two maps are symmetric, which can be easily seen from the following diagrams:

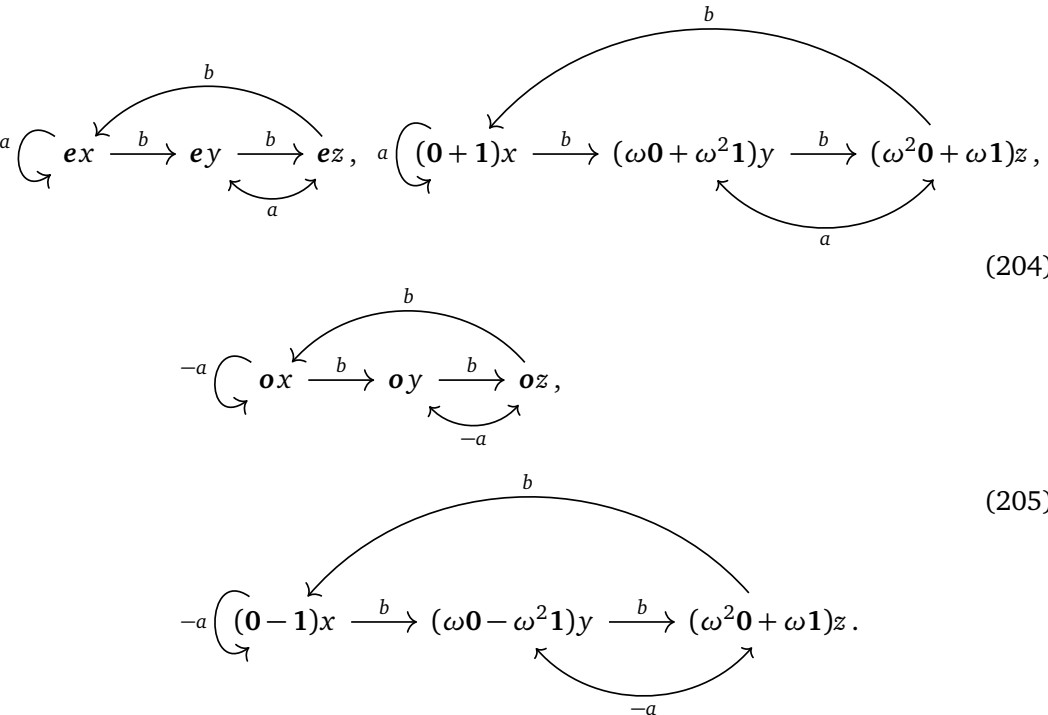

$$(204)$$

$$(205)$$

We conclude that $(\operatorname{Rep} S_3)_A \cong \operatorname{Rep} \mathbb{Z}_2$.

Now we check the free $A$-$A$-bimodules:

---

[19]Or in other words, we cannot find a subspace for $\langle ex, ey, ez \rangle$ such that this subspace is invaraint under both the right $A$-module action (i.e., the multiplication of $A$) and the group action (almost the same group action as in Diagram (199)).

1. $A \otimes \lambda_0 \otimes A$ has 9 basis vectors. As discussed before, $x\boldsymbol{e}x$ must be in a non-zero sub bimodule. We may check the cyclic $S_3$ representation generated by $x\boldsymbol{e}x$:

$$a \circlearrowleft x\boldsymbol{e}x \xrightarrow{\ b\ } y\boldsymbol{e}y \xrightarrow{\ b\ } z\boldsymbol{e}z. \qquad (206)$$

This sub $A$-$A$-bimodule is simple and isomorphic to $A$. It will be denoted by $\bar{\lambda}_0$ (we will soon see why this notation is reasonable). Then we check the cyclic $S_3$ representation generated by $x\boldsymbol{e}y$:

$$\begin{array}{c}
x\boldsymbol{e}y \xleftrightarrow{\ a\ } x\boldsymbol{e}z \\
\downarrow b \qquad\qquad \uparrow b \\
y\boldsymbol{e}z \xleftrightarrow{\ a\ } z\boldsymbol{e}y \\
\downarrow b \qquad\qquad \uparrow b \\
z\boldsymbol{e}x \xleftrightarrow{\ a\ } y\boldsymbol{e}x
\end{array} \qquad (207)$$

This sub bimodule is also simple and will be denoted by $\bar{\Lambda}$. Thus $A \otimes \lambda_0 \otimes A \cong \langle x\boldsymbol{e}x, y\boldsymbol{e}y, z\boldsymbol{e}z \rangle \oplus \langle x\boldsymbol{e}y, x\boldsymbol{e}z, y\boldsymbol{e}z, z\boldsymbol{e}y, z\boldsymbol{e}x, y\boldsymbol{e}x \rangle = \bar{\lambda}_0 \oplus \bar{\Lambda}$.

2. $A \otimes \lambda_1 \otimes A$ has 9 basis vectors. The cyclic $S_3$ representation generated by $x\boldsymbol{o}x$ is

$$-a \circlearrowleft x\boldsymbol{o}x \xrightarrow{\ b\ } y\boldsymbol{o}y \xrightarrow{\ b\ } z\boldsymbol{o}z. \qquad (208)$$

It is simple and will be denoted by $\bar{\lambda}_1$. The cyclic $S_3$ representation generated by $x\boldsymbol{o}y$ is:

$$\begin{array}{c}
x\boldsymbol{o}y \xleftrightarrow{\ a\ } -x\boldsymbol{o}z \\
\downarrow b \qquad\qquad \uparrow b \\
y\boldsymbol{o}z \xleftrightarrow{\ a\ } -z\boldsymbol{o}y \\
\downarrow b \qquad\qquad \uparrow b \\
z\boldsymbol{o}x \xleftrightarrow{\ a\ } -y\boldsymbol{o}x
\end{array} \qquad (209)$$

It is isomorphic to $\bar{\Lambda}$ (just identify vectors at corresponding nodes in the two diagrams). Thus $A \otimes \lambda_1 \otimes A \cong \bar{\lambda}_1 \oplus \bar{\Lambda}$.

3. $A \otimes \Lambda \otimes A \cong A \otimes ((\lambda_0 \otimes A) \oplus (\lambda_1 \otimes A)) \cong (A \otimes \lambda_0 \otimes A) \oplus (A \otimes \lambda_1 \otimes A) \cong \bar{\lambda}_0 \oplus \bar{\Lambda} \oplus \bar{\lambda}_1 \oplus \bar{\Lambda}$.

Next we compute the monoidal structure $\underset{A}{\otimes}$ in $_A(\mathrm{Rep}\,S_3)_A$. Again, due to the special multiplication of $A$, whenever the $x, y, z$ in the middle does not match, the tensor over $A$ is zero, for example, $x\boldsymbol{o}y \underset{A}{\otimes} z\boldsymbol{e}y = 0$. We thus use the abbreviation $x\boldsymbol{o}y\boldsymbol{e}z := x\boldsymbol{o}y \underset{A}{\otimes} y\boldsymbol{e}z$. Then we can easily compute:

1. $\bar{\lambda}_0 \underset{A}{\otimes} \bar{\lambda}_0 = \langle x\boldsymbol{e}x\boldsymbol{e}x, y\boldsymbol{e}y\boldsymbol{e}y, z\boldsymbol{e}z\boldsymbol{e}z \rangle \cong \bar{\lambda}_0$.

2. $\bar{\lambda}_0 \underset{A}{\otimes} \bar{\lambda}_1 = \langle x\boldsymbol{e}x\boldsymbol{o}x, y\boldsymbol{e}y\boldsymbol{o}y, z\boldsymbol{e}z\boldsymbol{o}z \rangle \cong \bar{\lambda}_1$.

3. $\bar{\lambda}_1 \underset{A}{\otimes} \bar{\lambda}_0 = \langle xoxex, yoyey, zozez \rangle \cong \bar{\lambda}_1$.

4. $\bar{\lambda}_1 \underset{A}{\otimes} \bar{\lambda}_1 = \langle xoxox, yoyoy, zozoz \rangle \cong \bar{\lambda}_0$.

5. $\bar{\Lambda} \cong \bar{\Lambda} \underset{A}{\otimes} \bar{\lambda}_0 \cong \bar{\Lambda} \underset{A}{\otimes} \bar{\lambda}_1 \cong \bar{\lambda}_0 \underset{A}{\otimes} \bar{\Lambda} \cong \bar{\lambda}_1 \underset{A}{\otimes} \bar{\Lambda}$.

6. $\bar{\Lambda} \underset{A}{\otimes} \bar{\Lambda} \cong \bar{\lambda}_0 \oplus \bar{\lambda}_1 \oplus \bar{\Lambda}$, where as sub bimodules of $\bar{\Lambda} \underset{A}{\otimes} \bar{\Lambda}$,

$$
\begin{aligned}
\langle xe(y+z)ex, ye(z+x)ey, ze(x+y)ez \rangle &\cong \bar{\lambda}_0, \\
\langle xe(y-z)ex, ye(z-x)ey, ze(x-y)ez \rangle &\cong \bar{\lambda}_1, \\
\langle xeyez, xezey, yezex, yexez, zexey, zeyex \rangle &\cong \bar{\Lambda}. \quad (210)
\end{aligned}
$$

Therefore, we conclude that $_A(\mathrm{Rep}\, S_3)_A \cong \mathrm{Rep}\, S_3$ as fusion categories.

In the end we compute the superconducting quantum currents. As an example we compute the bimodule $(C_b, 1) \otimes A$. The basis vectors are

$$
b \otimes x, b \otimes y, b \otimes z, b^2 \otimes x, b^2 \otimes y, b^2 \otimes z. \quad (211)
$$

The right $A$ action is easy. To compute the left $A$ action (the first diagram in Diagram (156)), note that the half-braiding is just the group action on $A$ (recall Eq. (A.15)):

$$
\begin{aligned}
\beta : (C_b, 1) \otimes A &\to A \otimes (C_b, 1), \\
b \otimes x &\mapsto bx \otimes b = y \otimes b, \quad (212) \\
&\cdots
\end{aligned}
$$

Thus the left action is

$$
\begin{aligned}
y \cdot (b \otimes x) &= bx \cdot (b \otimes x) = b \otimes (x \cdot x) = b \otimes x, \\
z \cdot (b \otimes y) &= b \otimes y, \quad x \cdot (b \otimes z) = b \otimes z, \\
z \cdot (b^2 \otimes x) = b^2 \otimes x, \quad x \cdot (b^2 \otimes y) &= b^2 \otimes y, \quad y \cdot (b^2 \otimes z) = b^2 \otimes z, \quad (213)
\end{aligned}
$$

while all others are zero. We can then identify, for example, $b \otimes x$ with $yex$, just by checking the two-sided action of $A$. It is then easy to see $(C_b, 1) \otimes A \cong \bar{\Lambda}$ as bimodules. We can similarly compute

$$
\begin{aligned}
(C_1, \lambda_0) \otimes A &\cong \bar{\lambda}_0 \cong A, \\
(C_1, \lambda_1) \otimes A &\cong \bar{\lambda}_1, \\
(C_1, \Lambda) \otimes A &\cong A \oplus \bar{\lambda}_1, \\
(C_b, 1) \otimes A \cong (C_b, \omega) \otimes A \cong (C_b, \omega^2) \otimes A &\cong \bar{\Lambda}, \\
(C_a, +) \otimes A &\cong A \oplus \bar{\Lambda}, \\
(C_a, -) \otimes A &\cong \bar{\lambda}_1 \oplus \bar{\Lambda}. \quad (214)
\end{aligned}
$$

Therefore using the adjunction

$$
\mathrm{Hom}_{A(\mathrm{Rep}\, S_3)_A}(Q \otimes A, A) \cong \mathrm{Hom}_{Z_1(\mathrm{Rep}\, S_3)}(Q, [A, A]), \quad (215)
$$

we conclude that

$$
[A, A] \cong (C_1, \lambda_0) \oplus (C_1, \Lambda) \oplus (C_a, +). \quad (216)
$$

**Example 6.12.** Let $A = \lambda_0$ the trivial algebra, we have $_A(\text{Rep}\, S_3)_A \cong \text{Rep}\, S_3$ and in this case

$$[A,A] \cong (C_1, \lambda_0) \oplus (C_b, 1) \oplus (C_a, +), \tag{217}$$

which corresponds to condensing pure $S_3$ fluxes. It is not just a coincidence that exchanging $(C_1, \Lambda)$ and $(C_b, 1)$ in the Lagrangian algebra leads to the same category of excitations; in fact, exchanging $(C_1, \Lambda)$ and $(C_b, 1)$ is moreover a braided auto-equivalence of $Z_1(\text{Rep}\, S_3)$.

**Example 6.13.** Let $A = \text{Fun}(S_3)$, we have $_A(\text{Rep}\, S_3)_A \cong \mathbf{Vec}_{S_3}$ and in this case

$$[A,A] \cong (C_1, \lambda_0) \oplus (C_1, \lambda_1) \oplus (C_1, \Lambda) \oplus (C_1, \Lambda) \cong \text{Fun}(S_3), \tag{218}$$

which corresponds to condensing all $S_3$ charges.

**Example 6.14.** Let $A = \text{Fun}(S_3/\mathbb{Z}_3)$, we have $_A(\text{Rep}\, S_3)_A \cong \mathbf{Vec}_{S_3}$ and in this case

$$[A,A] \cong (C_1, \lambda_0) \oplus (C_1, \lambda_1) \oplus (C_b, 1) \oplus (C_b, 1). \tag{219}$$

Note that the two Lagrangian algebra for $\text{Fun}(S_3)$ and $\text{Fun}(S_3/\mathbb{Z}_3, \mathbb{C})$ again differ by exchanging $(C_1, \Lambda)$ and $(C_b, 1)$.

**Remark 6.15.** If we begin with $\mathcal{C} = \mathbf{Vec}_{S_3}$ and follow the same algorithm to compute the fixed-point models, we will obtain four models whose emergent symmetries are in one-to-one correspondence with those obtained from $\text{Rep}\, S_3$. Usually we think $\text{Rep}\, S_3$ as the category of symmetry charges while $\mathbf{Vec}_{S_3}$ as the category of symmetry defects. It turns out symmetry charge becomes a relative notion; we can equally consider $\mathbf{Vec}_{S_3}$ as the symmetry charges and correspondingly $\text{Rep}\, S_3$ as the symmetry defects. The two perspectives lead to the same classification of fixed-point models (or gapped phases). Choosing the category of symmetry charges is like choosing an inertial frame of reference.

# 7 Conclusion

In this paper, we established the general formulation for quantum currents. Given the category $\mathcal{C}$ whose objects are symmetry charges and morphisms are symmetric operators, we showed that quantum currents can be identified with the Drinfeld center $Z_1(\mathcal{C})$ of $\mathcal{C}$.

We also gave a rigorous analysis on the renormalization process and fixed points in 1+1D. We showed that the fixed-points correspond to Frobenius algebras in $\mathcal{C}$, and in turn Lagrangian algebras in $Z_1(\mathcal{C})$. From the quantum current point of view, it is the superconducting of quantum currents that determine the fixed-points. Since fixed-points represent phases of matter, the superconducting of quantum currents also determines gapped phases.

The Frobenius algebra fixed-point model is constructed by symmetric operators in $\mathcal{C}$, so, *a priori*, it has the symmetry $\mathcal{C}$. But in the end, the symmetry $\mathcal{C}$ is (partially) spontaneously broken; a new emergent symmetry (the category of excitations) is observed, which turns out to be the category of bimodules over the Frobenius algebra. Quantum currents provide an invariant for all gapped phases arising in this way. Mathematically, the fusion categories of excitations in these phases are Morita equivalent. Physically, these phases share the same holographic categorical symmetry; the holographic categorical symmetry remains the same upon spontaneous symmetry breaking.

Let's collect all relevant notions regarding the phases sharing the same holographic categorical symmetry $Z_1(\mathcal{C})$ for a global view. We begin with one of them exhibiting the category of excitations as $\mathcal{C}$. First, there is 2-category $\text{Alg}(\mathcal{C})$ whose objects are Frobenius algebras in $\mathcal{C}$, 1-morphisms are bimodules in $\mathcal{C}$ and 2-morphims are bimodule maps. Second, there is a 2-category $_{\mathcal{C}}\mathbf{2Vec}$ whose objects are left $\mathcal{C}$-module categories in $\mathbf{2Vec}$, 1-morphisms are $\mathcal{C}$-module

Table 5: Ingredients of 1+1D holographic categorical symmetry $Z_1(\mathcal{C})$.

| $\Sigma\mathcal{C}$ | $\mathrm{Alg}(\mathcal{C})$ | $_\mathcal{C}\mathbf{2Vec}$ |
|---|---|---|
| objects | Frobenius algebras (fixed-point Hamiltonians and ground states) | $\mathcal{C}$-modules (all fixed-point boundary conditions) |
| 1-morphisms | bimodules (fixed-point defects) | $\mathcal{C}$-module functors |
| 2-morphisms | bimodule maps (boundary/defect change) | natural transformations |

functors and 2-morphisms are $\mathcal{C}$-module natural transformations. These two 2-categories are equivalent under the following identification

$$\mathrm{Alg}(\mathcal{C}) \to {}_\mathcal{C}\mathbf{2Vec}, \tag{220}$$
$$A \mapsto \mathcal{C}_A,$$
$$_AM_B \mapsto - \underset{A}{\otimes} M.$$

Thus we introduce the notation $\Sigma\mathcal{C} \cong \mathrm{Alg}(\mathcal{C}) \cong {}_\mathcal{C}\mathbf{2Vec}$, known as the *delooping* or *condensation completion* [4,12,17,41] of $\mathcal{C}$. Moreover, $\Sigma\mathcal{C}$ does not depend on the beginning choice: for any $\mathcal{D}$ that is Morita equivalent to $\mathcal{C}$, $\Sigma\mathcal{D} \cong \Sigma\mathcal{C}$ (this is in fact an alternative definition of Morita equivalence). Physically, this result indicates that Morita-equivalent $\mathcal{D}$ and $\mathcal{C}$ should be viewed on equal footing; we can equally call objects in $\mathcal{D}$ as symmetry charges and consider $\mathcal{C}$ as a (generalized) SSB phase of $\mathcal{D}$. The collection of all generalized SSB phases $\Sigma\mathcal{C} \cong \Sigma\mathcal{D}$ does not depend on which we call as symmetry charges.

Note that the physical interpretations of the two realizations of $\Sigma\mathcal{C}$ are not exactly the same. The invariant $Z_1(\mathcal{C})$ of Morita equivalence is given by [13]

$$Z_1(\mathcal{C}) \cong \Omega Z_0(\Sigma\mathcal{C}) := \mathrm{Hom}_{\mathrm{Fun}(\Sigma\mathcal{C},\Sigma\mathcal{C})}(\mathrm{id}_{\Sigma\mathcal{C}}, \mathrm{id}_{\Sigma\mathcal{C}}). \tag{221}$$

We conclude these notions (together with the physical interpretations in parenthesis), in Table 5.

Note that in 1+1D, we can only talk about the *total* charge transported between subsystems. In higher dimensions, however, the charge distribution, as well as current density, is also of interest. Moreover, for higher symmetries, there can be extended charged object of intrinsic higher dimensions [4].

Our formulation for quantum current can be generalized to higher dimensions by replacing (1-)category with higher category. However, since a computable model of higher category is not available at the moment, detailed calculation is not possible. We just sketch the abstract formulation here:

1. A higher symmetry in $n+1$D is described by a fusion $n$-category $\mathcal{C}$, whose objects, 1-morphisms, ..., $n-1$ morphisms are symmetry charges of codimension 1, 2, ...,$n$, respectively, and $n$-morphisms are symmetric operators. The symmetry charges of various dimensions should automatically encode the information of, for example, charge density, as well as intrinsic higher dimensional charges.

2. The quantum currents are identified with the (higher analog of) Drinfeld center $Z_1(\mathcal{C})$. It should be a modular $n$-category and should encode the symmetry charge transport along various codimensions.

3. Fixed-point models are determined by higher analogs of Frobenius algebras in $\mathcal{C}$.

4. The quantum currents superconducting in the fixed-point models form higher analogs of Lagrangian algebras in $Z_1(\mathcal{C})$.

5. Spontaneous symmetry breaking does not change the holographic categorical symmetry or the quantum currents. Gapped phases in $n + 1$D connected to $\mathcal{C}$ by higher analog of spontaneous (higher) symmetry breaking, should be Morita equivalent to each other, and share the same holographic categorical symmetry.[20] They together form an $n + 1$-category $\Sigma\mathcal{C}$, the condensation completion of $\mathcal{C}$.

## Acknowledgments

**Funding information** This work is supported by start-up funding and Direct Grant No. 4053501 from The Chinese University of Hong Kong, and by funding from Hong Kong Research Grants Council (ECS No. 24304722).

## A  Drinfeld center of $\text{Rep}\,G$

It is well known that the Drinfeld center $Z_1(\text{Rep}\,G)$ of $\text{Rep}\,G$ is equivalent to the category of representations of the quantum double of $G$. The simple objects of $Z_1(\text{Rep}\,G)$ are given by pairs $(C_x, \tau : N_x \to GL(V))$, where

1. $x \in G$ is a group element;

2. $C_x = \{b \in G | \exists g \in G, b = gxg^{-1}\}$ is the conjugacy class containing $x$;

3. $N_x = \{b \in G | bx = xb\}$ is the centralizer subgroup of $x$ in $G$;

4. $V$ is a vector space, and $(V, \tau)$ is an irreducible representation of $N_x$.

A simple object $(C_x, \tau)$ also carries a representation of $G$. To describe such representation, we need to make some auxiliary choices. First, it is clear that $G/N_x \cong C_x$. Let $\{z_i\}_{z_i \in i, i \in G/N_x}$ be a chosen set of representatives of left cosets (Definition D.1). Then for any group element $h \in G$, there is a unique pair $i \in G/N_x, h' \in N_x$ such that $h = z_i h'$.

Now we form a vector space $C(C_x) \otimes V$ and define the group action of $G$ on it by

$$g \triangleright (z_i x z_i^{-1} \otimes y) = g(z_i x z_i^{-1})g^{-1} \otimes \tau_h(y) = z_k x z_k^{-1} \otimes \tau_{z_k^{-1} g z_i}(y), \tag{A.1}$$

where $z_k, h$ are determined by the unique coset decomposition of $g z_i$, i.e., $g z_i = z_k h, h \in N_x$. It is not hard to check that different of choices of $z_i$ lead to isomorphic $G$-representations on $C(C_x) \otimes V$.

**Example A.1.** Let $G = S_3$ (Example 2.12). For example we write down the set of left cosets $S_3/\{1, a\}$:

$$S_3/\{1, a\} = \{\{1, a\}, \{b, ab^2\}, \{b^2, ab\}\}, \tag{A.2}$$

where we can choose the representatives of these three left cosets to be $a, ab^2, ab$ respectively. Thereby, any group element in $S_3$ can be expressed by a representative in $\{a, ab^2, ab\}$ multiply with an element in subset $\{1, a\}$.

---

[20]Note that in 1+1D (for fusion 1-categories), $\mathcal{C}$ is Morita equivalent to $\mathcal{D}$ if and only if $Z_1(\mathcal{C}) \cong Z_1(\mathcal{D})$. In higher dimensions (for fusion 2-categories or higher), if $\mathcal{C}$ is Morita equivalent to $\mathcal{D}$, then $Z_1(\mathcal{C}) \cong Z_1(\mathcal{D})$; however, the converse is not true.

Next, we will recover the above results by directly solving the half-braiding conditions. Given object $(Q, \beta_{Q,-}) \in Z_1(\mathrm{Rep}\,G)$, it suffices to check the half-braiding $\beta_{Q,R}$ between $Q$ and $R = C(G)$, the regular representation (Definition D.3), as $R$ is "universal" in $\mathrm{Rep}\,G$: $R$ contains all possible irreducible representations. Our convention for group actions on $R$ is by left multiplication: $g \triangleright h = gh$. It is easy to check that the intertwiners between $R$ and $R$ itself is $\mathrm{Hom}(R, R) \cong C(G)$, by right multiplication. We will write for $w \in C(G)$, $r_w \in \mathrm{Hom}(R, R)$,

$$r_w : R \to R,$$
$$h \mapsto r_w(h) = hw. \tag{A.3}$$

Moreover, we have intertwiners

$$m_y : R \otimes R \to R,$$
$$g \otimes h \mapsto \delta_{gy,h}\, g, \tag{A.4}$$
$$\Delta_y : R \to R \otimes R,$$
$$g \mapsto g \otimes gy, \tag{A.5}$$

for each $y \in G$. They satisfy

$$m_{y_1}\Delta_{y_2} = \delta_{y_1,y_2}\mathrm{id}_R, \qquad \sum_{y \in G} \Delta_y m_y = \mathrm{id}_{R \otimes R}, \tag{A.6}$$

by which we know that $R \otimes R$ is the direct sum of $|G|$ copies of $R$. With these preparations, we now examine the form of $Q$ and $\beta_{Q,R}$. Firstly, since $\beta_{Q,R}$ is symmetric and natural in the $R$ component, we have for any $a \in Q$,

$$\begin{aligned}
\beta_{Q,R}(ga \otimes h) &= \beta_{Q,R}(g(a \otimes g^{-1}h)) \\
&= \beta_{Q,R}(\mathrm{id}_Q \otimes r_{g^{-1}h}(g(a \otimes e))) \\
&= (r_{g^{-1}h} \otimes \mathrm{id}_Q)g\beta_{Q,R}(a \otimes e). \tag{A.7}
\end{aligned}$$

Second, we want to prove that $Q$ is graded by $G$. Consider the following linear map for each $h \in G$,

$$P_h : Q \to Q,$$
$$a \mapsto (\delta_h \otimes \mathrm{id}_Q)\beta_{Q,R}(a \otimes e), \tag{A.8}$$

where $\delta_h : R \to \mathbb{C}$ is defined by $\delta_h(g) = \delta_{h,g}$. We have

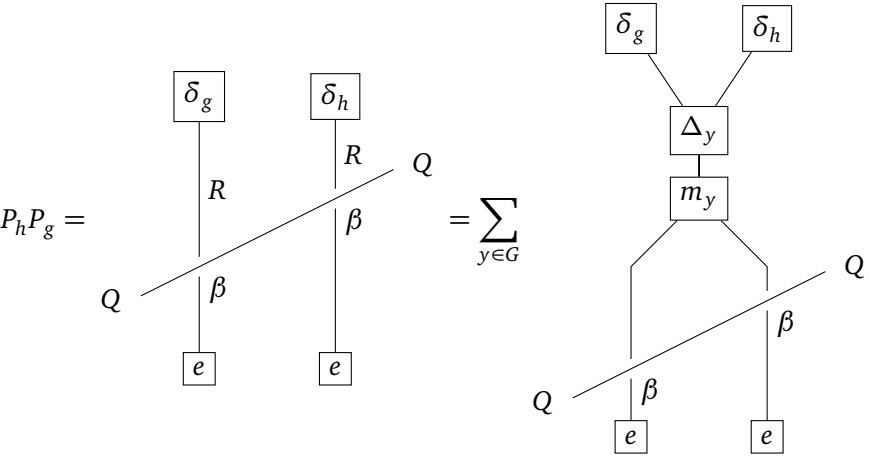

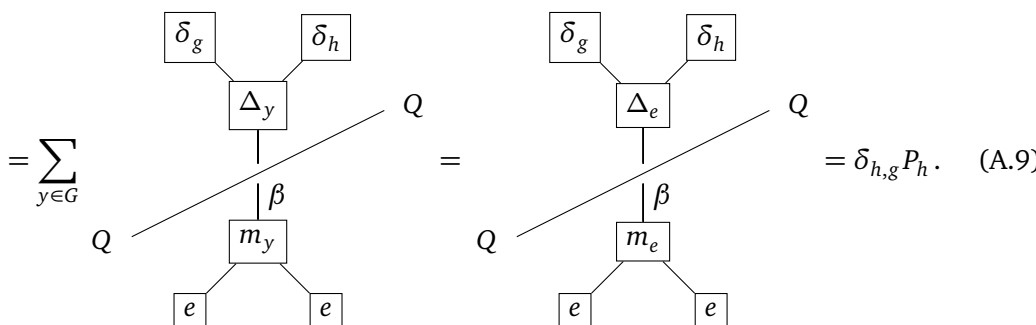

$$= \sum_{y \in G} \qquad = \qquad = \delta_{h,g} P_h . \quad \text{(A.9)}$$

Thus we conclude that $P_h$ are mutually orthogonal projections. Also using the fact that $\sum_{h \in G} \delta_h$ is an intertwiner, we have $\sum_h P_h = \mathrm{id}_Q$. Therefore, $Q = \oplus_{h \in G} P_h Q$, i.e., $Q$ is graded by $G$.

Now we focus on the subspace $P_x Q$. Let $a \in P_x Q$, i.e., $(\delta_x \otimes \mathrm{id}_Q) \beta_{Q,R}(a \otimes e) = a$ or $\beta_{Q,R}(a \otimes e) = x \otimes a$. We have

$$\beta_{Q,R}(ga \otimes e) = (r_{g^{-1}} \otimes \mathrm{id}_Q) g \beta_{Q,R}(a \otimes e) = gxg^{-1} \otimes ga, \quad \text{(A.10)}$$

which means that $ga \in P_{gxg^{-1}}Q$. It is then clear that $P_x Q$ carries a representation of $N_x$. We make the choice $z_i$ for representatives of $G/N_x$ as before. $P_{z_i x z_i^{-1}}Q$ carries a representation of $N_{z_i x z_i^{-1}}$. $N_{z_i x z_i^{-1}}$ is isomorphic to $N_x$ via

$$N_x \to N_{z_i x z_i^{-1}},$$
$$h \mapsto z_i h z_i^{-1}. \quad \text{(A.11)}$$

The representation carried by $P_{z_i x z_i^{-1}}Q$ is also is isomorphic to that carried by $P_x Q$. To see this, suppose $b \in P_{z_i x z_i^{-1}}Q$, then $z_i^{-1} b \in P_x$. For $h \in N_x$ we have

$$h z_i^{-1} b = z_i^{-1}(z_i h z_i^{-1}) b. \quad \text{(A.12)}$$

This equation means that action by $z_i^{-1}$ is an intertwiner from $P_{z_i x z_i^{-1}}$ to $P_x$, which is clearly invertible. Moreover, for a general $g \in G$, $g z_i = z_k h$, $h \in N_x$ and $b \in N_{z_i x z_i^{-1}}, z_i^{-1} b \in N_x$ we have

$$g b = z_k z_k^{-1} g z_i (z_i^{-1} b) = z_k h (z_i^{-1} b) \in P_{z_k x z_k^{-1}}Q, \quad \text{(A.13)}$$

which agrees with Eq. (A.1).

We conclude that a general object $(Q, \beta_{Q,-})$ in the Drinfeld center $Z_1(\mathrm{Rep}\, G)$ is a representation $Q \in \mathrm{Rep}\, G$ whose underlying vector space is graded by $G$, such that for $a \in Q$ graded by $x \in G$,

1. $ga$ is graded by $gxg^{-1}$.

2. For the regular representation $R = C(G)$, $h \in R$,

$$\beta_{Q,R}(a \otimes h) = xh \otimes a. \quad \text{(A.14)}$$

   For a general representation $(V, \rho)$, $b \in V$,

$$\beta_{Q,V}(a \otimes b) = \rho_x(b) \otimes a. \quad \text{(A.15)}$$

A simple object in $Z_1(\mathrm{Rep}\, G)$ is then given by restricting the grading by $G$ to only a conjugacy class, and the representation of $N_x$ carried by the $x$-graded subspace to an irreducible one. One can check that for a simple object $(X, \beta_{X,-}) \in Z_1(\mathrm{Rep}\, G)$, $X$ is a cyclic representation of $G$.

# B  Some basic concepts in category theory

**Definition B.1** (Module category)**.** Let $(\mathcal{C}, \otimes, \mathbf{1}, \alpha, \lambda, \rho)$ be a monoidal category, where $\mathbf{1}$ is the tensor unit, $\alpha, \lambda, \rho$ are natural isomorphisms called associator, left unitor and right unitor respectively. A left $\mathcal{C}$-module is a category $\mathcal{M}$ equipped with a monoidal functor $\mathcal{C} \to \mathrm{Fun}(\mathcal{M}, \mathcal{M})$, or equivalently, an action functor $\rhd \colon \mathcal{C} \times \mathcal{M} \to \mathcal{M}$ equipped with

- An associator: a natural transformation $\alpha \colon (- \otimes -) \rhd - \to - \rhd (- \rhd -)$ such that $\forall X, Y, Z \in \mathcal{C}, M \in \mathcal{M}$,

$$
\begin{array}{ccc}
((X \otimes Y) \otimes Z) \rhd M & \xrightarrow{\ \alpha_{X \otimes Y, Z, M}\ } & (X \otimes Y) \rhd (Z \rhd M) \\
{\scriptstyle \alpha_{X,Y,Z} \rhd \mathrm{id}_M} \downarrow & & \downarrow {\scriptstyle \alpha_{X,Y,Z \rhd M}} \\
(X \otimes (Y \otimes Z)) \rhd M \xrightarrow{\alpha_{X,Y \otimes Z, M}} X \rhd ((Y \otimes Z) \rhd M) \xrightarrow{\alpha_{Y,Z,M}} X \rhd (Y \rhd (Z \rhd M))
\end{array}
\tag{B.1}
$$

- A unitor: a natural transformation $\mu \colon \mathbf{1} \rhd - \to \mathrm{id}_{\mathcal{M}}$ such that $\forall X \in \mathcal{C}, M \in \mathcal{M}$,

$$
\begin{array}{ccc}
(\mathbf{1} \otimes X) \rhd M & \xrightarrow{\ \alpha_{\mathbf{1},X,m}\ } & \mathbf{1} \rhd (X \rhd M) \\
 & {\scriptstyle \lambda_X \rhd \mathrm{id}_M} \searrow \quad \swarrow {\scriptstyle \mu_{X \rhd M}} & \\
 & X \rhd M &
\end{array}
\tag{B.2}
$$

$$
\begin{array}{ccc}
(X \otimes \mathbf{1}) \rhd M & \xrightarrow{\ \alpha_{X,\mathbf{1},m}\ } & X \rhd (\mathbf{1} \rhd M) \\
 & {\scriptstyle \rho_X \rhd \mathrm{id}_M} \searrow \quad \swarrow {\scriptstyle \mathrm{id}_X \rhd \mu_M} & \\
 & X \rhd M &
\end{array}
\tag{B.3}
$$

A right $\mathcal{C}$-module is defined similarly.

**Definition B.2** (Enriched category)**.** Let $(\mathcal{C}, \otimes, \mathbf{1})$ be a monoidal category. An $\mathcal{C}$-enriched category $^{\mathcal{C}}\mathcal{M}$ consists of

- Objects: $\mathrm{Ob}(^{\mathcal{C}}\mathcal{M})$;

- Hom-objects: $^{\mathcal{C}}\mathcal{M}(X, Y)$ in $\mathcal{C}$ for every pair $X, Y \in {}^{\mathcal{C}}\mathcal{M}$;

- Unit morphisms: $1_X \colon \mathbf{1} \to {}^{\mathcal{C}}\mathcal{M}(X, X)$ in $\mathcal{C}$ for every $X \in {}^{\mathcal{C}}\mathcal{M}$;

- Composition of morphisms: $^{\mathcal{C}}\mathcal{M}(Y, Z) \otimes {}^{\mathcal{C}}\mathcal{M}(X, Y) \xrightarrow{\circ} {}^{\mathcal{C}}\mathcal{M}(X, Z)$ in $\mathcal{C}$ for every triple $X, Y, Z \in {}^{\mathcal{C}}\mathcal{M}$;

such that $\forall X, Y, Z, W \in {}^{\mathcal{C}}\mathcal{M}$, the following diagrams commute:

$$
\begin{array}{ccc}
{}^{\mathcal{C}}\mathcal{M}(Y,Z) \otimes {}^{\mathcal{C}}\mathcal{M}(X,Y) \otimes {}^{\mathcal{C}}\mathcal{M}(W,X) & \xrightarrow{\ \mathrm{id}\otimes\circ\ } & {}^{\mathcal{C}}\mathcal{M}(Y,Z) \otimes {}^{\mathcal{C}}\mathcal{M}(W,Y) \\
{\scriptstyle \circ\otimes\mathrm{id}} \downarrow & & \downarrow {\scriptstyle \circ} \\
{}^{\mathcal{C}}\mathcal{M}(X,Z) \otimes {}^{\mathcal{C}}\mathcal{M}(W,Z) & \xrightarrow{\quad\quad\circ\quad\quad} & {}^{\mathcal{C}}\mathcal{M}(W,Z)
\end{array}
\tag{B.4}
$$

$$
\begin{array}{ccccc}
\mathbf{1} \otimes {}^{\mathcal{C}}\mathcal{M}(X,Y) & \xrightarrow{\ \cong\ } & {}^{\mathcal{C}}\mathcal{M}(X,Y) & \xleftarrow{\ \cong\ } & {}^{\mathcal{C}}\mathcal{M}(X,Y) \otimes \mathbf{1} \\
{\scriptstyle 1_Y \otimes \mathrm{id}} \downarrow & \nearrow {\scriptstyle \circ} & & \nwarrow {\scriptstyle \circ} & \downarrow {\scriptstyle \mathrm{id} \otimes 1_X} \\
{}^{\mathcal{C}}\mathcal{M}(Y,Y) \otimes {}^{\mathcal{C}}\mathcal{M}(X,Y) & & & & {}^{\mathcal{C}}\mathcal{M}(X,Y) \otimes {}^{\mathcal{C}}\mathcal{M}(X,X)
\end{array}
\tag{B.5}
$$

$\mathcal{C}$ is called the *background category* of $^{\mathcal{C}}\mathcal{M}$.

**Definition B.3** (Internal Hom)**.** Let $\mathcal{A}$ be a monoidal category and $\mathcal{L}$ be a left $\mathcal{C}$-module. Given $M, N \in \mathcal{M}$, if the functor $\mathrm{Hom}_{\mathcal{M}}(- \rhd M.N) : \mathcal{C}^{\mathrm{op}} \to \mathbf{Vec}$ is representable and the representing object in $\mathcal{C}$ is denoted as $[M, N]$, i.e., there is a natural isomorphism

$$\mathrm{Hom}_{\mathcal{M}}(- \rhd M, N) \cong \mathrm{Hom}_{\mathcal{C}}(-, [M, N]), \tag{B.6}$$

then $[M, N]$ is called the *internal hom*. And $\mathcal{M}$ is called *enriched in* $\mathcal{C}$ if the internal hom $[M, N]$ exists in $\mathcal{C}$ for all $M, N \in \mathcal{M}$.

**Proposition B.4.** If $\mathcal{M}$ is enriched in $\mathcal{C}$, $\mathcal{M}$ is promoted to a $\mathcal{C}$-enriched category $^{\mathcal{C}}\mathcal{M}$. (Details please refer to Ref. [27]) This is called the *canonical construction* of enriched category.

**Definition B.5** (**Vec**$_G$)**.** The category of $G$-graded vector spaces is a unitary fusion category consists of

- Objects: Finite-dimensional $G$-graded vector spaces $\{V := \oplus_{g \in G} V_g\}$, where $V_g$ is the sub vector space of $V$ graded by $g$.

- Morphisms: $\mathrm{Hom}_{\mathbf{Vec}_G}(V, W) := \{f \in \mathrm{Hom}_{\mathbf{Vec}}(\oplus_{g \in G} V_g, \oplus_{h \in G} W_h) | f(V_g) \subset W_g\}$.

- Tensor product: $(V \otimes W)_g = \oplus_{ab=g} V_a \otimes W_b$.

- Tensor unit: One dimensional vector space graded by $e \in G$ the identity element.

A simple object is a one-dimensional vector space graded by $g \in G$. One-dimensional vector spaces are isomorphic if and only if they are graded by the same $g$.

**Definition B.6** (Monomorphism)**.** In category $\mathcal{C}$, a morphism $f : X \to Y$ is called a monomorphism if it is left-cancellative, i.e., $\forall Z \in \mathcal{C}$ and $\forall g_1, g_2 : Z \to X$,

$$f g_1 = f g_2 \Rightarrow g_1 = g_2. \tag{B.7}$$

**Definition B.7** (Epimorphism)**.** In category $\mathcal{C}$, a morphism $f : X \to Y$ is called an epimorphism if it is right-cancellative, i.e., $\forall Z \in \mathcal{C}$ and $\forall g_1, g_2 : Y \to Z$,

$$g_1 f = g_2 f \Rightarrow g_1 = g_2. \tag{B.8}$$

**Definition B.8** (Image)**.** Let $f : X \to Y$ be a morphism in $\mathcal{C}$. The image of $f$ is an object $\mathrm{Im} f \in \mathcal{C}$ together with a monomorphism $j : \mathrm{Im} f \to Y$ satisfying

- There exists a morphism $i : X \to \mathrm{Im} f$ such that $f = ji$, called a *factorization* of $f$.

- For any triple $(I', i' : X \to I', j' : I' \to Y)$ where $j'$ is a monomorphism and $f = j'i'$, there exists a unique morphism $v : \mathrm{Im} f \to I'$ such that $j = j'v$.

The universal property of kernel can be depicted by the following commutative diagram,

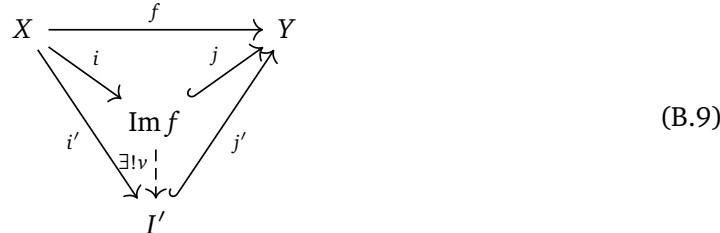

$$\tag{B.9}$$

If $\mathrm{Im} f$ exists, it is unique up to a unique isomorphism, and the following defined (co)equalizer and (co)kernel all have this property.

**Definition B.9** (Coequalizer)**.** Let $f, g : X \to Y$ be a pair of morphisms in $\mathcal{C}$. The coequalizer of $f, g$ is an object $C$ together with a morphism $c : Y \to C$ such that

- $cf = cg$.

- For any pair $(C', c' : Y \to C')$ such that $c'f = c'g$, there exists a unique morphism $\gamma : C \to C'$ such that $c' = \gamma c$.

In terms of diagram,

$$
X \underset{g}{\overset{f}{\rightrightarrows}} Y \xrightarrow{c} C \\
\underset{c'}{\searrow} \quad \downarrow \exists! \gamma \\
C'
\tag{B.10}
$$

Denote the coequalizer of $f$ and $g$ as $\mathrm{coeq}(f, g)$. The equalizer $\mathrm{eq}(f, g)$ is similarly defined.

**Example B.10.** In the category of sets denoted as **Set**, a coequalizer of two maps $f, g : X \to Y$ is the quotient set $Y / \sim$, where the equivalence relation $\sim$ is generated by $f(x) \sim g(x), \forall x \in X$.

Below we always assume $\mathcal{C}$ to be an additive category.

**Definition B.11** (Kernel)**.** Let $\mathcal{C}$ be an additive category and $f : X \to Y$ is a morphism in $\mathcal{C}$. The kernel of $f$ is an object $\ker f$ together with a morphism $k : \ker f \to X$ such that

- $fk = 0$.

- For any pair $(K', k' : K' \to X)$ such that $fk' = 0$, there exists a unique morphism $l : K' \to \ker f$ such that $kl = k'$.

In terms of diagram,

$$
\ker f \xrightarrow{k} X \xrightarrow{f} Y \\
\exists! l \uparrow \quad \nearrow k' \\
K'
\tag{B.11}
$$

**Definition B.12** (Cokernel)**.** Let $f : X \to Y$ be a morphism in $\mathcal{C}$. The cokernel of $f$ is an object $\mathrm{coker} f$ together with a morphism $c : Y \to \mathrm{coker} f$ such that

- $cf = 0$.

- For any pair $(C', c' : Y \to C')$ such that $c'f = 0$, there exists a unique morphism $l : \mathrm{coker} f \to C'$ such that $lc = c'$.

In terms of diagram,

$$
X \xrightarrow{f} Y \xrightarrow{c} \mathrm{coker} f \\
\underset{c'}{\searrow} \quad \downarrow \exists! l \\
C'
\tag{B.12}
$$

**Example B.13.** Given $f, g : X \to Y$ in $\mathcal{C}$, $\mathrm{eq}(f, g) = \ker(f - g)$, $\mathrm{coeq}(f, g) = \mathrm{coker}(f - g)$.

**Remark B.14.** Let $f : X \to Y$ be a morphism in an abelian category. $f$ is a monomorphism if and only if $\ker f = 0$, and $f$ is an epimorphism if and only if $\mathrm{coker} f = 0$.

# C  Algebra and module category

**Definition C.1** (Algebra)**.** Let $\mathcal{C}$ be a monoidal category. An (associative unital) algebra in $\mathcal{C}$ is a triple $(A, m, \eta)$, which is an object $A \in \mathcal{C}$ together with a multiplication $m : A \otimes A \to A$ and a unit morphism $\eta : \mathbf{1} \to A$ satisfying associativity and identity:

$$
\begin{array}{ccc}
(A \otimes A) \otimes A & \xrightarrow{\ \alpha_{A,A,A}\ } & A \otimes (A \otimes A) \\
{\scriptstyle m \otimes \mathrm{id}_A}\downarrow & & \downarrow{\scriptstyle \mathrm{id}_A \otimes m} \\
A \otimes A & & A \otimes A \\
& {\scriptstyle m}\searrow \quad \swarrow{\scriptstyle m} & \\
& A &
\end{array}
\tag{C.1}
$$

$$
\begin{array}{ccc}
\mathbf{1} \otimes A & & A \otimes \mathbf{1} \\
{\scriptstyle \eta \otimes \mathrm{id}_A}\downarrow \ {\scriptstyle \mathrm{id}_A}\searrow & & {\scriptstyle \mathrm{id}_A}\swarrow \ \downarrow{\scriptstyle \mathrm{id}_A \otimes \eta} \\
A \otimes A \xrightarrow{\ m\ } & A & \xleftarrow{\ m\ } A \otimes A
\end{array}
\tag{C.2}
$$

where $\mathrm{id}_A : A \to A$ is the unique identity map on $A$.

**Definition C.2** (Algebra homomorphism)**.** Given two algebras $(A_1, m_1, \eta_1)$ and $(A_2, m_2, \eta_2)$ in $\mathcal{C}$, an algebra homomorphism between them is a morphism $f : A_1 \to A_2$ such that

$$
\begin{array}{ccc}
A_1 \otimes A_1 & \xrightarrow{\ f \otimes f\ } & A_2 \otimes A_2 \\
{\scriptstyle m_1}\downarrow & & \downarrow{\scriptstyle m_2} \\
A_1 & \xrightarrow{\ f\ } & A_2
\end{array}
\quad , \quad
\begin{array}{ccc}
A_1 & \xrightarrow{\ f\ } & A_2 \\
{\scriptstyle \eta_1}\nwarrow & & \nearrow{\scriptstyle \eta_2} \\
& \mathbf{1} &
\end{array}
\tag{C.3}
$$

**Definition C.3** (Module over an algebra)**.** Given an algebra $(A, m, \eta)$ in $\mathcal{C}$, a right $A$-module is a pair $(M, \rho)$, where $M \in \mathcal{C}$ and $\rho : M \otimes A \to M$ is an action morphism such that

$$
\begin{array}{ccc}
(M \otimes A) \otimes A & \xrightarrow{\ \alpha_{M,A,A}\ } & M \otimes (A \otimes A) \\
{\scriptstyle \rho \otimes \mathrm{id}_A}\downarrow & & \downarrow{\scriptstyle \mathrm{id}_M \otimes m} \\
M \otimes A & & M \otimes A \\
& {\scriptstyle \rho}\searrow \quad \swarrow{\scriptstyle \rho} & \\
& M &
\end{array}
\quad , \quad
\begin{array}{ccc}
& M \otimes \mathbf{1} & \\
{\scriptstyle \mathrm{id}_M \otimes \eta}\downarrow & & \searrow{\scriptstyle \mathrm{id}_M} \\
M \otimes A & \xrightarrow{\ \rho\ } & M
\end{array}
\tag{C.4}
$$

A left $A$-module is defined similarly.

**Definition C.4** (Bimodule)**.** Given two algebras $A$ and $B$ in $\mathcal{C}$, an $A$-$B$-bimodule is a triple $(M, \lambda, \rho)$ such that

- $(M, \lambda)$ is a left $A$-module and $(M, \rho)$ right $B$-module.

- The diagram commutes:

$$
\begin{array}{ccc}
A \otimes M \otimes B & \xrightarrow{\ \lambda \otimes \mathrm{id}_B\ } & M \otimes B \\
{\scriptstyle \mathrm{id}_A \otimes \rho}\downarrow & & \downarrow{\scriptstyle \rho} \\
A \otimes M & \xrightarrow{\ \lambda\ } & M
\end{array}
\tag{C.5}
$$

**Definition C.5** (Relative tensor product). Given an algebra $A$ in $\mathcal{C}$, a right $A$-module $(M,\rho)$ and a left $A$-module $(N,\lambda)$, the relative tensor product of $M$ and $N$ over $A$, denoted by $M \underset{A}{\otimes} N$, is the coequalizer of $\rho \otimes \mathrm{id}_N$ and $\mathrm{id}_M \otimes \lambda$:

$$M \otimes A \otimes N \underset{\mathrm{id}_M \otimes \lambda}{\overset{\rho \otimes \mathrm{id}_N}{\rightrightarrows}} M \otimes N \longrightarrow M \underset{A}{\otimes} N. \tag{C.6}$$

**Proposition C.6.** For a right $A$-module $M$ and left $A$-module $N$, $M \underset{A}{\otimes} A \cong M$ and $A \underset{A}{\otimes} N \cong N$.

*Proof.* We show that $(M,\rho)$ is the coequalizer:

$$M \otimes A \otimes A \underset{\mathrm{id}_M \otimes m}{\overset{\rho \otimes \mathrm{id}_A}{\rightrightarrows}} M \otimes A \overset{\rho}{\longrightarrow} M. \tag{C.7}$$

First, from Diagram (C.4), we have $\rho(\rho \otimes \mathrm{id}_A) = \rho(\mathrm{id}_M \otimes m)$. Second, for any pair $(X, f : M \otimes A \to X)$ such that $f(\rho \otimes \mathrm{id}_A) = f(\mathrm{id}_M \otimes m)$, there exists a unique morphism $\gamma = f(\mathrm{id}_M \otimes \eta)$ such that $f = \gamma\rho$. These are two conditions in Definition (B.9), and by the fact that a coequalizer, if exists, is unique up to a unique isomorphism, we conclude that $M \underset{A}{\otimes} A \cong M$. The other part is similar.

$\square$

**Definition C.7** (Module map). Given two $A$-modules $(M,\rho)$ and $(N,\tau)$, a morphism $f : M \to N$ between them is called a left $A$-module map if

$$\begin{array}{ccc} M \otimes A & \overset{f \otimes \mathrm{id}_A}{\longrightarrow} & N \otimes A \\ \rho \downarrow & & \downarrow \tau \\ M & \underset{f}{\longrightarrow} & N \end{array} \tag{C.8}$$

A right $A$-module map is defined similarly. If an isomorphism is a module map, it is automatically an isomorphism between modules.

**Definition C.8** (Bimodule map). Given two $A$-$B$-bimodules $M$ and $N$, a morphism $f : M \to N$ is called an $A$-$B$-bimodule map if it is both a left $A$-module map and a right $B$-module map.

**Definition C.9** (Category of modules over an algebra). Given an algebra $A$ in $\mathcal{C}$, the category of right $A$-modules $\mathcal{C}_A$ consists of:

- Objects: Right A-modules.

- Morphisms: $A$-module maps.

The category of left $A$-modules denoted as $_A\mathcal{C}$ is defined similarly. And given another algebra $B$ in $\mathcal{C}$, the category of $A$-$B$-bimodules denoted as $_A\mathcal{C}_B$ consists of $A$-$B$-bimodules as objects and $A$-$B$-bimodule maps as morphisms.

**Remark C.10.** Given a monoidal category $\mathcal{C}$ and a algebra $A$ in $\mathcal{C}$, the category of right $A$-modules $\mathcal{C}_A$ is a left $\mathcal{C}$-module category. Explicitly, the module action functor is defined by

$$\rhd : \mathcal{C} \times \mathcal{C}_A \to \mathcal{C}_A,$$
$$(X, (M,\rho)) \mapsto \left( X \otimes M, (X \otimes M) \otimes A \overset{\alpha_{X,M,A}}{\longrightarrow} X \otimes (M \otimes A) \overset{\mathrm{id}_X \otimes \rho}{\longrightarrow} X \otimes M \right). \tag{C.9}$$

Then we can check that $\mathcal{C}_A$ together with functor $\rhd$, associator $(X \otimes Y) \otimes M \to X \otimes (Y \otimes M)$ and unitor $\mathbf{1} \otimes M \to M$ satisfies diagrams (B.1) and (B.3), and thus is a left $\mathcal{C}$-module.

**Definition C.11** (Group algebra)**.** Denote the category of finite dimensional vector spaces as **Vec**. A group algebra is a triple $(\mathbb{C}[G], m, \eta)$ in **Vec**, where $G$ is a basis of $\mathbb{C}[G]$, the multiplication is defined as

$$m : \mathbb{C}[G] \times \mathbb{C}[G] \to \mathbb{C}[G],$$

$$\left( \sum_{g \in G} a_g g, \sum_{h \in G} b_h h \right) \mapsto \sum_{gh} a_g b_h gh, \tag{C.10}$$

and the unit morphism is

$$\eta : \mathbb{C} \to \mathbb{C}[G],$$
$$1 \mapsto e, \tag{C.11}$$

where $e \in G$ is the identity element.

**Remark C.12.** The category of all left $\mathbb{C}[G]$-modules $_{\mathbb{C}[G]}\textbf{Vec}$ is exactly $\mathrm{Rep}\, G$.

**Remark C.13.** For any algebra $A \in \mathcal{C}$, for each $X \in \mathcal{C}$ we can construct a free $A$-module through the $\mathcal{C}$-module functor ($\mathcal{C}$ itself is a $\mathcal{C}$-module category)

$$\mathrm{Free} : \mathcal{C} \to \mathcal{C}_A,$$

$$X \mapsto (X \otimes A, (X \otimes A) \otimes A \xrightarrow{\alpha_{X,A,A}} X \otimes (A \otimes A) \xrightarrow{\mathrm{id}_X \otimes m} X \otimes A). \tag{C.12}$$

There is also a forgetful $\mathcal{C}$-module functor $\mathrm{Forg} : \mathcal{C}_A \to \mathcal{C}$, $\mathrm{Forg}(M, \rho) = M$. These two functors are adjoints to each other, i.e.,

$$\mathrm{Hom}_{\mathcal{C}_A}(\mathrm{Free}(X), (M, \rho)) \cong \mathrm{Hom}_{\mathcal{C}}(X, \mathrm{Forg}(M, \rho)). \tag{C.13}$$

It is also a common notation to suppress the $A$-action, by writing $- \otimes A := \mathrm{Free}$,

$$\mathrm{Hom}_{\mathcal{C}_A}(X \otimes A, M) \cong \mathrm{Hom}_{\mathcal{C}}(X, M). \tag{C.14}$$

**Definition C.14** (Algebra Morita equivalence)**.** Two algebras $A, B$ in $\mathcal{C}$ are called Morita equivalent if $\mathcal{C}_A$ and $\mathcal{C}_B$ are equivalent $\mathcal{C}$-module categories.

# D  Some concepts in group representation theory

**Definition D.1** (Coset and quotient group)**.** Let $H$ be a subgroup of group $G$. A left coset of $H$ in $G$ is a set

$$gH = \{gh | h \in H\}, \quad \text{for some} \quad g \in G, \tag{D.1}$$

where $g$ is called a representative of coset $gH$.

And the set of all left cosets of $H$ in $G$ is denoted as $G/H$, i.e.,

$$G/H = \{gH | g \in G\}. \tag{D.2}$$

It is a group when $H$ is a normal subgroup $gHg^{-1} = H, \forall g \in G$, and called a quotient group. Right cosets are similarly defined.

**Definition D.2** (Cyclic representation of an algebra in **Vec**)**.** Given an algebra $A$ in **Vec** and a representation of $A$ (an $A$-module) denoted as $(V, \rho)$, $v \in V$ is called a cyclic vector if $Av = V$. And $(V, \rho)$ is called a cyclic representation if there exists a cyclic vector in $V$.

**Definition D.3** (Regular representation of a group)**.** A (left) regular representation of a group $G$ is an object $(C(G), \rho) \in \text{Rep } G$, where the group action of each $g$ is defined as

$$\rho_g : C(G) \to C(G),$$
$$h \mapsto \rho_g(h) = gh. \tag{D.3}$$

**Remark D.4.** A cyclic representation of group algebra $\mathbb{C}[G]$ in **Vec** is a cyclic representation in $\text{Rep } G$. The dimension of a cyclic representation in $\text{Rep } G$ cannot be larger than the number of group elements in $G$.

**Proposition D.5.** Given an irrep $X_a \in \text{Rep } G$, the dual representation $X_a^*$ is isomorphic to $X_a$ if there exists a trivial representation $\mathbf{1}$ in the direct sum decomposition of $X_a \otimes X_a$, i.e.,

$$X_a \otimes X_a \cong \mathbf{1} \oplus \dots \tag{D.4}$$

*Proof.* By Schur's lemma, we have

$$\mathbb{C} \cong \text{Hom}(X_a, X_a) \cong \text{Hom}(X_a \otimes X_a^*, \mathbf{1}) \cong \text{Hom}(X_a^*, X_a^*), \tag{D.5}$$

which means that $X_a^*$ is also an irrep. From isomorphism (D.4),

$$\text{Hom}(X_a, X_a^*) \cong \text{Hom}(X_a \otimes X_a, \mathbf{1}) \neq 0. \tag{D.6}$$

Again by Schur's lemma, the two irreps are isomorphic $X_a \cong X_a^*$. □

# E  Module functor

**Definition E.1** (Module functor)**.** Let $\mathcal{C}$ be a monoidal category and $\mathcal{M}, \mathcal{N}$ be two left $\mathcal{C}$-modules with associator $\alpha$ amd $\alpha'$, a $\mathcal{C}$-module functor is a functor $F : \mathcal{M} \to \mathcal{N}$ equipped with a natural isomorphism

$$s_{X,M} : F(X \otimes M) \to X \otimes F(M), \quad \forall X \in \mathcal{C}, \, M \in \mathcal{M}, \tag{E.1}$$

such that $\forall X, Y \in \mathcal{C}, M \in \mathcal{M}$,

$$
\begin{array}{ccccc}
F(X \rhd (Y \rhd M)) & \xleftarrow{F(\alpha_{X,Y,M})} & F((X \otimes Y) \rhd M) & \xrightarrow{s_{X \otimes Y, M}} & (X \otimes Y) \rhd F(M) \\
{\scriptstyle s_{X, Y \otimes M}} \downarrow & & & & \downarrow {\scriptstyle \alpha'_{X,Y,F(M)}} \\
X \rhd F(Y \rhd M) & & \xrightarrow{\quad \text{id}_X \otimes s_{Y,M} \quad} & & X \rhd (Y \rhd F(M))
\end{array}
\tag{E.2}
$$

and

$$
\begin{array}{ccc}
F(\mathbf{1} \rhd M) & \xrightarrow{s_{\mathbf{1},M}} & \mathbf{1} \rhd F(M) \\
{\scriptstyle F(\mu_M)} \searrow & & \swarrow {\scriptstyle \mu_{F(M)}} \\
& F(M) &
\end{array}
\tag{E.3}
$$

**Definition E.2** (Module natural transformation)**.** Let $(F, s)$ and $(G, t)$ be two $\mathcal{C}$-module functors. A module natural transformation between them is a natural transformation $v : F \Rightarrow G$ such that $\forall X \in \mathcal{C}, M \in \mathcal{M}$,

$$
\begin{array}{ccc}
F(X \rhd M) & \xrightarrow{s_{X,M}} & X \rhd F(M) \\
{\scriptstyle v_{X \otimes M}} \downarrow & & \downarrow {\scriptstyle \text{id}_X \rhd v_M} \\
G(X \rhd M) & \xrightarrow{t_{X,M}} & X \rhd G(M)
\end{array}
\tag{E.4}
$$

**Remark E.3.** Let $\mathcal{M}, \mathcal{N}$ be two left $\mathcal{C}$-modules. We denote by $\mathrm{Fun}_{\mathcal{C}}(\mathcal{M}, \mathcal{N})$ the category of left $\mathcal{C}$-module functors from $\mathcal{M}$ to $\mathcal{N}$ and module natural transformations.

**Remark E.4.** Let $\mathcal{C}_A$ and $\mathcal{C}_B$ be the categories of right $A$-modules and $B$-modules for some algebras $A, B$ in $\mathcal{C}$. The category of module functors $\mathrm{Fun}_{\mathcal{C}}(\mathcal{C}_A, \mathcal{C}_B)$ is equivalent to the category of $A$-$B$-bimodules ${}_A\mathcal{C}_B$ via the functor

$$
\begin{aligned}
{}_A\mathcal{C}_B &\to \mathrm{Fun}_{\mathcal{C}}(\mathcal{C}_A, \mathcal{C}_B), \\
M &\mapsto - \underset{A}{\otimes} M.
\end{aligned}
\tag{E.5}
$$

# F  Frobenius algebra, separable algebra and Lagrangian algebra

**Definition F.1** (Coalgebra). A (unital associative) coalgebra in a monoidal category $\mathcal{C}$ is a triple $(C, \Delta, \epsilon)$, which is an object $C \in \mathcal{C}$ together with a comultiplication $\Delta : C \to C \otimes C$ and a counit morphism $\epsilon : C \to \mathbf{1}$ satisfying coassociativity and coidentity:

$$
\begin{array}{ccc}
(C \otimes C) \otimes C & \xrightarrow{\ \alpha_{C,C,C}\ } & C \otimes (C \otimes C) \\
{\scriptstyle \Delta \otimes \mathrm{id}_C} \Big\uparrow & & \Big\uparrow {\scriptstyle \mathrm{id}_C \otimes \Delta} \\
C \otimes C & & C \otimes C \\
& {\scriptstyle \Delta} \nwarrow \quad \nearrow {\scriptstyle \Delta} & \\
& C &
\end{array}
\tag{F.1}
$$

$$
\begin{array}{ccccc}
\mathbf{1} \otimes C & & & & C \otimes \mathbf{1} \\
{\scriptstyle \epsilon \otimes \mathrm{id}_C} \Big\uparrow & {\scriptstyle \lambda_C} \searrow & & {\scriptstyle \rho_C} \swarrow & \Big\uparrow {\scriptstyle \mathrm{id}_C \otimes \epsilon} \\
C \otimes C & \xleftarrow[\ \Delta\ ]{} & C & \xrightarrow[\ \Delta\ ]{} & C \otimes C
\end{array}
\tag{F.2}
$$

**Definition F.2** (Frobenius algebra). A Frobenius algebra in $\mathcal{C}$ is a tuple $(A, m, \eta, \Delta, \epsilon)$ satisfying

- $(A, m, \eta)$ is an algebra and $(A, \Delta, \epsilon)$ is a coalgebra.

- The Frobenius condition:

$$
(\mathrm{id}_A \otimes m)\alpha_{A,A,A}(\Delta \otimes \mathrm{id}_A) = \Delta m = (m \otimes \mathrm{id}_A)\alpha_{A,A,A}^{-1}(\mathrm{id}_A \otimes \Delta).
\tag{F.3}
$$

**Definition F.3** (Isometric algebra). Given a unitary fusion category (UFC) $\mathcal{C}$, an algebra $(A, m, \eta)$ in $\mathcal{C}$ is called isometric if $m m^\dagger = \mathrm{id}_A$ (the $\dagger$ strcture is from the UFC $\mathcal{C}$).

**Remark F.4.** By Theorem 5.8, an isometric algebra $(A, m, \eta)$ is a Frobenius algebra by taking $\Delta = m^\dagger$, $\epsilon = \eta^\dagger$.

**Remark F.5.** Given a algebra $(A, m, \eta)$ in a UFC $\mathcal{C}$, $m^\dagger$ is an $A$-$A$-bimodule map if and only if it satisfies the Frobenius condition (F.3).

**Definition F.6** (Algebra of function on a group). The algebra of $\mathbb{C}$-valued function on a group $G$ is a triple $(\mathrm{Fun}(G), \delta, \varepsilon)$ in $\mathrm{Rep}\, G$, where the multiplication is defined as

$$
\begin{aligned}
\delta : \mathrm{Fun}(G) \times \mathrm{Fun}(G) &\to \mathrm{Fun}(G), \\
\left( \sum_g a_g \delta_g, \sum_k b_k \delta_k \right) &\mapsto \sum_g a_g b_g \delta_g,
\end{aligned}
\tag{F.4}
$$

where $a_g, b_g \in \mathbb{C}$, $\delta_g$ denotes the delta function $\delta_g(h) = \delta_{g,h}$. The function $\sum_g a_g \delta_g$ is thus

$$\sum_g a_g \delta_g(h) = a_h. \tag{F.5}$$

In calculations we may abuse the notation, drop the $\delta$ symbol and write $g \equiv \delta_g$ as basis vectors of Fun($G$). In this sense, the coefficients of the *formal* linear combination $\sum_g a_g g$ and the functions on $G$ determines each other. The unit morphism is

$$\varepsilon : \mathbb{C} \to \text{Fun}(G),$$
$$1 \mapsto \sum_g g, \tag{F.6}$$

or the function $\varepsilon(g) = 1$ for any $g$.

**Remark F.7.** We have $\text{Rep}\,G_{\text{Fun}(G)} \cong \mathbf{Vec}$, $\mathbf{Vec}_{\text{Fun}(G)} \cong \mathbf{Vec}_G$, $_{\text{Fun}(G)}\text{Rep}\,G_{\text{Fun}(G)} \cong {}_{\text{Fun}(G)}\mathbf{Vec} \cong \mathbf{Vec}_G$.

**Proposition F.8.** Algebras in $\text{Rep}\,G$ are classified by (up to Morita equivalence) by $(H \subset G, \omega_2)$ [28], where $\omega_2 \in H^2(H, U(1))$ is a 2-cocycle.

**Example F.9.** For the case of trivial $\omega_2$, we consider the set of algebras $\{(\text{Fun}(G/H), \delta, \varepsilon) | H \subset G\}$, where $\text{Fun}(G/H)$ is the $\mathbb{C}$-valued function of cosets $G/H$, and the multiplication is defined as

$$\delta : \text{Fun}(G/H) \otimes \text{Fun}(G/H) \to \text{Fun}(G/H),$$
$$\left( \sum_{gH} a_{gH} gH, \sum_{kH} b_{kH} kH \right) \mapsto \sum_{gH,kH} \delta_{gH,kH} a_{gH} b_{kH} gH = \sum_{gH} a_{gH} b_{gH} gH. \tag{F.7}$$

Here we abuse the notation $gH$ for the delta function which is 1 on the coset $gH$ and 0 on other cosets.

**Example F.10.** Given a subgroup $H \subset G$, one algebra in the Morita class $(H, \omega_2 \in H^2(H, U(1)))$ can be realized as a sub-representation of $\text{Fun}(G) \otimes \text{Fun}(G)$,

$$\langle x \otimes y, x \in G, y \in G, x^{-1}y \in H \rangle, \tag{F.8}$$

with multiplication

$$(x \otimes y) \cdot (w \otimes z) = \frac{1}{\sqrt{|H|}} \delta_{yw} \omega_2(x^{-1}y, y^{-1}z) x \otimes z. \tag{F.9}$$

**Remark F.11.** The algebras in Example F.9 and Example F.10 are all isometric Frobenius algebras.

**Proposition F.12.** For any Frobenius algebra $A = \text{Fun}(G/H)$ in $\text{Rep}\,G$, $\text{Rep}\,G_A$ is equivalent to $\text{Rep}\,H$ [8].

**Definition F.13** (Separable algebra). An algebra $(A, m, \eta)$ in a monoidal category $\mathcal{C}$ is called separable if there is an $A$-$A$-bimodule map $e : A \to A \otimes A$ such that $me = \text{id}_A$.

**Proposition F.14.** Let $A$ be an isometric algebra in a unitary fusion category $\mathcal{C}$. Any $A$-module is a direct summand of some free module. Moreover, $\mathcal{C}_A$ is both semisimple and unitary. Similar results also hold for $A$-$A'$-bimodules.

*Proof.* Since $m^\dagger$ is automatically an $A$-$A$-bimodule map, an isometric algebra $(A, m, \eta)$ is automatically separable, i.e., we have

$$A \otimes A \xrightleftharpoons[e]{m} A, \tag{F.10}$$

such that $me = \mathrm{id}_A$ where $e = m^\dagger$. Recall Definition 2.2, $A \otimes A \cong A \oplus \ker m$. In other words, $A$ is a direct summand of $A \otimes A$, both as $A$-$A$-bimodules.

The remaining proof is essentially as in [35] Proposition 2.7; we elaborate more on the details for the reader's convenience. Given any right $A$-module $M$, $M \cong M \underset{A}{\otimes} A$ is a direct summand of the free module $M \otimes A \cong M \underset{A}{\otimes} (A \otimes A)$, where

$$M \underset{A}{\otimes} (A \otimes A) \cong M \underset{A}{\otimes} (A \oplus \ker m) \cong (M \underset{A}{\otimes} A) \oplus (M \underset{A}{\otimes} \ker m). \tag{F.11}$$

Semisimpleness of an abelian category is equivalent to that any object is projective, i.e., $\mathrm{Hom}(X, -)$ preserves colimits for all $X$. Since $\mathcal{C}$ is semisimple, $\mathrm{Hom}_{\mathcal{C}_A}(M \otimes A, -) \cong \mathrm{Hom}_{\mathcal{C}}(M, -)$ preserves colimits, i.e., all free modules are projective. We have shown in the above that any $A$-module is a direct summand of a free module, and is thus also projective. Therefore, $\mathcal{C}_A$ is semisimple. The proof for ${}_A\mathcal{C}'_A$ is similar.

The unitary structure is inherited from $\mathcal{C}$: given an $A$-module map $f : M \otimes A \to N \otimes A$, since the action on free modules are all partially isometric, $f^\dagger$ is automatically an $A$-module map due to Theorem 5.20. Since any $A$-module is a direct summand of some free module, the unitary structure is induced. $\square$

**Remark F.15.** Thus to find all simple $A$-modules, we only need to decompose the free modules $i \otimes A$ for all simples $i$. Similarly, given any bimodule $B$, $B \cong A \underset{A}{\otimes} B \underset{A'}{\otimes} A'$ is a direct summand of the free bimodule $A \otimes B \otimes A' \cong (A \otimes A) \underset{A}{\otimes} B \underset{A'}{\otimes} (A' \otimes A')$. Thus all simple bimodules can be found by decomposing the free bimodules $A \otimes i \otimes A'$ for all simples $i$.

**Definition F.16** (Commutative algebra). An algebra $(A, m, \eta)$ in a braided monoidal category $\mathcal{B}$ is called commutative if

$$
\begin{array}{ccc}
A \otimes A & \xrightarrow{\ \beta_{A,A}\ } & A \otimes A \\
& {\scriptstyle m}\searrow \quad \swarrow {\scriptstyle m} & \\
& A &
\end{array}
\tag{F.12}
$$

where $\beta$ is the braiding isomorphism in $\mathcal{B}$.

Similarly, a coalgebra $(C, \Delta, \epsilon)$ in $\mathcal{B}$ is called co-commutative if $\beta_{C,C}\Delta = \Delta$.

**Definition F.17** (Local module). Given an algebra $A$ in a braided category $\mathcal{B}$, a right $A$-module $(M, \rho)$ is called local if

$$
\begin{array}{ccc}
M \otimes A & \xrightarrow{\ \rho\ } & M \\
{\scriptstyle \beta_{M,A}}\downarrow & & \uparrow{\scriptstyle \rho} \\
A \otimes M & \xrightarrow[\ \beta_{A,M}\ ]{} & M \otimes A
\end{array}
\tag{F.13}
$$

We denote by $\mathcal{C}_A^{\mathrm{loc}}$ the full subcategory of $\mathcal{C}$ consisting of local right $A$-modules.

**Definition F.18** (Lagrangian algebra). A commutative algebra $A_L$ in a braided category $\mathcal{B}$ is called a Lagrangian algebra if any local $A_L$-module in $\mathcal{B}$ is a direct sum of copies of $A_L$.

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
