# Peer review of "Quantum Current and Holographic Categorical Symmetry"

_SciPost Physics, doi:SciPost Phys. 16, 053 (2024)_

## Round 2 · Referee Report · Anonymous (Referee 1) · 2023-11-7

Report

The manuscript provides an important progress towards a physical understanding of the “holographic categorical symmetry,” which relates a global symmetry in n-dimensions to a topological order with a gapped boundary in (n+1)-dimensions. In particular, the authors show that the mathematical data describing the topological order in (n+1)-dimensions can also be understood as describing “quantum currents” of the physical system in n-dimensions. The concept of “quantum currents” appears to be novel and insightful. The emphasis is on 1-dimensional models, with a possible generalization to higher dimensions discussed at the end.

The authors also discuss a way of constructing commuting projector Hamiltonian lattice models with a categorical symmetry, which are at the fixed point of the lattice renormalization group flow which is rigorously defined in the manuscript. The concept of “condensed quantum currents” is introduced, which holds the same mathematical information as a gapped boundary of a topological order in (n+1)-dimensions.

Overall, the manuscript is very well-written, and contains interesting results together with many explicit examples to guide the readers through various mathematically abstract concepts. The referee thus recommends publication of the manuscript in SciPost.

Requested changes

Below are minor suggestions:
- It appears that the notation for the objects in the Drinfeld Center in Eq. (2.24) follows that in Appendix A, rather than the notation of Definition 2.10. It will be helpful if Appendix A is explicitly referred to around Eq. (2.24).
- A lattice version of Noether current had been previously discussed in the literature, for instance in https://arxiv.org/pdf/2201.01327.pdf. It might be helpful if the relation to the current work (if any) is briefly mentioned.
- In the construction of fixed point models based on Frobenius algebra objects, it might be interesting to briefly mention how one may realize different symmetry protected topological phases within the general framework, when the symmetry is not spontaneously broken.

Below please find some typos:
- Below (2.19) “orthogonal complement *of* ker U in V”
- Below (4.34) “we can *similarly* extract”
- Definition 4.13 “a quantum current (Q,\beta) is a collection *of* symmetric operators”
- Theorem 5.12 “Given *a* Frobenius algebra”
- Top of page 44, “… at fixed-point is no *longer* C”

  • validity: top
  • significance: top
  • originality: top
  • clarity: top
  • formatting: perfect
  • grammar: perfect

Author:  Tian Lan  on 2023-11-16  [id 4120]

(in reply to Report 1 on 2023-11-07)

We thank the referee very much for the positive assessment.

We will revise the manuscript according to the suggestions: 1. The realization of 1+1D symmetry protected topological phases is already mentioned in the current version in section 6.2, using the Frobenius algebra corresponding to $(H\subset G, \omega_2 \in H^2(H,U(1))$ constructed in Example E.10. Here $H$ is the unbroken subgroup and $\omega_2$ describes the SPT invariant. For symmetry unbroken case one just takes $H=G$. We will make this part more explicit in the revision. 2. We will add the following paragraph in the introduction discussing the relation between our quantized current and the previous works: We like to comment on the difference between our formulation and the traditional notion of current in quantum mechanics or current operator in quantum field theory. The current carried by a charged quantum particle is traditionally defined as the charge times the probability current. Based on such notion, one can only conclude that the expectation value of the charge is locally conserved. Similarly, a current operator in quantum field theory is such an operator that its expectation value (correlation function) satisfies a local conservation condition. We consider these traditional notions of current as only "semi-classical", in that (1) the local conservation is only satisfied on average, at a macroscopic or statistical level; (2) they can not be used to deal with the discrete or quantized charge transport in a single quantum mechanical process; and (3) they usually require continuous space-time and continuous symmetry. There are recent works [2201.01327] extending the "semi-classical" current to lattice systems. Our formulation, on the contrary, is truly quantum: it can apply to discrete space and discrete symmetry, and can be used to analyse quantized charge transport exactly instead of on average. 3. We also plan to add clarifications to some arguments and remarks.

---

## Round 2 · Referee Report · Anonymous (Referee 2) · 2023-12-21

Strengths

  1. The authors proposed a notion of quantum current, which provides a new way of understanding the relation between symmetries in 1+1 dimensions and topological orders in one higher dimension.
  2. Condensation of the quantum currents is analyzed in detail in fixed-point models.
  3. Several concrete examples are discussed thoroughly, which helps the reader to understand the general concepts proposed in the paper.
  4. The writing of the manuscript is mostly clear.

Weaknesses

  1. Several statements are confusing and imprecise (please see the requested changes below).

Report

This paper provides a new perspective on understanding the correspondence between symmetries in 1+1d and topological orders in 2+1d. Although the relation itself is well-known and has been studied extensively in the literature, the viewpoint taken in this paper seems to be original. In particular, for general 1+1d lattice systems with a group symmetry $G$, the authors pointed out that quantum currents, which are $G$-symmetric extended operators carrying a charge from one end to the other without affecting the charge in the middle, are in one-to-one correspondence with objects of the Drinfeld center of the representation category $\mathrm{Rep}G$, which describes the topological order in one higher dimension. In addition, for more general finite symmetries, the authors showed that condensed quantum currents (i.e., quantum currents that commute with the Hamiltonian) of 1+1d fixed-point models form Lagrangian algebras, which are associated with topological boundary conditions of the bulk topological order.

The results of this paper are insightful and shed new light on the correspondence between symmetries and topological orders. As the authors have noticed in the manuscript, the quantum currents defined in this paper are very similar to patch operators discussed in arXiv:1912.13492 and 2203.03596. However, the analysis in this paper looks more systematic. As such, this paper would be a valuable addition to the existing literature. Thus, I would recommend this manuscript for publication.

Requested changes

The following is the list of questions that would be good to address before publication. 1. Symmetry is usually defined by the category of symmetry operators/defects, cf. arXiv:1412.5148. What is the category of symmetry operators/defects when the objects of a general fusion category $\mathcal{C}$ are interpreted as symmetry charges as mentioned in Remark 2.9? 2. In Section 5, when $\mathcal{C}$ is a general fusion category, what is the state space of the model constructed from a Frobenius algebra object $A \in \mathcal{C}$? Do you assume that $\mathcal{C}$ is realized as the representation category of a (weak) Hopf algebra as in Remark 2.9 so that one can think of $A$ as a vector space? 3. In the paragraph above Example 5.36, the last sentence "The ground state breaks symmetry unless A is the trivial algebra" is confusing because the ground state subspace on a periodic chain would be the center of $A$ if $A$ is a semisimple algebra over $\mathbb{C}$, cf. arXiv:hep-th/9212154. Do you claim that the ground states on an infinite chain break the symmetry even if the center of $A$ is one-dimensional? 4. It seems that defects analyzed in Section 5 are implicitly identified with excitations in Section 6. Could you clarify how the defects in Section 5 relate to the excitations of the fixed-point models? 5. In Remark 6.15, why is it the case that "if we begin with $\mathcal{C} = \mathrm{Vec}_{S_3}$, we will obtain the four models in the same phases as those obtained from $\mathrm{Rep} S_3$"? I suppose the models obtained from $\mathrm{Vec}_{S_3}$ and $\mathrm{Rep} S_3$ have different symmetries, and thus, they should be in different phases. More specifically, the models obtained from $\mathrm{Rep} S_3$ should have $\mathrm{Vec}_{S_3}$ symmetry, while those obtained from $\mathrm{Vec}_{S_3}$ should have $\mathrm{Rep} S_3$ symmetry. The gapped phases of these models would not be the same but related by the gauging of $S_3$ symmetry. In particular, gapped phases with $\mathrm{Vec}_{S_3}$ symmetry and $\mathrm{Rep} S_3$ symmetry have different numbers of ground states on a circle, cf. arxiv:2310.03784.

Please also find minor comments below: 1. It would be more common to call "a trivial bimodule" in the caption of Table 2 a regular bimodule. The same comment applies to the sentence above (6.15). 2. It would be helpful to add a reference to Remark 6.1, e.g., ref [31] of the current manuscript.

There are also some small typos: 1. The left-hand side of (2.3) would be a typo for $gf$. 2. Below (2.19), $U^{\dagger}U = \mathrm{id}_W$ would be a typo for $U^{\dagger}U = \mathrm{id}_V$. 3. In the last table on page 11, "Representation of centrlizer subgroup" would be a typo for "Representation of centralizer subgroup". 4. Below (3.10), $v_V^{a; n}$ would be a typo for $v_V^{a; k}$. 5. In the second last line on page 36, "Moreover, to preserver" would be a typo for "Moreover, to preserve". 6. In Theorem 5.35, "bimodule maps in ${}A \mathcal{C}$}^{\prime" would be a typo for "bimodule maps in ${}A \mathcal{C}$}". 7. In Table 4 on page 53, for $A = \mathrm{Fun}(S_3/\mathbb{Z}_2)$, $\mathcal{C}_A$ would be $\mathrm{Rep} \mathbb{Z}_2$ and the unbroken symmetry would be $\mathbb{Z}_2$. Similarly, for $A = \mathrm{Fun}(S_3/\mathbb{Z}_3)$, $\mathcal{C}_A$ would be $\mathrm{Rep} \mathbb{Z}_3$ and the unbroken symmetry would be $\mathbb{Z}_3$. 8. In Example 6.14, $\mathbb{C}$ in "$\mathrm{Fun}(S_3/\mathbb{Z}_3, \mathbb{C})$" would be unnecessary.

  • validity: high
  • significance: high
  • originality: high
  • clarity: high
  • formatting: excellent
  • grammar: -

Author:  Tian Lan  on 2023-12-22  [id 4202]

(in reply to Report 2 on 2023-12-21)

We thank the referee for the postive comments and really careful reading of our manuscript. Our reply to the questions: 1.2. A general fusion category can always be realized as the representations of (weak) Hopf algebras, and one can think the (weak) Hopf algebra as the global symmetry operators. The representations and the (weak) Hopf algebras have underlying vector spaces. The problem with a genuine weak Hopf algebra is that the local tensor product structure of the lattice system is no longer the usual one. Given two representations over a weak Hopf algebra, the usual vector space tensor product of them is no longer a representation. Instead, one needs to take the relative tensor product over a subalgebra of the weak Hopf algebra. We will add another remark in the resubmission explaining this issue. 3. Thanks for pointing out this. The typo is modified to "The ground state breaks symmetry unless A is a Morita trivial algebra". Indeed, Morita equivalent algebras have the same center. 4. This has been clarified in the beginning of 5.4, "excitations are viewed as defects in the same model (i.e., between A and A)". Excitations are always a special kind of defects. The converse is true generically, but may not be true in a specific lattice model, when the lattice model is not large (or universal) enough to contain all possible types of excitations. Our univesal model 6.2 is large enough for a finite group $G$ by Remark E15, and contain all possible excitation/defect types. 5. We have modified the statement to "we will obtain the four models whose emergent symmetries are in one-to-one correspondence with those obtained from $\mathrm{Rep} S_3$". The referee is correct that models with different explicit symmetry, related by gauging or duality, are not physically the same phase. What we want to emphasize here is the correspondence of emergent symmetries bewteen Morita equivalent fusion categories.

The minor comments and typos will also be incorporated in the resubmission.

---

## Round 3 · Referee Report · Anonymous (Referee 2) · 2024-1-14

Report

The authors have addressed all the questions raised in my previous report. I would thus recommend the paper for publication in its current form.

---

## Round 3 · List of Changes

1. The terminology that the quantum current is "condensed" is changed to "superconducting". A superconducting quantum current can transport charges over a long distance without costing any energy. This terminology“superconducting” fits better with the idea of quantum current than condensation.
  2. A paragraph and several sentences are add in the introduction commenting on the relation to the traditional notion of current in quantum mechanics, and other recent related works.
  3. A sentence is added in Section 6.2 on the physical meaning of $(H,\omega_2)$. Table 1 is revised with a new entry on how to realize phases with nontrivial $\omega_2$.
  4. A new remark is added under Convention 1, explaining the case when $\mathcal C$ is a generic fusion category and the relation to weak Hopf algebras.
  5. More physical explanations are added around the definition of quantum current.
  6. Other minor modifications and improvements.

---

## Editorial Decision

published